# *WizardMath*: Empowering Mathematical Reasoning for Large Language Models via *Reinforced Evol-Instruct*

**Haipeng Luo**[1*]  **Qingfeng Sun**[2*]  **Can Xu**[2†]  **Pu Zhao**[2]  **Jianguang Lou**[2]  **Chongyang Tao**[2]
**Xiubo Geng**[2]  **Qingwei Lin**[2]  **Shifeng Chen**[3†]  **Yansong Tang**[1†]  **Dongmei Zhang**[2]
[1]Shenzhen International Graduate School, Tsinghua University
[2]Microsoft Corporation
[3]Shenzhen Institute of Advanced Technology, Chinese Academy of Sciences
{luohp24@mails., tang.yansong@sz.}tsinghua.edu.cn
{caxu,qins,puzhao,jlou,chotao,xigeng,qlin,dongmeiz}@microsoft.com
{shifeng.chen}@siat.ac.cn

## Abstract

Large language models (LLMs), such as GPT-4, have shown remarkable performance in natural language processing (NLP) tasks, including challenging mathematical reasoning. However, most existing open-source models are only pre-trained on large-scale internet data and without math-related optimization. In this paper, we present *WizardMath*, which enhances the mathematical CoT reasoning abilities of LLMs without using external python tools, by applying our proposed *Reinforcement Learning from Evol-Instruct Feedback* (**RLEIF**) method to the domain of math. Through extensive experiments on two mathematical reasoning benchmarks, namely GSM8k and MATH, we reveal the extraordinary capabilities of our model. Remarkably, *WizardMath*-Mistral 7B surpasses top-tier open-source LLMs by a substantial margin with higher data efficiency. Furthermore, *WizardMath* 70B even outperforms GPT-3.5-Turbo, Claude 2, Gemini Pro and GPT-4-early-version. Additionally, our preliminary exploration highlights the pivotal role of instruction evolution and process supervision in achieving exceptional math performance. For more details refer to https://github.com/nlpxucan/WizardLM.

## 1 Introduction

Recently, Large-scale language models (LLMs) have garnered significant attention and become the go-to approach for numerous natural language processing (NLP) tasks, including open domain conversation (Ouyang et al., 2022; OpenAI, 2023; Touvron et al., 2023a), coding (Chen et al., 2021; Wang et al., 2021; Li et al., 2023b) and math (Taylor et al., 2022; Lewkowycz et al., 2022; Shao et al., 2024; Yang et al., 2024). A conspicuous example is ChatGPT[1] , developed by OpenAI. This model uses extensive pre-training on large-scale internet data and further fine-tuning with specific instruction data and methods. As a result, it achieves state-of-the-art zero-shot performance on various benchmarks. Subsequently, Anthropic, Google, and Meta also launched their competitive products one after another. Notably, Meta's series of Llama (Touvron et al., 2023a;b; Dubey et al., 2024) have sparked an open-source revolution and quickly narrowed the gap with those closed-source LLMs. This trend also gradually stimulates the releases of Mistral (Jiang et al., 2023), Alpaca (Taori et al., 2023), Vicuna (Chiang et al., 2023), and WizardLM (Xu et al., 2023), etc. However, these open models still struggle with the scenarios which require complex multi-step quantitative reasoning, such as solving mathematical and science challenges (Ahn et al., 2024; Long et al., 2024).

Chain-of-thought (CoT) (Wei et al., 2022) proposes to design better prompts to generate step-by-step solutions, which can lead to improved performance. Self-Consistency (Wang et al., 2022)

---

[*]  Equal contribution. Work done during the internship of Luo at Microsoft Research.
[†]  Corresponding author.

[1]  https://openai.com/

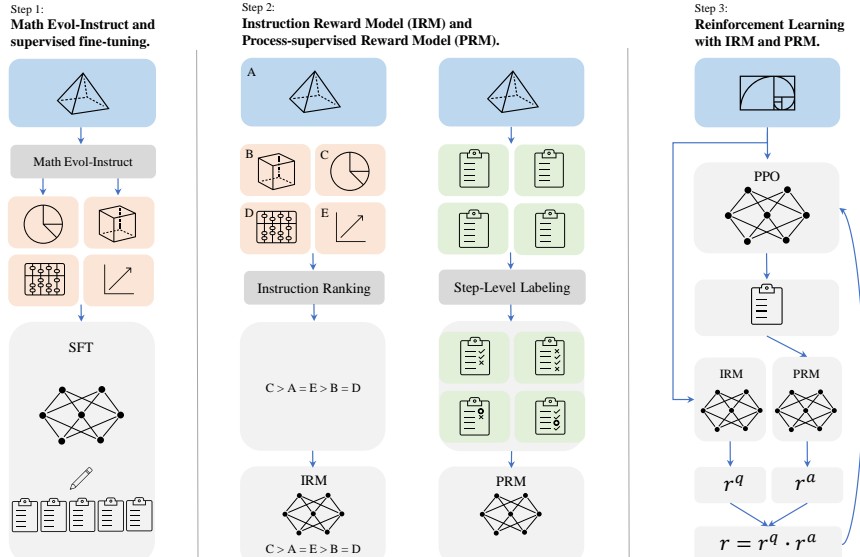

Figure 1: A diagram illustrating the three steps of our *Reinforcement Learning from Evol-Instruct Feedback* (**RLEIF**). For a detailed explanation of the training pipeline, refer to Appendix A.6

also achieves remarkable performance on many reasoning benchmarks, which generates several possible answers from the model and selects the correct one based on majority vote (Fu et al., 2022). Llemma (Azerbayev et al., 2023) and MathPile (Wang et al., 2023c) continue pretraining LLMs with math corpus to improve domain capacity. MetaMath (Yu et al., 2023b) and Xwin-Math (Li et al., 2024a) bootstraps mathematical questions by augmenting the question from multiple perspectives. MAmmoTH (Yue et al., 2023) and TORA (Gou et al., 2023) presents a unique hybrid of CoT and program-of-thought (PoT) to ensure extensive coverage of diverse fields in math. Recently, Evol-Instruct is an effective method for large-scale data synthesis using LLMs. It has been widely verified and proven to be effective in enhancing the model's instruction following capability. It employs In-depth Evolving and In-breadth Evolving to automate the generation of diverse and complex open-domain instructions using LLMs, instead of relying on human-crafted instruction datasets. In-depth Evolving incrementally enhances instruction complexity by introducing additional constraints, deepening, concretizing, increasing reasoning steps, and complicating input. In-breadth Evolving focuses on improving topic diversity and dataset richness by creating entirely new instructions. To enhance the correctness of each step in the model's generation process, (Wang et al., 2024a; Chen et al., 2024a; Lightman et al., 2023) finds that process supervision with reinforcement learning significantly outperforms outcome supervision for solving challenging MATH problems.

Inspired by *Evol-Instruct* and Process-supervised Reinforcement Learning, this work aims to enhance the mathematical reasoning abilities of the LLMs. As shown in the Figure 1, we propose a new method named *Reinforcement Learning from Evol-Instruct Feedback* (**RLEIF**), which could firstly generate diverse math instructions data by brand-new *Math Evol-Instruct*, which includes two downward evolution and upward evolution progress to produce the grade school math and challenging high school math respectively. However different from WizardLM (Xu et al., 2023) and WizardCoder (Luo et al., 2023), which mainly focus on the SFT stage and are susceptible to learning hallucinated information from the teacher model, we innovatively introduce PRM to address the False-Positive issue in the problem-solving process. Moreover, to prevent instruction evolution from spiraling out of control, we incorporate an instruction reward model (IRM) as a mitigating strategy. Thus, we train an instruction reward model (IRM) and a process-supervised reward model (PRM) (Lightman et al., 2023; Uesato et al., 2022; Wang et al., 2024a; Chen et al., 2024a), the former indicates the quality of the evolved instruction and the latter offers feedback for each reasoning step in the solution. Initially, we finetune LLMs with the evolved math data. Immediately, we leverage GPT-4 to produce the ranking order of instructions, and the correctness of each reasoning step, then optimize the LLMs to obtain the reward models. Finally, we implement the step-by-step PPO to train our *WizardMath*.

We perform experiments on two widely used mathematical reasoning benchmarks, namely GSM8k (Cobbe et al., 2021) and MATH (Hendrycks et al., 2021) covering math problems from grade to high school levels, the results show that our *WizardMath* outperforms all other open-source LLMs at the same model size, achieving state-of-the-art performance. For instance, *WizardMath*-70B significantly outperforms MetaMath-70B by a significant margin on GSM8k (92.8 vs. 82.3) and on MATH (58.6 vs. 26.6). Specifically, *WizardMath*-Mistral-7B observed a substantial improvement in pass@1 with an increase of +12.8 (90.7. vs. 77.9) on GSM8k, and +26.8 (55.4 vs. 28.6) on MATH compared to MetaMath-Mistral-7B. Notably, our 70B model even also significantly surpasses those powerful proprietary LLMs, such as GPT-3.5-Turbo, Claude 2 (Bai et al., 2022), Mistral Medium (Jiang et al., 2024), Gemini-Pro (Team, 2023), PaLM-2 (Anil et al., 2023) and GPT-4-early-version.

The main contributions of this work are as follows:

- We introduce *WizardMath* model, which enhances the LLMs' mathematical reasoning abilities across a range of problem difficulties, from grade to high school levels.
- We propose a new fully AI-powered automatic reinforcement learning method, *Reinforcement Learning from Evol-Instruct Feedback* (**RLEIF**), alongside *Math Evol-Instruct* and Process Supervision, for improving reasoning performance.
- *WizardMath* surpasses top-tier open-source LLMs by a substantial margin with higher data efficiency and also significantly outperforms various proprietary LLMs on both GSM8k and MATH, demonstrate the effectiveness of our **RLEIF**.

## 2   RELATED WORK

**Large Language Models.** LLMs have significantly advanced Natural Language Processing, with models like OpenAI's GPT Series (Brown et al., 2020a; OpenAI, 2023), Anthropic's Claude (Bai et al., 2022), Google's PaLM (Chowdhery et al., 2022; Anil et al., 2023), Gemini (Team, 2023), and Gemma (Team et al., 2024) featuring billions of parameters and trained on massive textual datasets. The AI field has also seen a rise in open-source LLMs such as Mistral (Jiang et al., 2023), Llama Series (Touvron et al., 2023a;b; Dubey et al., 2024; Taylor et al., 2022), DeepSeek (Bi et al., 2024; Shao et al., 2024), Qwen (Bai et al., 2023; Yang et al., 2024) etc. Notably, Llama serves as a foundational model for supervised fine-tuning, leading to the development of models like Alpaca, Vicuna (Taori et al., 2023; Chiang et al., 2023).

**Large Language Models For Mathematical reasoning.** NLP models face challenges with complex reasoning, including mathematical (Long et al., 2024; Zhang et al., 2024b; Xia et al., 2024), commonsense (Talmor et al., 2019). Significant research focuses on Mathematical Word Problems (MWP), which demand understanding of mathematical concepts and multi-step reasoning (Zheng et al., 2023; Zhao et al., 2023; Yuan et al., 2023a). Models are tested on various MWP benchmarks (Roy & Roth, 2015; Hendrycks et al., 2021). Techniques like Chain-of-Thought Prompting (Wei et al., 2022), Least-to-Most prompting (Zhou et al., 2022), and Complex CoT (Fu et al., 2022) enhance reasoning by introducing multiple steps and breaking problems into sub-problems. There are some models aimed at improving math CoT reasoning skills such as MetaMath (Yu et al., 2023b), MathScale (Tang et al., 2024), Xwin-Math (Li et al., 2024a), DART-Math (Tong et al., 2024) etc. Some models enhance mathematical reasoning by integrating python tools, such as TORA (Gou et al., 2023), MAmmoTH (Yue et al., 2023), Openmathinstruct (Toshniwal et al., 2024), NuminaMath (Li et al., 2024b) etc. In our work, we mainly improve the CoT reasoning ability of mathematics without using external Python tools.

**Reinforcement Learning for Large Language Models.** State-of-the-art models often display logical errors and illusions, particularly in domains requiring complex, multi-step reasoning, leading to significant challenges (Bubeck et al., 2023; Maynez et al., 2020). Strategies such as training reward models help discriminate between desirable and undesirable outputs (Lightman et al., 2023; Wu et al., 2023; Chen et al., 2024b). Historically, outcome-based approaches focused on algorithmic tasks (Li et al., 2016; Cai et al., 2017; Yu et al., 2023a), while recent research demonstrates the efficacy of reward models or validators in enhancing model performance (Cobbe et al., 2021; Wang et al., 2023a;b; Li et al., 2022). Reward models have also been incorporated into reinforcement learning pipelines and employed in rejection sampling to align Large Language Models (LLMs) with human preferences (Shen et al., 2021; Bai et al., 2022; Yuan et al., 2023b; Dong et al., 2023;

Song et al., 2023; Touvron et al., 2023b; Rafailov et al., 2024; Meng et al., 2024). A contrast is drawn between outcome-supervised and process-supervised reward models, with the latter being more effective at addressing discrepancies arising from incorrect reasoning paths leading to correct outcomes (Uesato et al., 2022; Zelikman et al., 2022; Creswell et al., 2022). Recent advances have promoted process-based supervision through manual annotation, significantly benefiting LLMs over outcome-based approaches (Lightman et al., 2023; Wang et al., 2024a; Sun et al., 2024; Chen et al., 2024a; Wang et al., 2024b; Zhang et al., 2024a). In our study, we leverage AI models like ChatGPT to automatically offer process annotation to improve the efficiency of this research line.

## 3 METHOD

In this section, we elaborate on the details of our *WizardMath*. Following WizardLM and PRMs (Lightman et al., 2023), we propose *Reinforcement Learning from Evol-Instruct Feedback* (**RLEIF**) method, which integrates the math *Evol-Instruct* and reinforced instruction and process supervision to evolve GSM8k and MATH, and fine-tune the pre-trained language models with the evolved data and reward models.

### 3.1 MATH EVOL-INSTRUCT

Motivated by the Evol-Instruct (Xu et al., 2023) method proposed by WiazrdLM and its effective application on WizardCoder (Luo et al., 2023), this work attempts to make math instructions with various complexities and diversity to enhance the pre-trained LLMs. Specifically, we adapt Evol-Instruct to a new paradigm including two evolution lines:

1) Downward evolution: It enhances instructions by making the questions easier. For example i): revising high difficulty questions to lower difficulty, or ii) producing a new and easier question with another different topic.

2) Upward evolution: Derived from original Evol-Instruct method, it deepens and generates new and harder questions by i) adding more constraints, ii) concretizing, iii) increasing reasoning.

The complete prompts of above evolution are shown in Appendix A.1. For each instruction, we use GPT-4 to evolve 5 rounds (2 downward and 3 upward) of new instructions progressively, each new one is generated by the previous round of evolution.

### 3.2 REWARD MODELS

Considering the necessity of quality control for evolved instructions and inspired by PRMs (Lightman et al., 2023), we train two reward models to predict the quality of the instructions and the correctness of each step in the answer respectively:

**Instruction Reward Model (IRM)**   This model aims to judge the quality of the evolved instructions on two aspects: i) Difficulty, and ii) Definition. To produce the ranking list training data of IRM, we leverage GPT-4 to rank the quality between those evolved instructions and original instruction. The one with high difficulty and clear definition will deserve a higher ranking. The detailed prompt of above ranking process is shown in the Appendix A.2.

Specifically, given an math instructions $q$, IRM ($Q \rightarrow \mathbb{R}$) assigns a score to $q$ to indicate its quality. We optimize ORM via the following pairwise ranking loss:

$$\mathcal{L}_{IRM} = -\log \sigma(r_j^q - r_k^q - m) \tag{1}$$

where $r_j^q$ is the reward of chosen instruction and $r_k^q$ is the reward of rejected instruction, $m$ is the margin.

**Process-supervised Reward Model (PRM)**   As there is no simple way to support highly precise process supervision without professional and expensive human-labelers, we depend on GPT-4 to provide process supervision, and ask it to assess the correctness of each step in the solutions generated by our model to produce PRM training data. The detailed prompt of above step level labeling process is shown in the Appendix A.3.

For exactly, given an math instructions $q$ and its answer $a$, PRM ($Q \times A \to \mathbb{R}^+$) assigns a score to each step of $a$, we train PRM with the following cross-entropy loss:

$$\mathcal{L}_{PRM} = \sum_{i=1}^{L} y_i \log r_i^a + (1 - y_i) \log(1 - r_i^a) \tag{2}$$

where $L$ is the reasoning steps of answer $a$. $y_i$ is the ground-truth label of the $i$-th step of answer $a$, $y_i = 1$ if $a_i$ is correct, otherwise $y_i = 0$. $r_i^a$ is the reward score (assigned by PRM) of the $i$-th step of answer $a$.

### 3.3 REINFORCEMENT LEARNING WITH IRM AND PRM

Immediately, we exploit reinforcement learning to optimize LLMs. Following (Lightman et al., 2023), we employ step by step Proximal Policy Optimization (PPO) to reward both instruction and each reasoning step.

For each math instruction $q$ and generated answer $a$, we use IRM to assign instruction reward $r^q$, and use the minimum score across all reasoning steps to represent the final reward score $r^a$ of the answer $a$ assigned by PRM. Then we apply a product as the final reward of this instruction-answer pair:

$$r = r^q \cdot r^a \tag{3}$$

### 3.4 PRM FOR VERIFICATION

Following (Lightman et al., 2023) and (Li et al., 2023c), we leverage both majority voting and PRM verifier to aggregate the predictions of different reasoning paths.

$$\hat{a} = \arg\max_a \sum_{i=1}^{N} \mathbb{I}_{a_i=a} \cdot PRM(q, a_i) \tag{4}$$

where $PRM(q, a_i)$ is the score of the $i$-th reasoning path assigned by PRM for instruction $q$. $\mathbb{I}_{a_i=a}$ is an indicator function that returns 1(or 0) if $a_i = a$.

## 4 EXPERIMENT

This section provides a comprehensive overview of the advanced models. Subsequently, we mainly elucidate the performance metrics of our models on two prevalent mathematical benchmarks from grade to high school problems: GSM8k (Cobbe et al., 2021) and MATH (Hendrycks et al., 2021).

### 4.1 EXPERIMENTAL SETUP

**SFT Training Data.** Firstly, use the GSM8k and MATH training sets as the initial seed collection, then employ both upward and downward math Evol-Instruct approach for five rounds. Each round need to evolve the initial instructions 6 times, and the temperature parameter is set to 0.7. Next, we remove duplicate instructions 17k. Hence, a total of 448k unique instructions were obtained. Subsequently, 30k data were excluded by the data filtering method to avoid contamination, ultimately leaving 418k data. Finally, we use GPT-4-0613 to generate the answer with a step-by-step format, and leverage them for supervised fine-tuning.

**Reward Models Training Data.** To train the reward models, We conducted additional 5 rounds of evolution on the initial instruction set and obtain 90k instructions. we use GPT-4-0613 to rank each instruction list with the quality from 1 to 6 as the training data of IRM. To obtain the training data of PRM, We use our Llama-2 70B SFT model to generate 5 answers for each instruction, and GPT-4-0613 is employed to assign correctness judgement for each reasoning step.

**Implementation Details.** We employ our method on two open-source foundational models Llama 2 (Touvron et al., 2023b) and Mistral-7B (Jiang et al., 2023). Llama 2 encompasses three distinct parameter sizes: 7B, 13B, and 70B. We utilize GPT-4-0613 for instruction evolution and the training data construction of reward models. For SFT, we train 3 epochs, and the learning rate is 2e-5, 1e-5 and 5e-6 for Llama 2 7B/13B, 70B and Mistral-7B. The batch size is 512, and the sequence length is 2048. For the reward model, we train Llama 2 and Mistral-7B with learning rate 4e-6 and 1e-6 for one epoch. For RL, the lr is 4e-7 and 1e-7 for Llama 2 and Mistral-7B and train one epoch.

## 4.2 MAIN RESULTS

Table 1 shows the CoT (Wei et al., 2022) pass@1 results of the current state-of-the-art models on GSM8k and MATH. In this study, to ensure equitable and cohesive evaluations, we report the socres of all models within the settings of **greedy decoding and CoT without using any external python tool**.

**Comparing with the proprietary Models.** As shown in the Table 1, our *WizardMath* demonstrates notable superiority over various proprietary LLMs on the GSM8k and MATH benchmarks in terms of pass@1:

1) *WizardMath-Llama 70B*, the largest model, demonstrated exceptional performance on the GSM8k and MATH , surpassing earlier versions of GPT-4, Claude-2, and Gemini Pro, and performing on par with GPT-4-0314. It significantly outperformed GPT-3.5-Turbo by 11.2% on GSM8k and by 15.5% on MATH.

2) *WizardMath-Mistral 7B*, the smaller-sized model, outperformed Baichuan 3 on GSM8k (90.7 vs. 87.6) and surpassed GPT-4-0314 on MATH (55.4 vs. 52.6), significantly exceeding the performance of GPT-3.5-Turbo and Gemini Pro. Meanwhile, WizardMath-Mathstral, trained on Mathstral-7B-v0.1, demonstrated performance comparable to GPT-4-turbo-0125. Additionally, WizardMath-Qwen, trained on Qwen2.5-Math, surpassed GPT-4-2024-0513 on MATH (77.8 vs. 76.6).

**Comparing with the Open-Source Models.** The results presented in Table 1 unequivocally indicate that our *WizardMath-Llama 70B* exhibits a significant performance superiority over strong models in both the GSM8k and MATH benchmarks with higher data efficiency across the range from 0.1B to 70B parameters. The detailed results are as follows:

1) With the same model parameter size, our model surpasses the previous best model such as MetaMath, MAmmoTH2-Plus, Xwin-Math. Particularly, *WizardMath-Llama 70B* achieves a substantial improvement of 10.5% on GSM8K and 32.0% on MATH compared to MetaMath-Llama 70B in testing accuracy. In the Table 2, we show the detailed results of MATH subtopics with our WizardMath 70B model. Specifically, *WizardMath-Mistral 7B*

Table 1: The models' CoT pass@1 results on GSM8k and MATH without using any external python tool.

| Model | Base | Params | GSM8k | MATH |
|---|---|---|---|---|
| *Proprietary models* | | | | |
| GPT-o1 (OpenAI, 2023) | - | - | - | 94.8 |
| GPT-o1-mini | - | - | - | 90.0 |
| Gemini-1.5 002 | - | - | - | 86.5 |
| Claude 3.5 Sonnet (Bai et al., 2022) | - | - | 96.4 | 71.1 |
| GPT-4o-2024-0513 | - | - | 96.1 | 76.6 |
| GPT-4-turbo-0125 (OpenAI, 2023) | - | - | 94.2 | 64.5 |
| GPT-4-0314 | - | - | 94.7 | 52.6 |
| GPT-4 (original version) | - | - | 92.0 | 42.5 |
| Baichuan-3 (Yang et al., 2023) | - | - | 88.2 | 49.2 |
| GLM-4 (GLM et al., 2024) | - | - | 87.6 | 47.9 |
| Gemini Pro (Team, 2023) | - | - | 86.5 | 32.6 |
| Claude2 | - | - | 85.2 | 32.5 |
| GPT-3.5-Turbo | - | - | 81.6 | 43.1 |
| PaLM2 (Anil et al., 2023) | - | - | 80.7 | 34.3 |
| Minerva (Lewkowycz et al., 2022) | - | 540B | 58.8 | 33.6 |
| GPT3.5 (Brown et al., 2020a) | - | - | 57.1 | - |
| *Open-Source Models (0.1B-3B)* | | | | |
| GPT-2-Small (Brown et al., 2020b) | - | 0.1B | 6.9 | 5.4 |
| GPT-2-Medium (Brown et al., 2020b) | - | 0.3B | 11.2 | 6.2 |
| GPT-2-Large (Brown et al., 2020b) | - | 0.7B | 13.6 | 6.4 |
| GPT-2-XL (Brown et al., 2020b) | - | 1.5B | 15.4 | 6.9 |
| WizardMath-GPT | GPT-2-Small | 0.1B | 26.4 | 12.3 |
| WizardMath-GPT | GPT-2-Medium | 0.3B | 38.7 | 15.6 |
| WizardMath-GPT | GPT-2-Large | 0.7B | 50.1 | 21.2 |
| WizardMath-GPT | GPT-2-XL | 1.5B | 58.9 | 25.4 |
| WizardMath-Qwen | Qwen-Math-2.5 | 1.5B | 86.7 | 68.6 |
| Llama-3.2-Instruct (Dubey et al., 2024) | Llama 3.2 | 1B | 44.4 | 30.6 |
| WizardMath-Llama | Llama 3.2 | 1B | 63.3 | 33.5 |
| Llama-3.2-Instruct | Llama 3.2 | 3B | 77.7 | 48.0 |
| WizardMath-Llama | Llama 3.2 | 3B | 85.5 | 49.9 |
| *Open-Source Models (7B-8B)* | | | | |
| Llama-2 (Touvron et al., 2023b) | - | 7B | 14.6 | 2.5 |
| MAmmoTH-CoT (Yue et al., 2023) | Llama-2 | 7B | 50.5 | 10.4 |
| MathScale (Tang et al., 2024) | Llama-2 | 7B | 66.3 | 31.1 |
| MetaMath (Yu et al., 2023b) | Llama-2 | 7B | 66.5 | 19.8 |
| MuggleMath (Li et al., 2023a) | Llama-2 | 7B | 68.4 | - |
| Skywork-Math (Zeng et al., 2024) | Llama-2 | 7B | 72.9 | 47.7 |
| Math-Shepherd (Wang et al., 2024a) | Llama-2 | 7B | 73.2 | 21.6 |
| Xwin-Math (Li et al., 2024a) | Llama-2 | 7B | 82.6 | 40.6 |
| WizardMath-Llama | Llama-2 | 7B | 84.1 | 43.5 |
| Mistral-v0.1 (Jiang et al., 2023) | - | 7B | 42.9 | 12.9 |
| MathScale (Tang et al., 2024) | Mistral-v0.1 | 7B | 74.8 | 35.2 |
| MMIQC (Liu & Yao, 2024) | Mistral-v0.1 | 7B | 74.8 | 36.0 |
| MetaMath (Yu et al., 2023b) | Mistral-v0.1 | 7B | 77.9 | 28.6 |
| KPMath-Plus (Huang et al., 2024) | Mistral-v0.1 | 7B | 82.1 | 46.8 |
| DART-Math (Tong et al., 2024) | Mistral-v0.1 | 7B | 82.6 | 43.5 |
| Skywork-Math (Zeng et al., 2024) | Mistral-v0.1 | 7B | 83.9 | 51.2 |
| Math-Shepherd (Wang et al., 2024a) | Mistral-v0.1 | 7B | 84.1 | 33.0 |
| MAmmoTH2-Plus (Yue et al., 2024) | Mistral-v0.1 | 7B | 84.7 | 45.0 |
| JiuZhang3.0 (Zhou et al., 2024) | Mistral-v0.1 | 7B | 88.6 | 52.8 |
| Xwin-Math (Li et al., 2024a) | Mistral-v0.1 | 7B | 89.2 | 43.7 |
| WizardMath-Mistral | Mistral-v0.1 | 7B | 90.7 | 55.4 |
| WizardMath-Mistral | Mistral-v0.3 | 7B | 90.4 | 55.6 |
| WizardMath-Mathstral | Mathstral-v0.1 | 7B | 93.8 | 70.9 |
| WizardMath-Qwen | Qwen2.5-Math | 7B | 93.9 | 77.8 |
| WizardMath-Qwen | Qwen2.5 | 7B | 94.0 | 74.5 |
| DeepSeekMath-Base (Shao et al., 2024) | - | 7B | 64.2 | 36.2 |
| NuminaMath-CoT (Li et al., 2024b) | DeepseekMath | 7B | 75.4 | 55.2 |
| MMIQC (Liu & Yao, 2024) | DeepSeekMath | 7B | 79.0 | 45.3 |
| KPMath-Plus (Huang et al., 2024) | DeepSeekMath | 7B | 83.9 | 48.8 |
| DeepSeekMath-RL (Shao et al., 2024) | DeepSeekMath | 7B | 88.2 | 51.7 |
| DART-Math (Tong et al., 2024) | DeepSeekMath | 7B | 88.2 | 52.9 |
| WizardMath-DeepSeek | DeepSeekMath | 7B | 91.0 | 64.6 |
| MetaMath (Yu et al., 2023b) | Llama 3 | 8B | 77.3 | 20.6 |
| MMIQC (Liu & Yao, 2024) | Llama 3 | 8B | 77.6 | 29.5 |
| DART-Math (Tong et al., 2024) | Llama 3 | 8B | 82.5 | 45.3 |
| MAmmoTH2-Plus (Yue et al., 2024) | Llama 3 | 8B | 84.1 | 42.8 |
| Llama 3.1-Instruct (Dubey et al., 2024) | Llama 3 | 8B | 84.5 | 51.9 |
| JiuZhang3.0 (Zhou et al., 2024) | Llama 3 | 8B | 88.6 | 51.0 |
| WizardMath-Llama | Llama 3 | 8B | 90.3 | 58.8 |
| *Open-Source Models (13B)* | | | | |
| Llama-2 (Touvron et al., 2023b) | - | 13B | 28.7 | 3.9 |
| MAmmoTH-CoT (Yue et al., 2023) | Llama 2 | 13B | 56.3 | 12.9 |
| MathScale (Tang et al., 2024) | Llama 2 | 13B | 71.3 | 33.8 |
| MetaMath (Yu et al., 2023b) | Llama 2 | 13B | 72.3 | 22.4 |
| MuggleMath (Li et al., 2023a) | Llama 2 | 13B | 74.0 | - |
| KPMath-Plus (Huang et al., 2024) | Llama 2 | 13B | 81.6 | 41.0 |
| Xwin-Math (Li et al., 2024a) | Llama 2 | 13B | 88.1 | 44.9 |
| WizardMath-Llama | Llama 2 | 13B | 89.7 | 50.6 |
| *Open-Source Models (70B)* | | | | |
| Llama-2 (Touvron et al., 2023b) | - | 70B | 56.8 | 13.5 |
| MAmmoTH-CoT (Yue et al., 2023) | Llama-2 | 70B | 72.4 | 21.1 |
| MetaMath (Yu et al., 2023b) | Llama-2 | 70B | 82.3 | 26.6 |
| KPMath-Plus (Huang et al., 2024) | Llama-2 | 70B | 87.4 | 48.6 |
| Xwin-Math (Li et al., 2024a) | Llama-2 | 70B | 90.6 | 52.8 |
| WizardMath-Llama | Llama-2 | 70B | 92.8 | 58.6 |

also surpasses top-tier open source models, outperforming MetaMath-Mistral 7B with a notable margin (90.7 vs 77.9 on GSM8k) and (55.4 vs 28.6 on MATH). It demonstrats the effectiveness

Table 2: Results of pass@1 (%) on MATH subtopics (i.e., Intermediate Algebra, Geometry) with WizardMath 70B model.

| MATH subtopics | WizardMath 70B |
|---|---|
| Intermediate Algebra | 36.3 |
| Precalculus | 38.9 |
| Geometry | 48.3 |
| Number Theory | 58.5 |
| Counting & Probability | 54.8 |
| Prealgebra | 74.6 |
| Algebra | 78.5 |
| Overall | **58.6** |

Table 3: Explore the effects of PRM and IRM during PPO training.

| Models | GSM8K | MATH |
|---|---|---|
| GPT-2-XL-1.5B: WizardMath-SFT | 51.9 | 18.3 |
| + PRM | 55.8 | 22.1 |
| + PRM + IRM | **58.9** | **25.4** |
| Llama2-7B: WizardMath-SFT | 77.4 | 35.6 |
| + PRM | 81.7 | 39.9 |
| + PRM + IRM | **84.1** | **43.5** |
| Mistral-7B: WizardMath-SFT | 82.8 | 48.1 |
| + PRM | 87.2 | 52.7 |
| + PRM + IRM | **90.7** | **55.4** |

of our RLEIF method in enhancing mathematical reasoning capabilities across a range of problem difficulties, from grade to high school levels.

2) By employing diverse pre-trained models (i.e., GPT-2, Llama 2, Mistral, Qwen, DeepSeek) as base models, WizardMath demonstrated notable advancements on the GSM8k and MATH benchmarks. Specifically, WizardMath-Llama2-7B, based on Llama2-7B, improved performance by 69.5% on GSM8k and 41.0% on MATH. Similarly, WizardMath-GPT2-XL, built on GPT2-XL, achieved a 43.5% improvement on GSM8k and 18.5% on MATH, performing on par with Llama2-70B and outperforming GPT-3.5 on GSM8k. This demonstrates that our RLEIF method is equally effective for smaller models in enhancing mathematical reasoning capabilities, proving its scalability and robustness across various model backbones.

### 4.3 ANALYSIS

**The impact of training data size**

We are curious about to how the training data size of different dataset construction methods impact the reasoning capacity of LLMs. Thus we conduct different number of training instances from ours evolved data and MetaMathQA to fine tune Mistral 7B. As shown in the Figure 2, Math Evol-Instruct achieves superior data efficiency. Specifically, our model constantly outperforms MataMath by more than 3% ~ 6% on GSM8k and 15% ~ 20% on MATH under the same number of conditions. Our findings indicate that Math Evol-Instruct exhibits a higher potential upper bound compared to MetaMath, thus demonstrating the effectiveness of Evol-Instruct for math reasoning scenario.

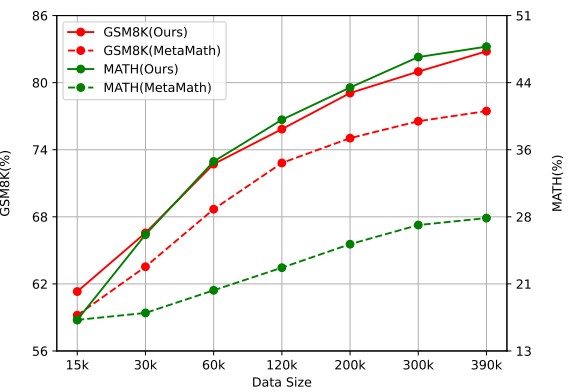

Figure 2: Accuracy of Mistral-7B fine-tuned in different sizes of augmentation data on GSM8K and MATH

**The impact of PRM and IRM during PPO training**

To verify the contributions of the instruction reward model and process-supervised reward model, we consider the following variants: (1) SFT + PRM: only use PRM in the PPO training. (2) SFT + PRM + IRM: use both IRM and PRM in the PPO training. As shown in Table 3, applying PRM alone for PPO training on GSM8k and MATH yields a 3%-4% improvement. When combined with IRM, an additional 2.5%-4% gain is observed. Thus, the integration of PRM and IRM results in a substantial overall improvement of 6%-8%. So, we can conclude that (1) PRM is crucial to WizardMath, since the variant with PRM significantly outperforms the SFT one without any PPO training (2) IRM also plays a key role in the success of reinforcement learning, as there is a remarkable improvement when

we combine PRM with IRM, further demonstrating the necessity of taking instruction's quality into account and correcting false positives in the problem-solving process when we optimize the LLMs.

Table 4: The effect of different reward models during PPO training

| Models | GSM8K | MATH |
|---|---|---|
| Llama2-7B: WizardMath-SFT | 77.4 | 35.6 |
| + ORM (ours) | 79.1 | 36.8 |
| + PRM800k | 79.7 | 38.7 |
| + Math-Shepherd | 80.3 | 38.2 |
| + PRM (ours) | **81.7** | **39.9** |
| Mistral-7B: WizardMath-SFT | 82.8 | 48.1 |
| + ORM (ours) | 84.6 | 49.6 |
| + PRM800k | 85.4 | 50.8 |
| + Math-Shepherd | 86.1 | 50.3 |
| + PRM (ours) | **87.2** | **52.7** |

Table 5: Results of reinforcement learning combined with validation. The SFT and Reward models are trained based on Mistral-7B. The verifier is based on 256 sample outputs.

| Generators | Verifiers | GSM8K | MATH |
|---|---|---|---|
| SFT | Self-Consistency | 90.7 | 57.5 |
| | ORM | 93.0 | 58.3 |
| | PRM | 93.9 | 61.7 |
| SFT + ORM | Self-Consistency | 91.2 | 57.7 |
| | ORM | 93.4 | 59.4 |
| | PRM | 94.1 | 63.3 |
| SFT + PRM | Self-Consistency | 92.3 | 59.3 |
| | ORM | 94.1 | 60.8 |
| | PRM | **95.2** | **64.7** |

**The impact of Evol-Instruct turns**. Table 6 illustrates the impact of combining downward and upward evolution in SFT training. Two rounds of downward evolution improved GSM8k by 14.8% (74.5 vs. 59.7) and MATH by 19.6% (34.7 vs. 15.1) over the original. Three rounds of upward evolution yielded a 18.9% improvement on GSM8k (78.6 vs. 59.7) and a 27.4% improvement on MATH (42.5 vs. 15.1). Furthermore, combining downward evolution based on upward evolution resulted in an additional 2.6% improvement on GSM8k (81.2 vs. 78.6), a total improvement of 21.5% over the original. Similarly, a 1.9% improvement on MATH (46.5 vs. 42.5), a 31.4% total improvement. These results underscore the complementary and significant effectiveness of upward and downward evolution.

Table 6: Impact of different Downward and Upward Evol-Instruct turns on Mistral-7B SFT. *D-i* refers to the *i* round of downward evolution, whereas *U-i* denotes the *i* round of upward evolution. *Ori* is the original manually annotated 7.5k data of GSM8k and MATH.

| Data | GSM8K | | | | | | | MATH | | | | | | |
|---|---|---|---|---|---|---|---|---|---|---|---|---|---|---|
| | Ori | D-1 | D-2 | U-1 | U-2 | U-3 | pass@1 | Ori | D-1 | D-2 | U-1 | U-2 | U-3 | pass@1 |
| Ori | ✓ | ✗ | ✗ | ✗ | ✗ | ✗ | 59.7 | ✓ | ✗ | ✗ | ✗ | ✗ | ✗ | 15.1 |
| Math Evol | ✓ | ✓ | ✗ | ✗ | ✗ | ✗ | 71.9 | ✓ | ✓ | ✗ | ✗ | ✗ | ✗ | 30.3 |
| | ✓ | ✗ | ✓ | ✗ | ✗ | ✗ | 70.5 | ✓ | ✗ | ✓ | ✗ | ✗ | ✗ | 28.7 |
| | ✓ | ✗ | ✗ | ✓ | ✗ | ✗ | 73.7 | ✓ | ✗ | ✗ | ✓ | ✗ | ✗ | 33.4 |
| | ✓ | ✗ | ✗ | ✗ | ✓ | ✗ | 71.6 | ✓ | ✗ | ✗ | ✗ | ✓ | ✗ | 32.6 |
| | ✓ | ✗ | ✗ | ✗ | ✗ | ✓ | 70.2 | ✓ | ✗ | ✗ | ✗ | ✗ | ✓ | 30.9 |
| | ✓ | ✓ | ✓ | ✗ | ✗ | ✗ | **74.5** | ✓ | ✓ | ✓ | ✗ | ✗ | ✗ | **34.7** |
| | ✓ | x | x | ✓ | ✓ | x | 77.1 | ✓ | x | x | ✓ | ✓ | x | 38.6 |
| | ✓ | x | x | ✓ | ✓ | ✓ | **78.6** | ✓ | x | x | ✓ | ✓ | ✓ | **42.5** |
| | ✓ | ✓ | ✓ | ✓ | ✗ | ✗ | 76.6 | ✓ | ✓ | ✓ | ✓ | ✗ | ✗ | 40.3 |
| | ✓ | ✓ | ✓ | ✓ | ✓ | ✗ | 79.8 | ✓ | ✓ | ✓ | ✓ | ✓ | ✗ | 44.6 |
| | ✓ | ✓ | ✓ | ✓ | ✓ | ✓ | **81.2** | ✓ | ✓ | ✓ | ✓ | ✓ | ✓ | **46.2** |

**ORM v.s. PRM; Human v.s. AI.** Table 4 presents the performance of different answer reward methods for LLMs in terms of pass@1. As is shown: 1) Our step-by-step PRM significantly enhances the performance of both Llama and Mistral based SFT models. Specifically, the Mistral-7B powered by our PRM achieves 87.2% and 52.7% on GSM8k and MATH respectively. 2) PRM models consistently outperforms ORM on both GSM8k and MATH, indicating the effectiveness of step-by-step supervision. 3) The PRM trained on our fully AI-labeled data outperforms both the manually annotated PRM800k and Math-Shepherd, which utilizes MCTS tree search for annotation. When training WizardMath-Mistral-SFT with PPO, our PRM improves upon PRM800k by 1.8% and Math-Shepherd by 1.1% on GSM8k, while surpassing PRM800k by 1.9% and Math-Shepherd by 2.4% on MATH. This demonstrates powerful AI can also provide good process supervision quality, highlighting the effectiveness of utilizing AI to construct PRM training data.

**PRM as Verifier.** Table 5 presents the performance comparison of various generators with different verifiers on GSM8K and MATH in terms of pass@256. We find that: 1) PRM verifier consistently demonstrates superior performance compared to Self-Consistency and ORM. Specifically, our SFT + PRM generator, enhanced by the PRM verifier, achieves 95.2% and 64.7% accuracy on GSM8K and MATH respectively. 2) When compared to ORM, PRM exhibits a more significant advantage on the more challenging MATH dataset which aligns with the findings in (Uesato et al., 2022) and (Lightman et al., 2023). This can be attributed to the fact that GSM8K involves fewer and less complex steps in problem-solving than MATH. 3) Particularly, the generator with PRM PPO training

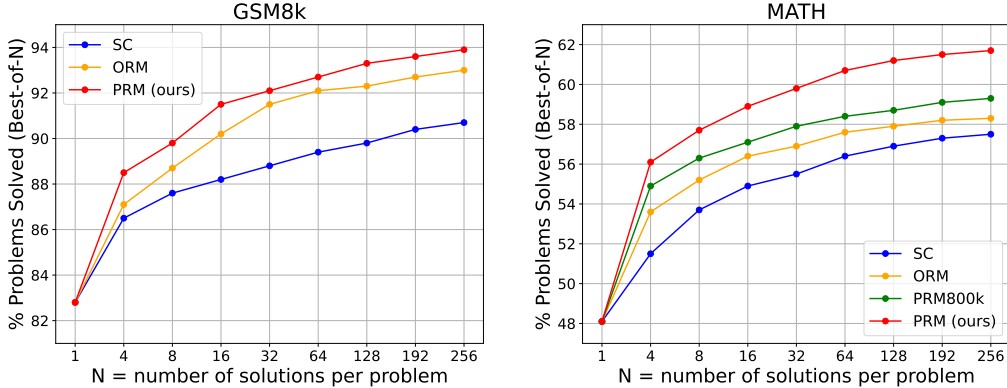

Figure 3: Performance of Mistral-7B SFT with different verification strategies.

surpasses those SFT and ORM PPO trained generators regardless of employing Self-Consistency, ORM, and the PRM verifiers. This further demonstrates the effectiveness of our PRM.

Figure 3 also shows the performance of different Verification strategies across a range of candidate numbers from 1 to 256 on two benchmarks. The main observations are as follows: 1) PRM verifiers consistently achieves superior performance compared to both ORM and majority voting, and this superiority becomes more evident as N increases. 2) For MATH benchmark, our PRM trained on the AI-annotated datasets slightly surpassed the human-annotated PRM800K.

**Performance of Out-of-Domain.** Table 7 presents the results of Wizard-Math on the 7 out-of-domain evaluation results covering K-12, college, and competition level math problems, highlighting the following salient observations: (1) With math Evol-Instruct and reinforcement learning, WizardMath consistently surpasses prior state-of-the-art open-source models (e.g. Meta-Math, MathScale) across all scales, and achieves improvement of 5%-10% across 7 tasks on average. (2) The accuracy of WizardMath-Mistral is about 5.0% higher than WizardMath-Llama on the same size. Especially it exceeds GPT-3.5-Turbo (45.7 vs. 37.9) while being comparable to GPT-4. This also indicates that Mistral-7B has more potential in mathematical reasoning. (3) Especially on difficult benchmarks (i.e., College Math, AGIE Gaokao Math), WizardMath outperforms MetaMath by a significant margin . This demonstrates our model and RLEIF method has stronger robustness and better significant generalization ability for invisible mathematical problems.

Table 7: Performance of WizardMath on the 7 out-of-domain evaluation results covering K-12, college, and competition level math problems. The results of models in the table refer to MWPBENCH (Tang et al., 2024). "AGIE" stands for AGIEval. We report the models' CoT pass@1 results on MwpBench without using any external python tool

| Models | College Math | TAL | Math23k | Ape210k | Gaokao Bench Math | AGIE Gaokao Math | AGIE SAT Math | AVG |
|---|---|---|---|---|---|---|---|---|
| *Proprietary models* | | | | | | | | |
| GPT-4 | **24.4** | **51.8** | **76.5** | **61.5** | **35.4** | **28.2** | **68.6** | **49.5** |
| GPT-3.5-Turbo | 21.6 | 42.9 | 62.5 | 44.0 | 23.2 | 15.3 | 55.8 | 37.9 |
| *Models based on LLaMA-2 13B* | | | | | | | | |
| LLaMA-2 13B | 1.2 | 6.3 | 9.5 | 7.9 | 0.7 | 0.4 | 6.8 | 4.7 |
| MAmmoTH-CoT | 6.5 | 17.3 | 39.5 | 28.1 | 5.9 | 4.9 | 20.5 | 17.5 |
| GAIR-Abel | 7.9 | 21.1 | 42.2 | 27.8 | 7.0 | 4.9 | 30.3 | 20.2 |
| MetaMath | 10.1 | 25.4 | 48.6 | 31.6 | 9.6 | 5.6 | 38.2 | 24.2 |
| MathScale 13B | 20.4 | 38.1 | 61.1 | 43.7 | 20.0 | 12.3 | 55.8 | 35.9 |
| WizardMath | 22.9 | 43.3 | 70.3 | 50.8 | 33.1 | 25.7 | 64.7 | 44.4 |
| *Models based on LLaMA-2 7B* | | | | | | | | |
| LLaMA-2 7B | 2.3 | 7.6 | 6.8 | 7.3 | 2.1 | 2.9 | 2.9 | 4.6 |
| MAmmoTH-CoT | 6.2 | 13.3 | 34.6 | 21.4 | 3.9 | 2.7 | 19.6 | 14.5 |
| GAIR-Abel | 6.6 | 18.3 | 35.4 | 24.5 | 4.3 | 4.4 | 23.5 | 16.7 |
| MetaMath | 9.4 | 22.5 | 44.0 | 29.9 | 5.9 | 5.1 | 36.2 | 21.9 |
| MathScale 7B | 20.9 | 35.2 | 59.0 | 41.8 | 19.6 | 12.6 | 57.8 | 35.3 |
| WizardMath | 21.2 | 40.2 | 67.3 | 46.1 | 28.9 | 18.7 | 62.7 | 40.7 |
| *Models based on Mistral 7B* | | | | | | | | |
| Mistral 7B | 7.5 | 17.9 | 18.5 | 15.5 | 6.2 | 5.9 | 22.5 | 13.4 |
| MetaMath Mistral | 15.7 | 31.4 | 55.1 | 38.1 | 15.3 | 10.1 | 50.9 | 30.9 |
| MathScale Mistral | 21.8 | 39.9 | 64.4 | 46.0 | 21.4 | 14.3 | 57.8 | 37.9 |
| WizardMath Mistral | 24.8 | 44.8 | 71.2 | 52.6 | 37.2 | 24.5 | 64.7 | 45.7 |

**Employ Open-source Model to Math Evol-Instruct.** In Table 19, we investigate the use of open-source models (i.e., Llama-3-70B-Instruct) as a substitute for GPT-4 during the SFT stage for Evol Instruct, employing the same evolution strategy. The results demonstrate that WizardMath-

Table 9: A case study from GSM8k test set. We rate the response using PRM and ORM. Red text denotes the wrong reasoning steps which PRM successfully detected, but ORM failed.

| Question | *The farm has 30 cows and the zoo has 20 sheep. The zoo has twice as many cows as the farm does. The farm has half as many sheep as the zoo does. How many animals do the farm and zoo have combined?*     ***Ground truth answer: 120*** |
|---|---|
| **Steps** | Step 1: we know that the farm has 30 cows. PRM Score: 0.95. 
 Step 2: we are told that the zoo has twice as many cows as the farm does. PRM Score: 0.97. 
 Step 3: So, we can calculate the number of cows in the zoo by multiplying the number of cows in the farm by 2. So, 2 * 30 = 60, So, the zoo has 60 cows. PRM Score: 0.97. 
 Step 4: we know that the farm has half as many sheep as the zoo does. PRM Score: 0.96. 
 Step 5: So, we can calculate the number of sheep in the farm by dividing the number of sheep in the zoo by 2. So, 20 / 2 = 10. So, the farm has 10 sheep. PRM Score: 0.93. 
 Step 6: Now, we can calculate the total number of animals in the farm and the zoo by adding the number of cows and the number of sheep in both places. PRM Score: 0.92. 
 Step 7: So, 30 (farm's cows) + 60 (zoo's cows) + 10 (farm's sheep) = 100. PRM Score: 0.11. 
 Step 8: So, the farm and the zoo have a total of 100 animals combined. PRM Score: 0.10. 
 Step 9: Therefore, the final answer is 100 . The answer is: 100. PRM Score: 0.06. ORM Score: 0.89. |

Llama3-Evol achieved a 33.8% improvement on GSM8k and a 30.6% improvement on MATH, indicating that the math evol instruct strategy remains effective on open-source models. However, compared to GPT-4 evolution, there is still a 5%-6% performance gap. Despite this, the strategy shows significant potential in balancing computational cost and accuracy.

Table 8: The impact of using open source models for Math-Evol and use Mistral-7B-v0.1 for SFT .

| Models | GSM8k | MATH |
|---|---|---|
| Mistral-7B-v0.1 | 42.9 | 12.9 |
| WizardMath-SFT-GPT-4-Evol | 82.8 | 48.1 |
| WizardMath-SFT-Llama3-Evol | 76.7 | 43.5 |

### 4.4 DATA CONTAMINATION CHECK

Apart from the performance analysis, we also investigate whether evolution leads to the data contamination between training data and test set. To address this consideration, we employ instructions in the GSM8k and MATH test set as queries to retrieve the top-5 samples from all evolved training data with an embedding model, gte-large (Li et al., 2023d). Additionally, we employ GPT-4 to provide similarity judgement between the test sets and the retrieved samples, and remove the top-2 similar instructions. The prompt and details are shown in Appendix A.4 and A.5. Figure 4 illustrates that the evolution process does not yield higher similarity scores.

### 4.5 CASE STUDY

**Evol-Instruct.** The Examples 3 and 4 in the Appendix A.1 shows the prompt and corresponding cases of GSM8k and MATH instruction evolution, demonstrating that the evolved instructions exhibit more complexity and diversity than the original training set.

**PRM v.s. ORM.** We present a comprehensive case study to illustrate the effectiveness of our PRM. As delineated in Table 9, PRM demonstrates precise performance on a challenge math problem from the GSM8k test set. Remarkably, our PRM effectively distinguished the incorrect solution, in the meanwhile the ORM struggled in this task. Furthermore, PRM demonstrated exceptional insight by accurately detecting the incorrect steps of the solution chosen by ORM, specifically the steps 7, 8, and 9. Subsequently, PRM also assigned lower score logits to these erroneous steps.

## 5 CONCLUSION

This paper introduces *WizardMath*, a mathematics model fine-tuned with ***RLEIF***. The experimental results demonstrate that *WizardMath* achieves SOTA performance surpassing existing open-source LLMs on GSM8k and MATH from grade to high school problems. Notably, *WizardMath* 70B exhibits superior performance compared to some of the well-known proprietary LLMs, including ChatGPT-3.5, Claude Instant, PaLM-2, Gemini Pro. Furthermore, our preliminary exploration highlights the pivotal role of instruction evolution and process supervision in achieving exceptional performance.

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

# A    APPENDIX

## A.1    MATH EVOLUTION PROMPTS

> **Example 1: Upward Evolution Prompt**
>
> **Step 1:** Understand the core concept and structure of the "#Instruction#". Identify the key elements such as variables, conditions, participants, actions, or processes that can be manipulated to increase complexity. Also, recognize the theme of the instruction and ensure it remains consistent throughout the evolution.
>
> **Step 2:** Formulate a comprehensive plan to increment the complexity of the "#Instruction#" based on the identified elements in Step 1. The plan should involve modifying or expanding at least three components from the list. It is crucial to ensure that all components in the instruction are logically interconnected and that the complexity increase is coherent and justified. The plan should avoid introducing variables or conditions without clear criteria for determining their values or without contributing to the overall complexity. In this step, consider adding more real-world constraints and dependencies between variables to make the problem more challenging. And you can also add more constraints, concretizing, increasing reasoning.
>
> **Step 3:** Implement the plan step by step to create the "#Rewritten Instruction#". Ensure the rewritten instruction maintains a logical sequence and avoids ambiguity or confusion. If additional variables or conditions are introduced, provide clear and unambiguous methods or criteria for determining their values. The "#Rewritten Instruction#" should not exceed the original "#Instruction#" by more than 30 words to ensure readability and comprehension.
>
> **Step 4:** Review the "#Rewritten Instruction#" thoroughly to identify any unreasonable elements or inconsistencies. Make sure the "#Rewritten Instruction#" is a more complex version of the "#Instruction#", and that it accurately reflects the intended increase in complexity. Adjust any part of the instruction that may lead to misunderstanding or ambiguity, and provide the "#Finally Rewritten Instruction#" without any supplementary explanation.
>
> Please reply strictly in the following format:
> Step 1
> #Elements Identified#:
> Step 2
> #Plan#:
> Step 3
> #Rewritten Instruction#:
> Step 4
> #Finally Rewritten Instruction#:
>
> #Instruction#:

---

**Example 2: Downward Evolution Prompt**

**Step 1:** Understand the "#Instruction#" and identify all the components that can be modified to decrease complexity, so that it makes the instruction easier. These components can be variables, conditions, participants, actions, etc. The key is to keep the core scenario unchanged while ensuring that any new elements introduced do not cause ambiguity or confusion.

**Step 2:** Develop a comprehensive plan to decrease the complexity of the "#Instruction#" based on the components identified in Step 1. The plan should involve modifying at least three components from the list. It is important to ensure that all components in the instruction are logically interconnected and that the complexity decrease is justifiable. The plan should avoid introducing variables or conditions without clear criteria for determining their values. Our goal is revising high difficulty questions to lower difficulty, or producing a new and easier question with another different topic.

**Step 3:** Implement the plan step by step to create the "#Rewritten Instruction#". Make sure the rewritten instruction maintains a logical sequence and avoids ambiguity or confusion. If additional variables or conditions are introduced, provide clear and unambiguous methods or criteria for determining their values. The "#Rewritten Instruction#" should not exceed the original "#Instruction#" by more than 20 words.

**Step 4:** Review the "#Rewritten Instruction#" thoroughly to identify any unreasonable elements. Make sure the "#Rewritten Instruction#" is a easier version of the "#Instruction#". Adjust any part of the instruction that may lead to misunderstanding or ambiguity, and provide the "#Finally Rewritten Instruction#" without any explanation.

Please reply strictly in the following format:
Step 1
#Elements Identified#:
Step 2
#Plan#:
Step 3
#Rewritten Instruction#:
Step 4
#Finally Rewritten Instruction#:

#Instruction#:

---

---

**Example 3: GSM8k Evol Instruction Case**

**Original Instruction 1:** Bill is trying to decide whether to make blueberry muffins or raspberry muffins. Blueberries cost $5.00 per 6 ounce carton and raspberries cost $3.00 per 8 ounce carton. If Bill is going to make 4 batches of muffins, and each batch takes 12 ounces of fruit, how much money would he save by using raspberries instead of blueberries?

**Evol Instruction 1:** Bill and Jane are contemplating between blueberry and raspberry muffins. Blueberries are $5.00 for a 6 ounce carton, with a 20% bulk discount. Raspberries are $3.00 for an 8 ounce carton. If they each make 6 batches of muffins, with each batch requiring 12 ounces of fruit, calculate the total money they would save by choosing raspberries over the discounted blueberries, given Jane's inclination towards raspberries.

**Original Instruction 2:** A snake's head is one-tenth its length. If a snake is 10 feet long, calculate the length of the rest of its body minus the head.

**Evol Instruction 2:** Given a snake's head is a certain fraction of its total length, and the snake's total length is a positive integer, determine the length of the snake's head by multiplying the total length by the fraction. Subtract this value from the total length to calculate the length of the rest of the snake's body.

**Original Instruction 3:** Thomas is training at the gym to prepare for a competition. He trained for 5 hours every day for a month (30 days). If he continues to train for the next 12 days, how many hours will he spend on training in total?

**Evol Instruction 3:** Thomas and James are preparing for a competition by training at the gym. They trained for 5 hours daily for a month (30 days), excluding a rest day each week. If they persist in training for the subsequent 12 days, adding an extra hour of training each week, what will be the total hours they have spent training?

**Original Instruction 4:** Travis is hired to take 638 bowls from the factory to the home goods store. The home goods store will pay the moving company a $100 fee, plus $3 for every bowl that is delivered safely. Travis must pay the home goods store $4 each for any bowls that are lost or broken. If 12 bowls are lost, 15 bowls are broken, and the rest are delivered safely, how much should Travis be paid?

**Evol Instruction 4:** Travis and his team are tasked with moving 1000 bowls and 500 plates from the factory to a home goods store. The store agrees to pay a $200 fee, plus $4 for each safely delivered bowl and $2 for each plate. However, Travis must compensate the store $5 for each lost or broken bowl and $3 for each plate. If they lose 20 bowls and 10 plates, and break 25 bowls and 15 plates, how much should the store pay Travis and his team?

**Original Instruction 5:** Gary is buying chlorine for his rectangular pool, which is 10 feet long, 8 feet wide, and 6 feet deep. Gary needs to buy one quart of chlorine for every 120 cubic feet of water in his pool. If chlorine costs $3 a quart, how much does Gary spend on chlorine?

**Evol Instruction 5:** Gary and John are purchasing chlorine for their cylindrical pools, with diameters of 12 feet and 10 feet, and depths of 8 feet and 6 feet respectively. They require one quart of chlorine per 100 cubic feet of pool water. Given that chlorine is priced at $4 per quart, calculate the total expenditure on chlorine for both Gary and John.

**Original Instruction 6:** Ken likes to bike when it's raining and can cycle 30 miles in 20 minutes during this time. However, when it's snowing Ken can't stand the cold and can only cycle 10 miles in 20 minutes. If it rains 3 times and snows 4 times in one week, how many miles did Ken reach if he cycles 1 hour a day?

**Evol Instruction 6:** In varying weather conditions, Ken's biking speed differs. He can cycle 30 miles in 20 minutes when it's raining, 10 miles in 20 minutes when it's snowing, and 20 miles in 20 minutes on sunny days. In a week, if it rains 4 times, snows 3 times, and is sunny 2 times, and Ken cycles for 1.5 hours each day, how many miles did he cover? Remember, after cycling for an hour, his speed decreases by 10%.

**Example 4: MATH Evol Instruction Case**

**Original Instruction 1:** Find the smallest positive integer whose cube ends in 888.
**Evol Instruction 1:** Determine the least positive whole number, denoted by 'x', whose cube terminates in 888 and is divisible by 3. Verify the result by checking the divisibility of the cube by 9.

**Original Instruction 2:** The sum of all the positive factors of integer $x$ is 24. If one of the factors is 3, what is the value of $x$?
**Evol Instruction 2:** Given that the summation of all positive factors of an integer $x$ is 24, and considering $x$ is a positive integer divisible by 3 with one of its factors being 3, determine the value of $x$ by first calculating the variable $S$ representing the sum of factors, and then solving for $x$.

**Original Instruction 3:** What is $2^{-1} + 2^{-2} + 2^{-3} + 2^{-4} + 2^{-5} + 2^{-6} \pmod{13}$? Express your answer as an integer from 0 to 12, inclusive.
**Evol Instruction 3:** Let S be the sum of the series $2^{-1} + 2^{-2} + 2^{-3} + 2^{-4} + 2^{-5} + 2^{-6}$. Calculate S by finding the sum of each term, then determine the value of $S \pmod{13}$. Utilize the properties of modular arithmetic and provide a step-by-step solution. Express the final answer as an integer from 0 to 12, inclusive.

**Original Instruction 4:** Find the greatest common divisor of 40304 and 30203.
**Evol Instruction 4:** Determine the greatest common divisor of the integers 40304 and 30203 by employing the Euclidean algorithm. Utilize prime factorization, considering the Fundamental Theorem of Arithmetic, and verify if both numbers are divisible by the same prime factors.

**Original Instruction 5:** Find the remainder when $2 \times 12 \times 22 \times 32 \times \ldots \times 72 \times 82 \times 92$ is divided by 5.
**Evol Instruction 5:** First, let $P$ represent the product of the series, which can be expressed as $P = \prod_{n=1}^{9}(2 + 10n)$. Next, calculate the value of $P$. Then, determine the remainder, denoted as $R$, when $P$ is divided by 5. Ensure that $R$ is a positive integer.

**Original Instruction 6:** Is the function $f(x) = \lfloor x \rfloor + \frac{1}{2}$ even, odd, or neither? Enter odd, even, or neither.
**Evol Instruction 6:** Determine if the function $f(x) = \lfloor x \rfloor + \frac{1}{2}$ exhibits parity (evenness or oddness) or neither, considering the mathematical definitions of even and odd functions. If $x > 0$, introduce a variable $y$ and compare $f(x)$ with $g(y) = y^2$. Provide a brief explanation for your answer.Enter f(x) is even, f(x) is odd, or f(x) is neither even nor odd.

## A.2    IRM PROMPT

---

**Example 5: Instruction Quality Ranking Prompt**

You are a senior mathematics grading teacher in university, very skilled in high difficulty fields such as Intermediate Algebra, Precalculus, Prealgebra, Number Theory, Geometry, Counting & Probability, Algebra and so on.
Your task is to act as an impartial judge to evaluate the quality of math problems based on their definition completeness and difficulty and rank a set of maths problems according to these criteria. Make sure that your assessment takes into account the following rules:

**1.** ** Problem statement completeness and correctness:**

- Assess the clarity and accuracy of the definition of each math problem. Ensure that the problem statement provides sufficient information, conditions, and constraints.
- Consider whether the problem allows for multiple interpretations or if further clarification is needed.
- Evaluate the clarity of mathematical notation and terminology used in the problem.

**2.**Conceptual difficulty:**

- Evaluates the complexity of each mathematical problem in terms of the underlying concepts involved. Ensure a solid and sound understanding of the underlying principles, or advanced mathematical concepts.
- Consider the depth of mathematical knowledge required to address and solve each problem.
- Assess whether the problem encourages critical thinking and the application of mathematical principles.

**3.**Computational complexity:**

- Examine the computational complexity of each problem. Judge whether it involves complex calculations, algebraic operations, or non-trivial numerical operations.
- Consider whether the problem requires sophisticated computational techniques or algorithms or whether it can be answered with existing mathematical knowledge.

**4.** ** Problem contextualisation:**

- Consider the relevance of each mathematical problem in the given context or practical application. as well as being relevant or having a meaningful meaning in the practical context.
- Evaluate whether the theory of the mathematical problem is detached from the facts, spurious, and non-existent.

Avoid any position biases and ensure that the order in which the math problems were presented does not influence your decision.
Do not allow the length of the problems to influence your evaluation.
Do not favor certain mathematical theory of the problems. Be as objective as possible.

Below is a list of a set of math problems that you need to rank according to the rules above from most complete and clear (1) to least complete and clear (N) based on the comprehensiveness and difficulty level of the maths problem. Also, consider the difficulty level from most challenging (1) to easiest (N). Your output needs to be placed in the <Rank> </Rank>section.
And Your output is in JSON list format, where each element is a dictionary with three keys:

- instruction: represents the math problem.
- score: represents the result of your ranking for the problem.
- reason: provide your explanation in detail for your ranking result.

[### Math Problems List ###]: <PROBLEMS_HERE >

---

| | Math instructions |
|---|---|
| GPT-4 Ranking | **Instruction 1.** Write $x^{10} + x^5 + 1$ as the product of two polynomials with integer coefficients. Rank: 2. Reason: The problem is well-defined and clear. It requires a deep understanding of polynomial factorization, making it conceptually difficult. The computational complexity is high as it involves the factorization of a high-degree polynomial.

**Instruction 2.** Express the following sum as a common fraction:

$$\frac{1}{1 \cdot 2} + \frac{1}{2 \cdot 3} + \frac{1}{3 \cdot 4} + \frac{1}{4 \cdot 5} + \cdots + \frac{1}{9 \cdot 10}.$$

Rank: 1. Reason: This problem is clear and complete, with a well-defined mathematical notation. It requires a good understanding of series and fractions, making it conceptually challenging. The computational complexity is moderate as it involves the summation of fractions.

**Instruction 3.** Given that I read 6 to 8 novels monthly, each novel having 200 to 350 pages, and considering a year may have 12 or 13 months due to a leap year or a year when I choose to read more, also taking into account that my reading speed can fluctuate between 1 to 2 novels per day, and the number of days in a month can be 28, 30, or 31, and the genre of the novel can influence the number of pages (for instance, a thriller may contain 300 to 350 pages, while a romance novel may contain 200 to 250 pages), calculate the total pages of novels I will read in a year. Rank: 3. Reason: This problem is complex and requires multiple steps to solve, making it conceptually and computationally challenging. It is well-contextualized and relevant, but the problem statement could be clearer and more concise.

**Instruction 4.** James decides to run 3 sprints 3 times a week. He runs 60 meters each sprint. How many total meters does he run a week? Rank: 4. Reason: This problem is clear, com plete, and well-contextualized. However, it is conceptually and computationally less challenging than the others, as it only requires basic multiplication.

**Instruction 5.** What is $\sqrt{2}$ times $\sqrt{3}$? Rank: 5. Reason: This problem is clear and complete, but it lacks a practical context. It is conceptually and computationa lly less challenging than the others, as it only requires knowledge of the multiplication of square roots.

**Instruction 6.** What is $\sqrt{-2}$ times $\sqrt{3}$? Rank: 6. Reason: This problem is clear and complete, but it lacks a practical context. It is conceptually challenging as it involves the multiplication of a real number and an imaginary number. However, the computational complexity is low. |

Table 10: A case study GPT-4 scoring the evolved instructions from two aspects:i) Difficulty, and ii) Definition.

## A.3 PRM PROMPT

---

**Example 6: PRM Step Level Labeling Prompt**

You are a senior mathematics grading teacher in university, very skilled in high difficulty fields such as Intermediate Algebra, Precalculus, Prealgebra, Number Theory, Geometry, Counting & Probability, Algebra and so on. Below is a mathematical problem and its corresponding solution, as well as a JSON list format for the solution, where each element is a dictionary with two keys:

- idx: represents the number of each step.
- value: represents each step in the problem-solving process.

Firstly please provide your judgement whether the solution is correct. Your judgment (which must be only True or False) needs to be placed in the <Judge> </Judge>section.

And then you need to judge whether each step is correct and give a score for each solving step in the JSON list which needs to be placed in the <Scores> </Scores>section.

There are three kinds of scores below:

- 1: indicates that the step is correct.
- 0: indicates that the step is ambiguity, meaningless, or subtly misleading, or not helpful to the entire problem-solving process.
- -1: indicates that the step is incorrect.

If this step leads to a final wrong answer, then rate -1. If not, rate 1 or 0.

Here are some rules about whether the solution's each step is correct:

- ## Problem-Solving Thoughts ##: You should first think about how to solve this problem, and then judge whether this step is correct.
- ## Calculation Accuracy ##: You should carefully check and verify whether each step is calculated correctly, including various mathematical numerical calculations. Notably you don't need to consider simplification.
- ## Logical Coherence ##: You should judge whether each step is logically coherent and reasonable.
- ## Basic Theories and Principles ##: You should judge whether each step correctly is using basic mathematical theories, principles, or formulas.

You need to constantly verify and check repeatedly whether each step is correct. And rate each step carefully, honestly, and without bias, order, or discrimination.

Your output is a JSON list format, where each element is a dictionary with three keys:

- idx: represents the number of each step.
- score: represents your rating for this step, which can only be -1, 0 and 1.
- reason: provide your explanation in detail for your rating whether each step of the problem-solving process is correct.

[### Problem ###]: <INSTRUCTION_HERE >

[### Solution JSON List ###]: <Solution JSON LIST >

---

| Question | *John and his two brothers decide to split the cost of an apartment. It is 40% more expensive than John's old apartment which costs $1200 per month. How much does John save per year by splitting the apartment compared to living alone?* |
|---|---|
| GPT-4 Labeling | Step 1: Calculate the cost of John's old apartment. John's old apartment costs $1200 per month. Score: 1.

Step 2: Calculate the cost of the new apartment. Score: 1.

Step 3: The new apartment costs 40% more than John's old apartment, so we need to find 40% of $1200 and add that to the cost of the old apartment. Score: 1.

Step 4: 40% of $1200 is (40/100) * $1200 = $480. So the new apartment costs $1200 + $480 = $1680 per month. Score: 1.

Step 5: Calculate the savings for John by splitting the apartment. Score: 1.

Step 6: If John lived alone, he would pay $1680 per month for the new apartment. Score: 1.

Step 7: However, since he splits the cost with his two brothers, he only needs to pay 1/3 of the total cost. Score: 1.

Step 8: So, John's share of the cost is $1680 / 3 = $560 per month. Score: 1.

Step 9: Calculate the savings per year for John. Score: 1.

Step 10: There are 12 months in a year, so the total savings for John in a year would be 12 * $560 = $6720. Score: -1. Reason: The step is incorrect. The calculation is wrong. The total savings for John in a year should be calculated as the difference between what he would pay living alone and what he pays when splitting the cost, multiplied by 12. The correct calculation should be (1200 - 560) * 12 = $7680.

Step 11: So, John saves $6720 per year by splitting the apartment compared to living alone. Score: -1. Reason: The step is incorrect. The final answer is wrong. The correct answer should be $7680.

Step 12: The answer is: 6720. Score: -1. Reason: The step is incorrect. The final answer is wrong. The correct answer should be $7680. |

Table 11: A case study from Mistral-7B model on GSM8k training set. Red text denotes the incorrect steps that GPT-4 to successfully label errors.

| Question | *Compute* $\cos 120°$. |
|---|---|
| GPT-4 Labeling | Step 1: First, we need to find a right triangle with an angle of $120°$ and then use the Pythagorean theorem to find the length of the hypotenuse. Score: 0.

Step 2: The first step is to visualize a rig ht triangle that has an angle of $120°$. Score: -1. Reason: This step is incorrect. A right triangle cannot have an angle of 120 degrees.

Step 3: We can do this by drawing a line and then rotating it $120°$ around the midpoint of the line. Score: -1. Reason: This step is incorrect. Rotating a line around its midpoint does not create a right triangle with an angle of 120 degrees.

Step 4: This will give us a triangle with two $60°$ angles and one $120°$ angle. Score: -1. Reason: This step is incorrect. A triangle with two angles of 60 degrees and one angle of 120 degrees is not a right triangle.

Step 5: Next, we need to find the length of the hypotenuse of this triangle. Score: 0.

Step 6: We can do this by using the Pythagorean theorem, which stat es that in a right triangle, the square of the length of the hypotenuse is equal to the sum of the squares of the lengths of the other two sides. Score: 0.

Step 7: In this case, the other two sides of the triangle are the two legs of the $60°$ angles, which are each 1 unit long. Score: -1. Reason: This step is incorrect. The sides of a right triangle do not correspond to the legs of the 60 degree angles in this context.

Step 8: So, we have $h^2 = 1 + 1 = 2$. This means that the length of the hypotenuse $h$ is $\sqrt{2}$.', 'Now, we need to find the cosine of $120°$. Score: -1. Reason: This step is incorrect. The calculation is based on incorrect assumptions from previous steps.

Step 9: The cosine of an angle is defined as the ratio of the length of the side opposite the angle to the length of the hypotenuse. Now, we need to find the cosine of $120°$. Score: -1. Reason: This step is incorrect. The cosine of 120 degrees is not 1/sqrt(2).

Step 10: The cosine of an angle is defined as the ratio of the length of the side opposite the angle to the length of the hypotenuse. Score: 1.

Step 11: In this case, the opposite side is the leg of the $60°$ angle, which is 1 unit long. So, we have $\cos 120° = \frac{1}{\sqrt{2}}$. Score: -1. Reason: This step is n ot entirely correct. The cosine of an angle is defined as the ratio of the adjacent side to the hypotenuse, not the opposite side to the hypotenuse.

Step 12: Finally, we need to simplify thi s expression. The square root of 2 is usually written as $\sqrt{2}$, so we can rewrite $\cos 120°$ as $\frac{1}{\sqrt{2}}$. Score: -1. Reason: This step is incorrect. The cosine of 120 degrees is not 1/sqrt(2).

Step 13: So, the final answer is $\boxed{\frac{1}{\sqrt{2}}}$. The answer is: $\frac{1}{\sqrt{2}}$. Score: -1. Reason: This step is incorrect. The final answer is not 1/sqrt(2). The correct answer is -1/2. |

Table 12: A case study from Mistral-7B model on MATH training set. Red text denotes the incorrect steps that GPT-4 to successfully label errors.

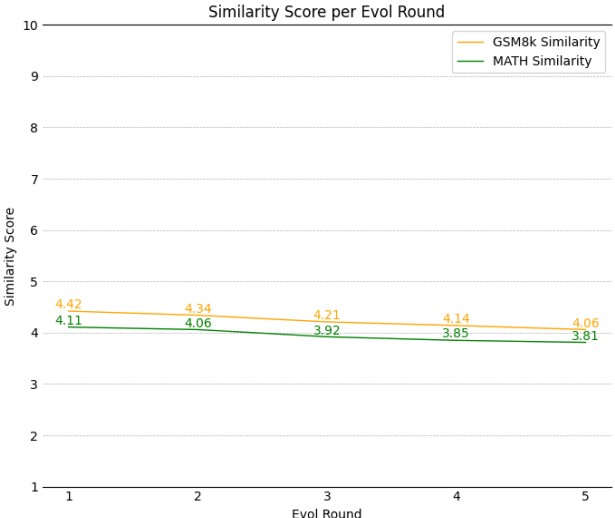

Figure 4: Average similarity scores between GSM8k, MATH samples and the top-1 retrieved data for each round.

## A.4 DATA CONTAMINATION CHECK

Apart from the performance analysis, we also investigate whether evolution leads to the data contamination between training data and test set. To address this consideration, we employ instructions in the GSM8k and MATH test set as queries to retrieve the top-5 samples from all evolved training data with an embedding model, gte-large (Li et al., 2023d). Additionally, we employ GPT-4 to provide similarity judgement between the test sets and the retrieved samples, and remove the top-2 similar instructions. The prompt and details are shown in Appendix A.5. Figure 4 in Appendix illustrates that the evolution process does not yield higher similarity scores. Furthermore, similarity scores across all rounds remain relatively low. These findings indicate that the primary source of performance gain is the introduction of more complex and comprehensive data based on our downward and upward instruction evolution.

## A.5 SIMILARITY CHECKING AND DATA FILTERING

The prompt formats to compute the similarity score between two given math problem tasks are as follow:

> **Example 7: System Prompt for Similarity Checking**
>
> Your task is to evaluate the similarity of the two given math problems. Please review the two math problem tasks carefully, paying close attention to the overlap in variables, conditions, participants, actions, or processes, topics, and contents and core concept and structure. Once you have carefully reviewed both math problem tasks, provide a similarity score between these two math problem tasks. The score should range from 1 to 10 (1: completely different math problem tasks; 10: identical math problem tasks). You only need to provide your score without any explanation.
> **# Problem-1**
> {task1}
>
> **# Problem-2**
> {task2}
>
> Your judgement score:

To thoroughly prevent data leakage from the GSM8k and MATH test datasets to the training dataset, we implemented an additional data filtering step. Utilizing the SOTA embeddings model, gte-large,

we treated all test samples as queries to extract the top 5 samples from the training data. Following this, GPT-4 was employed to evaluate the similarity between the retrieved samples and the test set.

## A.6    DETAILED EXPLANATION OF OUR METHOD FLOW.

We offer a detailed clarification of our method flow , including the significance of colors and shapes as well as the direction of the arrows in Figure 1, to facilitate clearer understanding.

In Figure 1, the various colored squares represent specific elements: **blue squares** denote original instructions, **orange squares** indicate evolved instructions, **cyan squares** signify model-generated solution processes, and **grey squares** correspond to a series of training-related operations such as supervised fine-tuning (SFT), reward modeling, and reinforcement learning (RL). To enhance the mathematical reasoning capabilities of large language models, we propose the **RLEIF method**, which integrates instruction evolution with reinforcement learning. This method consists of three primary steps:

1. **Instruction Evolution and SFT**
   In the first step, we apply upward and downward instruction evolution on the GSM8k and MATH datasets, generating evolved instructions for the SFT. On the leftmost side of Figure 1, the three blue arrows, from top to bottom, represent:

   (a) the adoption of the instruction evolution technique,

   (b) the generation of evolved instruction data, and

   (c) its application to SFT training.

2. **Reward Model Training**
   The second step involves two reward models: the **Instruction Quality Scoring Reward Model (IRM)** and the **Process-Supervised Reward Model (PRM)**, depicted in the central section of Figure 1.

   • **IRM**: We employ upward and downward evolution on a seed instruction, yielding five instructions (original + evolved). These instructions are ranked by quality (e.g., C > A = E > B > D) using GPT-4. Based on the rankings, we train the Instruction Ranking Model (IRM) to assess instruction quality. In Figure 1, this process is shown in the left-central segment: "A" represents the original instruction, while "B," "C," "D," and "E" denote the evolved instructions. The first blue arrow illustrates the ranking process via GPT-4, the second arrow shows the ranking outcomes, and the third arrow highlights the use of this ranked data to train the IRM.

   • **PRM**: In the middle-right section of Figure 1, the process for training the PRM is depicted. The SFT model generates step-by-step solutions from the given instructions, which are then evaluated and labeled by GPT-4. This labeled data is subsequently used to train the PRM.

3. **Reinforcement Learning with PPO**
   In the final step, we integrate the IRM and PRM within a **Proximal Policy Optimization (PPO)**-based reinforcement learning framework. As depicted in the far-right section of Figure 1, the process is as follows:

   (a) The first blue arrow represents instruction scoring by the IRM.

   (b) The second blue arrow shows PPO initialization and the start of reinforcement.

   (c) The third blue arrow illustrates the policy model generating responses based on instructions.

   (d) The fourth blue arrow shows the scoring of each response step using the PRM.

   (e) Arrows five through eight depict the combination of IRM and PRM scores to calculate the final reward score.

   (f) The ninth blue arrow highlights the use of the reward score for the PPO training.

By integrating instruction evolution and reward-based optimization, the RLEIF method significantly enhances the reasoning capabilities of large language models.

### A.7 COMPARE OUR WIZARDMATH-SFT MODEL ACROSS VARIOUS BASE MODELS (0.1B-70B) WITH THE SOTA MODELS ON THE GSM8K AND MATH BENCHMARKS.

To provide a more comprehensive and fair comparison, we have included the WizardMath-SFT results **in Table 13 and Table 14**. These results evaluate the performance of **WizardMath-SFT**, trained exclusively using SFT, against current SOTA models across various base models. The key findings are summarized as follows:

1. **Performance Comparison:**
   - On **Llama-2-7B** and **Mistral-7B-v0.1**, WizardMath-SFT performs marginally below SOTA models (i.e.,Xwin-Math and Skywork-Math) and outperforms existing other excellent models (i.e.,DART-Math).
   - On **Llama-2-13B** and **Llama-2-70B**, WizardMath-SFT achieves comparable performance to Xwin-Math.
   - On all various base models, WizardMath-SFT surpasses most existing SOTA models trained solely with SFT(i.e.,DART-Math).

   Notably, WizardMath-SFT achieves these results using only 418K synthetic data points, a significantly smaller dataset compared to DART-Math (580k-590k), Xwin-Math (1440K) and Skywork-Math (2500K).

2. **Comparison with advanced data synthesis methods (i.e., DART-Math, MetaMath)**
   As shown in the following Table 15, DART-Math demonstrates strong performance across various base models and the data synthesis method proposed by DART-Math shows the effectiveness and outstanding performance. Meanwhile, WizardMath-SFT demonstrates comparable or superior performance to advanced data synthesis methods, such as **DART-Math** and **MetaMath**, across all base models. Key observations include:
   - On **Mistral-7B-v0.1** and **DeepSeekMath**, WizardMath-SFT performs on par with DART-Math (Uniform & Prop2Diff) on GSM8k and surpasses DART-Math (Uniform & Prop2Diff) on MATH;
   - On **Llama3.2 1B**, **Llama3.2 3B**, **Llama3-8B**, and **Llama3.1-8B**, **Llama2-7B**, WizardMath-SFT exhibits a 2%–7% improvement over DART-Math (Uniform & Prop2Diff) on the GSM8k benchmark. On the **MATH** benchmark, WizardMath-SFT outperforms DART-Math (Uniform & Prop2Diff) by approximately 5% – 10%.

   These findings highlight the effectiveness of the proposed **Math Evol-Instruct** for enhancing mathematical reasoning capabilities.

Notably, to compare the advanced data synthesis methods such as DART-Math and MetaMath on different base models, and ensure the same training settings as in our paper during the SFT stage, we employ a learning rate of 2e-5 for the Llama series base models (i.e., Llama2 7B, Llama3.1 8B, Llama3.2 1B, and Llama3.2 3B) and a learning rate of 5e-6 for Mistral-7B-v0.1. All models are trained for 3 epochs with a batch size of 256, and 4 checkpoints are saved per epoch. Finally, we select the checkpoint with the highest accuracy on the GSM8k and MATH benchmarks for reporting.

Table 13: In the study, we compare the WizardMath-SFT/RL model across various base models (0.1B-3B) with the SOTA models on the GSM8k and Math benchmarks. We report the Chain of Thought (CoT) pass@1 results without using any external Python tools. The results from 7B to 70B are shown in Table 14.

| Model | Base | Params | GSM8k | MATH |
|---|---|---|---|---|
| Proprietary models | | | | |
| GPT-o1 (OpenAI, 2023) | - | - | - | 94.8 |
| GPT-o1-mini (OpenAI, 2023) | - | - | - | 90.0 |
| Gemini-1.5 002 (Team et al., 2023) | - | - | - | 86.5 |
| Claude 3.5 Sonnet (Bai et al., 2022) | - | - | 96.4 | 71.1 |
| GPT-4o-2024-0513 (OpenAI, 2023) | - | - | 96.1 | 76.6 |
| GPT-4-turbo-0125 (OpenAI, 2023) | - | - | 94.2 | 64.5 |
| GPT-4-0314 (OpenAI, 2023) | - | - | 94.7 | 52.6 |
| GPT-4 (original version) (OpenAI, 2023) | - | - | 92.0 | 42.5 |
| Baichuan-3 (Yang et al., 2023) | - | - | 88.2 | 49.2 |
| GLM-4 (GLM et al., 2024) | - | - | 87.6 | 47.9 |
| Gemini Pro (Team, 2023) | - | - | 86.5 | 32.6 |
| Claude2 (Bai et al., 2022) | - | - | 85.2 | 32.5 |
| GPT-3.5-Turbo (OpenAI, 2023) | - | - | 81.6 | 43.1 |
| PaLM2 (Anil et al., 2023) | - | - | 80.7 | 34.3 |
| Minerva (Lewkowycz et al., 2022) | - | 540B | 58.8 | 33.6 |
| GPT3.5 (Brown et al., 2020a) | - | - | 57.1 | - |
| Open-Source Models (0.1B-3B) | | | | |
| GPT-2-Small (Brown et al., 2020b) | - | 0.1B | 6.9 | 5.4 |
| GPT-2-Medium (Brown et al., 2020b) | - | 0.3B | 11.2 | 6.2 |
| GPT-2-Large (Brown et al., 2020b) | - | 0.7B | 13.6 | 6.4 |
| GPT-2-XL (Brown et al., 2020b) | - | 1.5B | 15.4 | 6.9 |
| WizardMath-GPT-SFT | GPT-2-Small | 0.1B | 21.2 | 9.1 |
| WizardMath-GPT-RL | GPT-2-Small | 0.1B | 26.4 | 12.3 |
| WizardMath-GPT-SFT | GPT-2-Medium | 0.3B | 30.6 | 11.4 |
| WizardMath-GPT-RL | GPT-2-Medium | 0.3B | 38.7 | 15.6 |
| WizardMath-GPT-SFT | GPT-2-Large | 0.7B | 43.7 | 16.4 |
| WizardMath-GPT-RL | GPT-2-Large | 0.7B | 50.1 | 21.2 |
| WizardMath-GPT-SFT | GPT-2-XL | 1.5B | 51.9 | 18.3 |
| WizardMath-GPT-RL | GPT-2-XL | 1.5B | 58.9 | 25.4 |
| WizardMath-Qwen-SFT | Qwen-Math-2.5 | 1.5B | 82.3 | 62.1 |
| WizardMath-Qwen-RL | Qwen-Math-2.5 | 1.5B | 86.7 | 68.6 |
| Llama-3.2-Instruct (Dubey et al., 2024) | Llama 3.2 | 1B | 44.4 | 30.6 |
| MetaMath (Yu et al., 2023b) | Llama 3.2 | 1B | 51.9 | 15.5 |
| DART-Math-Prop2Diff (Tong et al., 2024) | Llama 3.2 | 1B | 49.2 | 23.4 |
| DART-Math-Uniform (Tong et al., 2024) | Llama 3.2 | 1B | 55.8 | 22.0 |
| WizardMath-Llama-SFT | Llama 3.2 | 1B | 57.1 | 29.7 |
| WizardMath-Llama-RL | Llama 3.2 | 1B | 63.3 | 33.5 |
| Llama-3.2-Instruct (Dubey et al., 2024) | Llama 3.2 | 3B | 77.7 | 48.0 |
| MetaMath (Yu et al., 2023b) | Llama 3.2 | 3B | 72.6 | 25.9 |
| DART-Math-Prop2Diff (Tong et al., 2024) | Llama 3.2 | 3B | 74.0 | 37.8 |
| DART-Math-Uniform (Tong et al., 2024) | Llama 3.2 | 3B | 77.8 | 36.4 |
| WizardMath-Llama-SFT | Llama 3.2 | 3B | 80.3 | 45.2 |
| WizardMath-Llama-RL | Llama 3.2 | 3B | 85.5 | 49.9 |

Table 14: Continue Table 13, in this study, we compare the WizardMath-SFT/RL model across various base models (7B-70B) with the SOTA models on the GSM8k and Math benchmarks. We report the Chain of Thought (CoT) pass@1 results without using any external Python tools.

| Model | Base | Params | GSM8k | MATH |
|---|---|---|---|---|
| Open-Source Models (7B-8B) | | | | |
| Llama-2 (Touvron et al., 2023b) | - | 7B | 14.6 | 2.5 |
| MAmmoTH-CoT (Yue et al., 2023) | Llama-2 | 7B | 50.5 | 10.4 |
| MathScale (Tang et al., 2024) | Llama-2 | 7B | 66.3 | 31.1 |
| MetaMath (Yu et al., 2023b) | Llama-2 | 7B | 66.5 | 19.8 |
| MuggleMath (Li et al., 2023a) | Llama-2 | 7B | 68.4 | - |
| Skywork-Math (Zeng et al., 2024) | Llama-2 | 7B | 72.9 | 47.7 |
| Math-Shepherd (Wang et al., 2024a) | Llama-2 | 7B | 73.2 | 21.6 |
| DART-Math-Prop2Diff (Tong et al., 2024) | Llama-2 | 7B | 69.9 | 30.7 |
| DART-Math-Uniform (Tong et al., 2024) | Llama-2 | 7B | 73.8 | 29.5 |
| Xwin-Math (Li et al., 2024a) | Llama-2 | 7B | 82.6 | 40.6 |
| WizardMath-Llama-SFT | Llama-2 | 7B | 77.4 | 35.6 |
| WizardMath-Llama-RL | Llama-2 | 7B | 84.1 | 43.5 |
| Mistral-v0.1 (Jiang et al., 2023) | - | 7B | 42.9 | 12.9 |
| MathScale (Tang et al., 2024) | Mistral-v0.1 | 7B | 74.8 | 35.2 |
| MMIQC (Liu & Yao, 2024) | Mistral-v0.1 | 7B | 74.8 | 36.0 |
| MetaMath (Yu et al., 2023b) | Mistral-v0.1 | 7B | 77.9 | 28.6 |
| DART-Math-Prop2Diff (Tong et al., 2024) | Mistral-v0.1 | 7B | 81.1 | 45.5 |
| KPMath-Plus (Huang et al., 2024) | Mistral-v0.1 | 7B | 82.1 | 46.8 |
| DART-Math-Uniform (Tong et al., 2024) | Mistral-v0.1 | 7B | 82.6 | 43.5 |
| Skywork-Math (Zeng et al., 2024) | Mistral-v0.1 | 7B | 83.9 | 51.2 |
| Math-Shepherd (Wang et al., 2024a) | Mistral-v0.1 | 7B | 84.1 | 33.0 |
| MAmmoTH2-Plus (Yue et al., 2024) | Mistral-v0.1 | 7B | 84.7 | 45.0 |
| JiuZhang3.0 (Zhou et al., 2024) | Mistral-v0.1 | 7B | 88.6 | 52.8 |
| Xwin-Math (Li et al., 2024a) | Mistral-v0.1 | 7B | 89.2 | 43.7 |
| WizardMath-Mistral-SFT | Mistral-v0.1 | 7B | 82.8 | 48.1 |
| WizardMath-Mistral-RL | Mistral-v0.1 | 7B | 90.7 | 55.4 |
| WizardMath-Mistral-SFT | Mistral-v0.3 | 7B | 84.5 | 49.9 |
| WizardMath-Mistral-RL | Mistral-v0.3 | 7B | 90.4 | 55.6 |
| WizardMath-Mathstral-SFT | Mathstral-v0.1 | 7B | 88.3 | 64.2 |
| WizardMath-Mathstral-RL | Mathstral-v0.1 | 7B | 93.8 | 70.9 |
| Qwen2.5-Math-Base (Yang et al., 2024) | Qwen2.5-Math | 7B | 91.6 | 55.4 |
| WizardMath-Qwen-SFT | Qwen2.5-Math | 7B | 92.3 | 72.3 |
| WizardMath-Qwen-RL | Qwen2.5-Math | 7B | 93.9 | 77.8 |
| WizardMath-Qwen-SFT | Qwen2.5 | 7B | 89.8 | 68.1 |
| WizardMath-Qwen-RL | Qwen2.5 | 7B | 94.0 | 74.5 |
| DeepSeekMath-Base (Shao et al., 2024) | - | 7B | 64.2 | 36.2 |
| NuminaMath-CoT (Li et al., 2024b) | DeepseekMath | 7B | 75.4 | 55.2 |
| MMIQC (Liu & Yao, 2024) | DeepSeekMath | 7B | 79.0 | 45.3 |
| KPMath-Plus (Huang et al., 2024) | DeepSeekMath | 7B | 83.9 | 48.8 |
| DART-Math-Prop2Diff (Tong et al., 2024) | DeepSeekMath | 7B | 86.8 | 53.6 |
| DeepSeekMath-RL (Shao et al., 2024) | DeepSeekMath | 7B | 88.2 | 51.7 |
| DART-Math-Uniform (Tong et al., 2024) | DeepSeekMath | 7B | 88.2 | 52.9 |
| WizardMath-DeepSeek-SFT | DeepSeekMath | 7B | 88.9 | 58.2 |
| WizardMath-DeepSeek-RL | DeepSeekMath | 7B | 91.0 | 64.6 |
| MetaMath (Yu et al., 2023b) | Llama 3 | 8B | 77.3 | 20.6 |
| MMIQC (Liu & Yao, 2024) | Llama 3 | 8B | 77.6 | 29.5 |
| DART-Math-Prop2Diff (Tong et al., 2024) | Llama 3 | 8B | 81.1 | 46.6 |
| DART-Math-Uniform (Tong et al., 2024) | Llama 3 | 8B | 82.5 | 45.3 |
| MAmmoTH2-Plus (Yue et al., 2024) | Llama 3 | 8B | 84.1 | 42.8 |
| Llama 3.1-Instruct (Dubey et al., 2024) | Llama 3 | 8B | 84.5 | 51.9 |
| JiuZhang3.0 (Zhou et al., 2024) | Llama 3 | 8B | 88.6 | 51.0 |
| WizardMath-Llama-SFT | Llama 3 | 8B | 88.9 | 53.3 |
| WizardMath-Llama-RL | Llama 3 | 8B | 90.3 | 58.8 |
| MetaMath (Yu et al., 2023b) | Llama 3.1 | 8B | 80.4 | 35.4 |
| DART-Math-Prop2Diff (Tong et al., 2024) | Llama 3.1 | 8B | 84.3 | 46.5 |
| DART-Math-Uniform (Tong et al., 2024) | Llama 3.1 | 8B | 86.7 | 45.1 |
| WizardMath-Llama-SFT | Llama 3.1 | 8B | 89.2 | 55.8 |
| WizardMath-Llama-RL | Llama 3.1 | 8B | 93.4 | 62.3 |
| Open-Source Models (13B) | | | | |
| Llama-2 (Touvron et al., 2023b) | - | 13B | 28.7 | 3.9 |
| MAmmoTH-CoT (Yue et al., 2023) | Llama 2 | 13B | 56.3 | 12.9 |
| MathScale (Tang et al., 2024) | Llama 2 | 13B | 71.3 | 33.8 |
| MetaMath (Yu et al., 2023b) | Llama 2 | 13B | 72.3 | 22.4 |
| MuggleMath (Li et al., 2023a) | Llama 2 | 13B | 74.0 | - |
| KPMath-Plus (Huang et al., 2024) | Llama 2 | 13B | 81.6 | 41.0 |
| Xwin-Math (Li et al., 2024a) | Llama 2 | 13B | 88.1 | 44.9 |
| WizardMath-Llama-SFT | Llama 2 | 13B | 86.8 | 46.5 |
| WizardMath-Llama-RL | Llama 2 | 13B | 89.7 | 50.6 |
| Open-Source Models (70B) | | | | |
| Llama-2 (Touvron et al., 2023b) | - | 70B | 56.8 | 13.5 |
| MAmmoTH-CoT (Yue et al., 2023) | Llama-2 | 70B | 72.4 | 21.1 |
| MetaMath (Yu et al., 2023b) | Llama-2 | 70B | 82.3 | 26.6 |
| KPMath-Plus (Huang et al., 2024) | Llama-2 | 70B | 87.4 | 48.6 |
| Xwin-Math (Li et al., 2024a) | Llama-2 | 70B | 90.6 | 52.8 |
| WizardMath-Llama-SFT | Llama-2 | 70B | 89.5 | 54.4 |
| WizardMath-Llama-RL | Llama-2 | 70B | 92.8 | 58.6 |

Table 15: In this study, we mainly compare the performance of WizardMath-SFT with advanced data synthesis methods such as DART-Math and MetaMath on different base models under the GSM8k and MATH benchmarks in the SFT stage. We report the CoT pass@1 results of the model without relying on any external Python tools.

| Model | Base | Params | GSM8k | MATH |
|---|---|---|---|---|
| DART-Math-Prop2Diff | Llama 3.2 | 1B | 49.2 | 23.4 |
| MetaMath | Llama 3.2 | 1B | 51.9 | 15.5 |
| DART-Math-Uniform | Llama 3.2 | 1B | 55.8 | 22.0 |
| WizardMath-Llama-SFT | Llama 3.2 | 1B | 57.1 | 29.7 |
| MetaMath | Llama 3.2 | 3B | 72.6 | 25.9 |
| DART-Math-Prop2Diff | Llama 3.2 | 3B | 74.0 | 37.8 |
| DART-Math-Uniform | Llama 3.2 | 3B | 77.8 | 36.4 |
| WizardMath-Llama-SFT | Llama 3.2 | 3B | 80.3 | 45.2 |
| MetaMath | Llama-2 | 7B | 66.5 | 19.8 |
| DART-Math-Prop2Diff | Llama-2 | 7B | 69.9 | 30.7 |
| DART-Math-Uniform | Llama-2 | 7B | 73.8 | 29.5 |
| WizardMath-Llama-SFT | Llama-2 | 7B | 77.4 | 35.6 |
| MetaMath | Mistral-v0.1 | 7B | 77.9 | 28.6 |
| DART-Math-Prop2Diff | Mistral-v0.1 | 7B | 81.1 | 45.5 |
| DART-Math-Uniform | Mistral-v0.1 | 7B | 82.6 | 43.5 |
| WizardMath-Mistral-SFT | Mistral-v0.1 | 7B | 82.8 | 48.1 |
| DART-Math-Prop2Diff | DeepSeekMath | 7B | 86.8 | 53.6 |
| DART-Math-Uniform | DeepSeekMath | 7B | 88.2 | 52.9 |
| WizardMath-DeepSeek-SFT | DeepSeekMath | 7B | 88.9 | 58.2 |
| MetaMath | Llama 3 | 8B | 77.3 | 20.6 |
| DART-Math-Prop2Diff | Llama 3 | 8B | 81.1 | 46.6 |
| DART-Math-Uniform | Llama 3 | 8B | 82.5 | 45.3 |
| WizardMath-Llama-SFT | Llama 3 | 8B | 88.9 | 53.3 |
| MetaMath | Llama 3.1 | 8B | 80.4 | 35.4 |
| DART-Math-Prop2Diff | Llama 3.1 | 8B | 84.3 | 46.5 |
| DART-Math-Uniform | Llama 3.1 | 8B | 86.7 | 45.1 |
| WizardMath-Llama-SFT | Llama 3.1 | 8B | 89.2 | 55.8 |

Table 16: The performance of WizardMath on the GSM8k and MATH based on the Mathstral-7B-v0.1-Base, Qwen2.5-7B-Base, Qwen2.5-Math-1.5B-Base, and Qwen2.5-Math-7B-Base

| Models | Base | Params | GSM8k | MATH |
|---|---|---|---|---|
| Mathstral-v0.1-Base | - | 7B | 77.1 | 56.6 |
| WizardMath-Mathstral | Mathstral-v0.1-Base | 7B | 93.8 | 70.9 |
| Qwen2.5-Math-Base | - | 1.5B | 76.8 | 49.8 |
| WizardMath-Qwen2.5-Math | Qwen2.5-Math-Base | 1.5B | 86.7 | 68.6 |
| Qwen2.5-Math-Base | - | 7B | 91.6 | 55.4 |
| WizardMath-Qwen2.5-Math | Qwen2.5-Math-Base | 7B | 93.9 | 77.8 |
| Qwen2.5-Base | - | 7B | 85.4 | 49.8 |
| WizardMath-Qwen2.5 | Qwen2.5-Base | 7B | 94.0 | 74.5 |

Table 17: The impact of applying the proposed Instruction Quality Scoring Reward Model (IRM) and Process Supervised Reward Model (PRM) to PPO training across various SFT backbones (i.e., DART-Math, MetaMath, and Xwin-Math)

| Model | Base | Params | GSM8k | MATH |
|---|---|---|---|---|
| MetaMath-SFT | Llama-2 | 7B | 66.5 | 19.8 |
| MetaMath-RL | Llama-2 | 7B | 75.6 | 25.1 |
| DART-Math-Prop2Diff-SFT | Llama-2 | 7B | 69.9 | 30.7 |
| DART-Math-Prop2Diff-RL | Llama-2 | 7B | 76.8 | 37.1 |
| DART-Math-Uniform-SFT | Llama-2 | 7B | 73.8 | 29.5 |
| DART-Math-Uniform-RL | Llama-2 | 7B | 79.1 | 35.2 |
| Xwin-Math-SFT | Llama-2 | 7B | 82.6 | 40.6 |
| Xwin-Math-RL | Llama-2 | 7B | 88.2 | 48.5 |
| WizardMath-Llama-SFT | Llama-2 | 7B | 77.4 | 35.6 |
| WizardMath-Llama-RL | Llama-2 | 7B | 84.1 | 43.5 |
| MetaMath-SFT | Mistral-v0.1 | 7B | 77.9 | 28.6 |
| MetaMath-RL | Mistral-v0.1 | 7B | 86.4 | 35.2 |
| DART-Math-Prop2Diff-SFT | Mistral-v0.1 | 7B | 81.1 | 45.5 |
| DART-Math-Prop2Diff-RL | Mistral-v0.1 | 7B | 87.5 | 51.4 |
| DART-Math-Uniform-SFT | Mistral-v0.1 | 7B | 82.6 | 43.5 |
| DART-Math-Uniform-RL | Mistral-v0.1 | 7B | 88.1 | 48.7 |
| WizardMath-Mistral-SFT | Mistral-v0.1 | 7B | 82.8 | 48.1 |
| WizardMath-Mistral-RL | Mistral-v0.1 | 7B | 90.7 | 55.4 |

## A.8    THE PERFORMANCE OF WIZARDMATH ON THE OTHER DIFFERENT BASE MODELS

Table 16 supplements the performance improvements of Mathstral-7B-v0.1-Base, Qwen2.5-7B-Base, Qwen2.5-Math-1.5B-Base, and Qwen2.5-Math-7B-Base on the GSM8k and MATH datasets.

The results demonstrate that using Mathstral-7B-v0.1-Base as the base model, WizardMath-Mathstral improves performance by 16.7% on GSM8k (93.8 vs. 77.1) and 14.5% on MATH (70.9 vs. 56.6). When employing Qwen2.5-Math-1.5B-Base as the base model, WizardMath-Qwen2.5-Math-1.5B achieves 9.9% improvement on GSM8k (86.7 vs. 76.8) and 18.8% on MATH (68.6 vs. 49.8). Similarly, with Qwen2.5-Math-7B-Base, WizardMath-Qwen2.5-Math-7B shows a 2.3% increase on GSM8k (93.9 vs. 91.6) and 22.4% on MATH (77.8 vs. 55.4). Finally, using Qwen2.5-7B-Base as the base model, WizardMath-Qwen2.5-7B improves by 8.6% on GSM8k (94.0 vs. 85.4) and 24.7% on MATH (74.5 vs. 49.8).

Notably, both Mathstral-7B-v0.1-Base and Qwen2.5-Math-Base, pre-trained on extensive mathematical corpora, exhibit robust mathematical reasoning capabilities and deliver strong performance on GSM8k and MATH datasets. However, our proposed RLEIF method achieves substantial performance enhancements even with these highly math-optimized models. Specifically, on the MATH, RLEIF delivers a performance boost of 15% 25%, while on GSM8k, the improvement ranges from 8% 16% (with the exception of Qwen2.5-Math-7B-Base, which achieves a high baseline of 91.6 on GSM8k but still benefits from a 2.3% enhancement). These results underscore the continuous improvement enabled by our RLEIF method on models pre-trained with specialized mathematical corpora, further validating its effectiveness and scalability.

## A.9    THE IMPACT OF APPLYING THE PROPOSED INSTRUCTION QUALITY SCORING REWARD MODEL (IRM) AND PROCESS SUPERVISED REWARD MODEL (PRM) TO PPO TRAINING ACROSS VARIOUS SFT BACKBONES

Table 17 shows the impact of applying the proposed Instruction Quality Scoring Reward Model (IRM) and Process Supervised Reward Model (PRM) to PPO training across various SFT backbones (i.e., DART-Math, MetaMath, and Xwin-Math). The results demonstrate that incorporating our IRM and PRM during PPO training led to a performance improvement of 5% to 8% on both GSM8k and MATH for most SFT models. For instance:

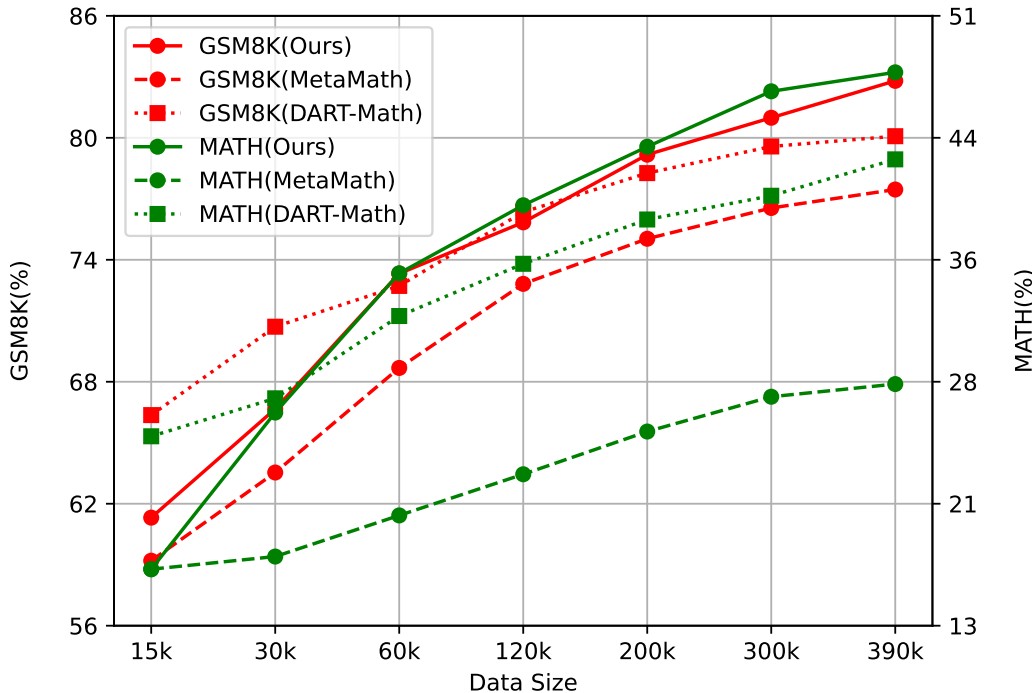

Figure 5: The performance of WizardMath Evol-instruct in comparison with DART-Math and MetaMath across different training data scales on the GSM8k and MATH benchmarks in the SFT stage. We use the Mistral-7B as base model

- **When using DART-Math as the SFT backbone based on Llama2-7B:**
  On GSM8k, after reinforcement learning training with IRM and PRM, Prop2Diff-RL improved by 6.9% (69.9% vs. 76.8%), and Uniform-RL improved by 5.3% (73.8% vs. 79.1%).
  On MATH, Prop2Diff-RL achieved a 6.4% gain (30.7% vs. 37.1%), and Uniform-RL improved by 5.7% (29.5% vs. 35.2%).

- **When using DART-Math as the SFT backbone based on Mistral-7B-v0.1:**
  On GSM8k, Prop2Diff-RL improved by 6.4% (81.1% vs. 87.5%), and Uniform-RL increased by 5.5% (82.6% vs. 88.1%).
  On MATH, Prop2Diff-RL rose by 5.9% (45.5% vs. 51.4%), and Uniform-RL saw a 5.2% enhancement (43.5% vs. 48.9%).

- **For the MetaMath models based on Llama2-7B and Mistral-7B-v0.1:**
  Training with PPO using IRM and PRM led to performance improvements of 8% to 9% on GSM8k and 5% to 8% on MATH.

- **Similarly, for the Xwin-Math-Llama2-7B model, performance on both GSM8k and MATH improved by 6% to 8%.**

These findings highlight the significant contributions of our IRM and PRM during reinforcement learning, consistently enhancing mathematical reasoning abilities of our SFT models while achieving robust generalization on different SFT backbones. This represents a key contribution of our study. Thus, our study primarily makes two core contributions:

1. The proposed Math Evol Instruct data synthesis method is also as effective and practical as the current state-of-the-art data synthesis methods, such as DART-Math, Skywork-Math and Xwin-Math in the SFT stage. It also significantly enhances the mathematical reasoning capabilities of our models.

2. The proposed IRM and PRM models substantially improve performance during the reinforcement learning phase. They not only continuously enhance the mathematical reasoning abilities of our

Table 18: The performance comparison of WizardMath-SFT with DART-Math, Xwin-Math, and Skywork-Math on the Llama2-7B base model on the MATH benchmark.

| Llama2 7B as the base model | Data size | MATH |
|---|---|---|
| DART-Math-Uniform | 591k | 29.5 |
| DART-Math-Prop2Diff | 585k | 30.7 |
| Xwin-Math | 1440k | 40.6 |
| Skywork-Math | 360k | 29.36 |
| Skywork-Math | 720k | 34.54 |
| Skywork-Math | 2500k | 47.7 |
| WizardMath-SFT | 418k | 35.6 |

SFT models but also achieve strong generalization across various SFT backbones (i.e., DART-Math). Outstanding performance is demonstrated on the GSM8k and MATH.

## A.10 COMPARE OUR APPROACH WITH THE ADVANCED METHOD FOR SFT USING SYNTHESIZED DATA, SUCH AS DARTMATH

As the volume of training data increases, WizardMath-Evol-Instruct consistently improves its performance on the GSM8k and MATH benchmarks, exhibiting a slightly higher growth rate than DART-Math in Figure 5. In the initial stages, WizardMath slightly underperforms compared to DART-Math. This advantage may stem from DART-Math being distilled from DeepSeekMath-RL, an advanced mathematical reasoning model pre-trained on 120B high-quality mathematical tokens, showcasing exceptional proficiency in mathematical reasoning. However, once the dataset exceeds 60k, its performance begins to surpasse DART-Math. At a data scale of 390k, WizardMath slightly outperforms DART-Math by 2%–3% on GSM8k and by 5%–6% on MATH. Additionally, WizardMath-Evol-Instruct consistently exceeds MetaMath at the same data scales, achieving increases of 3%–6% on GSM8k and 15%–20% on MATH. This performance gain is attributed to the efficiency of Math Evol-Instruct's upward and downward evolution processes. These findings demonstrate that our Math Evol-Instruct method is also as scalable and effective as DART-Math for the large-scale synthetic data.

## A.11 OUR MATH EVOL-INSTRUCT COMPARED TO OTHER SFT METHODS, SUCH AS DART-MATH, XWINMATH AND SKYWORK-MATH

In Table 18, we show the performance comparison of WizardMath-SFT with DART-Math, Xwin-Math and Skywork-Math on the Llama2-7B base model on the MATH benchmark.

- **WizardMath-SFT vs. DART-Math:**
  WizardMath-SFT, based on the Llama2-7B model, outperforms DART-Math-Uniform by 6.1% and DART-Math-Prop2Diff by 4.9% on the MATH. Notably, the amount of data used by WizardMath-SFT is only 70%–71% of DART-Math (418k vs. 591k; 418k vs. 585k).

- **WizardMath vs. Xwin-Math:**
  Although WizardMath-SFT is 5% lower than Xwin-Math on the MATH, the amount of data used is only 29.0% of Xwin-Math (418k vs. 1440k), which is much less than Xwin-Math. Moreover, Xwin-Math leverages GPT-4-turbo for data synthesis. However, WizardMath-SFT outperforms Xwin-Math on the MATH when using different backbones such as Mistral-7B-v0.1, Llama2-13B, and Llama2-70B as shown in Table 14. For instance, in Table 14, WizardMath-SFT exceeds Xwin-Math by 4.4% (48.1% vs. 43.7%) when using the Mistral-7B-v0.1 as the base model.

- **WizardMath vs. Skywork-Math:**
  WizardMath-SFT underperforms Skywork-Math-2500k on the MATH benchmark by 12.1%, but it uses only 16.7% of the amount of data used by Skywork-Math-2500k (418k vs. 2500k), which is much less than Skywork-Math. Furthermore, according to **Figure 5 About Synthetic Data Size in the Skywork-Math paper(Zeng et al., 2024)**, Skywork-Math-720k scores 34.54% on MATH, and Skywork-Math-360k scores 29.36%. Therefore, WizardMath-SFT-418k performs comparably to Skywork-Math-720k on MATH, and with the same amount of data, WizardMath-SFT outperforms Skywork-Math.

Table 19: The impact of using advanced open-source models(i.e., Llama-3.1-405B-Instruct) for PRM training data labeling and we use Mistral-7B-v0.1 as the base model.

| Models | AI-Label | GSM8k | MATH |
|---|---|---|---|
| WizardMath-SFT | - | 82.8 | 48.1 |
| + PRM-Llama-3.1-405B-Instruct | Llama-3.1-405B-Instruct | 85.8 | 51.5 |
| + PRM-GPT-4 | GPT-4 | 87.2 | 52.7 |

In summary, the Math Evol Instruct data synthesis method proposed in our study is as effective and practical as the current state-of-the-art data synthesis methods, such as DART-Math, Skywork-Math and Xwin-Math in the SFT stage. It significantly enhances the mathematical reasoning capabilities of the model, marking a key contribution of our work.

## A.12 FEASIBILITY OF USING ADVANCED OPEN-SOURCE MODELS INSTEAD OF GPT-4 TO LABEL PRM TRAINING DATA

We realize that there is a high cost of directly distilling GPT-4 in large-scale data scenarios, which is a limitation of this study. Additionally, manual annotation demands mathematical expertise and entails a challenging, time-intensive, and costly process. Moreover, our evolved instructions lack correct answers, limiting compatibility with the methods employed by Math-Shepherd(Wang et al., 2024a) which needs the correct answers.

To mitigate these challenges, we also explore the feasibility of leveraging advanced open-source models, such as Llama-3.1-405B-Instruct, instead of GPT-4 for PRM training data labeling, using the same label prompts and training settings. As shown in the Table 19, WizardMath-PRM-Llama-3.1-405B achieves **85.8%** on the GSM8k, outperforming WizardMath-SFT by **3.0%** and lagging behind WizardMath-PRM-GPT-4 by **1.4%**. On the MATH, it scores **51.5%**, exceeding WizardMath-SFT by **3.4%** with a **1.2%** gap compared to WizardMath-PRM-GPT-4. Balancing cost and accuracy, Llama-3.1-405B-Instruct demonstrates considerable potential as a substitute for GPT-4 in PRM training data labeling.

## A.13 HIGHLIGHT THE CORE CONTRIBUTIONS OF OUR METHOD.

We highlight the key contributions of our method as follows:

**1. Unlike WizardLM/WizardCoder, which primarily focus on increasing instruction difficulty, we are the first to propose the novel concept of downward evolution, a major distinction in instruction evolution.**

In Table 6, we provide a detailed analysis of the effects of downward evolution. Specifically, two rounds of downward evolution led to a remarkable improvement in GSM8k performance by 14.8% (74.5 vs. 59.7) and in MATH performance by 19.6% (34.7 vs. 15.1) compared to the original, significantly enhancing the model's mathematical reasoning capabilities. This demonstrates that Math Evol-Instruct is instrumental in significantly boosting the model's mathematical reasoning ability.

**2. In reinforcement learning (RL) training, we firstly propose the instruction quality scoring reward model (IRM) combined with the process supervision reward model (PRM) further enhancing WizardMath mathematical reasoning ability.** As demonstrated in Table 3, our method achieves a remarkable 5%–8% improvement in GSM8k and MATH performance over the SFT backbone across models of various sizes, leveraging PRM and IRM for the PPO training.

**3. We firstly propose to use AI to annotate the step-level PRM training data.** Additionally, the training datasets for SFT, PRM, and IRM are fully synthesized using AI systems. This fully AI-automated data generation pipeline ensures scalability.

**4. WizardMath demonstrates outstanding performance across a wide range of model scales, from 100M to 1B and 70B parameters, on the benchmarks such as GSM8k, MATH, and out-of-distribution (OOD) tasks like MWPBench(Tang et al., 2024).** It surpasses all existing open-source state-of-the-art models, showcasing the effectiveness and robustness of the RLEIF approach proposed in our study.

Table 20: The performance of WizardMath-SFT on the 7 out-of-domain evaluation results covering K-12, college, and competition level math problems compared with some SOTA models (i.e., DART-Math) in the SFT stage. The results of models in the table refer to MWPBENCH (Tang et al., 2024). "AGIE" stands for AGIEval. We report the models' CoT pass@1 results on MwpBench without using any external python tool

| Models | College Math | TAL | Math23k | Ape210k | Gaokao Bench Math | AGIE Gaokao Math | AGIE SAT Math | AVG |
|---|---|---|---|---|---|---|---|---|
| *Proprietary models* | | | | | | | | |
| GPT-4 | **24.4** | **51.8** | **76.5** | **61.5** | **35.4** | **28.2** | **68.6** | **49.5** |
| GPT-3.5-Turbo | 21.6 | 42.9 | 62.5 | 44.0 | 23.2 | 15.3 | 55.8 | 37.9 |
| *Models based on LLaMA-2 13B* | | | | | | | | |
| LLaMA-2 13B | 1.2 | 6.3 | 9.5 | 7.9 | 0.7 | 0.4 | 6.8 | 4.7 |
| MAmmoTH-CoT | 6.5 | 17.3 | 39.5 | 28.1 | 5.9 | 4.9 | 20.5 | 17.5 |
| GAIR-Abel | 7.9 | 21.1 | 42.2 | 27.8 | 7.0 | 4.9 | 30.3 | 20.2 |
| MetaMath | 10.1 | 25.4 | 48.6 | 31.6 | 9.6 | 5.6 | 38.2 | 24.2 |
| MathScale 13B | 20.4 | 38.1 | 61.1 | 43.7 | 20.0 | 12.3 | 55.8 | 35.9 |
| WizardMath-SFT | 22.2 | 42.5 | 65.9 | 47.6 | 31.6 | 23.5 | 59.7 | 41.9 |
| WizardMath-RL | **22.9** | **43.3** | **70.3** | **50.8** | **33.1** | **25.7** | **64.7** | **44.4** |
| *Models based on LLaMA-2 7B* | | | | | | | | |
| LLaMA-2 7B | 2.3 | 7.6 | 6.8 | 7.3 | 2.1 | 2.9 | 2.9 | 4.6 |
| MAmmoTH-CoT | 6.2 | 13.3 | 34.6 | 21.4 | 3.9 | 2.7 | 19.6 | 14.5 |
| GAIR-Abel | 6.6 | 18.3 | 35.4 | 24.5 | 4.3 | 4.4 | 23.5 | 16.7 |
| MetaMath | 9.4 | 22.5 | 44.0 | 29.9 | 5.9 | 5.1 | 36.2 | 21.9 |
| DART-Math-Uniform | 12 | 27.3 | 47.9 | 32.9 | 14.8 | 11.1 | 45.1 | 27.3 |
| DART-Math-Prop2Diff | 11.9 | 27.7 | 49.9 | 34.3 | 12.8 | 10.6 | 47.1 | 27.8 |
| Xwin-Math-V1.1 | 14.9 | 29.7 | 59.6 | 40.8 | 15.9 | 8.4 | 51.0 | 31.5 |
| MathScale 7B | 20.9 | 35.2 | 59.0 | 41.8 | 19.6 | 12.6 | 57.8 | 35.3 |
| WizardMath-SFT | 21.1 | 38.5 | 62.4 | 43.8 | 26.3 | 17.7 | 58.3 | 38.3 |
| WizardMath-RL | **21.2** | **40.2** | **67.3** | **46.1** | **28.9** | **18.7** | **62.7** | **40.7** |
| *Models based on Mistral 7B* | | | | | | | | |
| Mistral 7B | 7.5 | 17.9 | 18.5 | 15.5 | 6.2 | 5.9 | 22.5 | 13.4 |
| MetaMath Mistral | 15.7 | 31.4 | 55.1 | 38.1 | 15.3 | 10.1 | 50.9 | 30.9 |
| DART-Math-Uniform | 19.4 | 34.8 | 61.6 | 44.8 | 27.0 | 16.1 | 59.8 | 37.6 |
| MathScale Mistral | 21.8 | 39.9 | 64.4 | 46.0 | 21.4 | 14.3 | 57.8 | 37.9 |
| DART-Math-Prop2Diff | 19.9 | 37.4 | 62.2 | 44.9 | 27.2 | 18.1 | 62.7 | 38.9 |
| WizardMath-Mistral-SFT | 24.3 | 42.7 | 66.6 | 49.7 | 35.2 | 22.7 | 63.1 | 43.5 |
| WizardMath-Mistral-RL | **24.8** | **44.8** | **71.2** | **52.6** | **37.2** | **24.5** | **64.7** | **45.7** |

## A.14 THE PERFORMANCE OF WIZARDMATH-SFT ON THE OUT-OF-DOMAIN BENCHMARKS COMPARED WITH SOME SOTA MODELS (I.E., DART-MATH) IN THE SFT STAGE.

Table 20 presents the performance of WizardMath-SFT on 7 out-of-domain (OOD) evaluation tasks covering K-12, college, and competition-level math problems in the SFT stage. The results indicate that **WizardMath-SFT** consistently surpasses open-source state-of-the-art models (i.e., DART-Math, Xwin-Math, and MathScale) across various scales and tasks, achieving an average improvement of **3%-6%**. For instance:

- With the Llama2-7B base model, WizardMath-SFT outperformed DART-Math-Uniform by **11.0%** (38.3% vs. 27.3%) and DART-Math-Prop2Diff by **10.5%** (38.3% vs. 27.8%) on average.

- With the Mistral-7B base model, WizardMath-SFT achieved an average improvement of **5.9%** over DART-Math-Uniform (43.5% vs. 37.6%) and **4.6%** over DART-Math-Prop2Diff (43.5% vs. 38.9%).

These findings highlight the effectiveness of our **Math Evol-Instruct** method, demonstrating its robustness and superior generalization capabilities on out-of-domain tasks.

