# OpenReview forum: "WizardMath: Empowering Mathematical Reasoning for Large Language Models via Reinforced Evol-Instruct"
_ICLR.cc/2025/Conference — ICLR 2025 Oral_

### Official Review · Reviewer_Hirr · 2024-10-31

**Soundness:** 3
**Presentation:** 3
**Contribution:** 4
**Rating:** 8
**Confidence:** 4

**Summary:**

This paper introduces WizardMath, an innovative approach to improving mathematical reasoning capabilities in large language models (LLMs). WizardMath employs a unique reinforcement learning method called Reinforcement Learning from Evol-Instruct Feedback (RLEIF), which optimizes the model’s performance on mathematical reasoning without external tools. It specifically targets pretrained model to enhance their reasoning by leveraging evolutionary instruction (Evol-Instruct) to generate diverse math problems and using process-based supervision to ensure correctness at each reasoning step. Experiments demonstrate that WizardMath achieves higher accuracy than other leading models, even outperforming proprietary models like GPT-3.5 and Claude on GSM8k and MATH benchmarks.

**Strengths:**

1. This paper provides a scalable approach to enhance LLM reasoning capabilities in math without external computational tools.
2. Notable improvements in metrics across diverse benchmarks.
3. The introduction of dual reward models (IRM and PRM) effectively improves model reliability and accuracy.
4. The synthetic data and process supervision paradigm have had a great impact on the community.

**Weaknesses:**

1. The training data for RLEIF is derived partly from a designated training set and synthetic from existing models, such as GPT-4. How might the reliance on specific LLMs for generating synthetic data (like GPT-4) affect the scalability of this approach for researchers without access to such models? Have the authors explored alternative methods for generating diverse mathematical problems that don't depend on proprietary models?
2. While RLEIF has undergone extensive evaluation in the context of mathematical reasoning, it may also hold potential for applications in other reasoning domains. Have the authors considered applying RLEIF to other reasoning tasks like logical deduction or causal inference? It would be valuable to see if the benefits observed in mathematical reasoning transfer to these domains, and what modifications might be necessary.

**Questions:**

see above

**Details Of Ethics Concerns:**

I have no concern about ethics.

---

> ### Author Response · Authors · 2024-11-28
>
> Dear Reviewer Hirr,
>
> We would like to express our sincere gratitude for your insightful comments and the time you devoted to reviewing our work. Your expert feedback has been indispensable in guiding us toward refining our paper, enhancing its comprehensiveness, and strengthening its competitiveness. We deeply appreciate your constructive suggestions and continued support. Below, we provide detailed responses to the Weaknesses raised in your review, addressing each point systematically.
>
> Please find below a detailed discussion of the points you have raised:
>
> > **Weaknesses-1**:  The training data for RLEIF is derived partly from a designated training set and synthetic from existing models, such as GPT-4. How might the reliance on specific LLMs for generating synthetic data (like GPT-4) affect the scalability of this approach for researchers without access to such models? Have the authors explored alternative methods for generating diverse mathematical problems that don't depend on proprietary models?
>
> Thank you for your insightful questions and constructive Feedback.
>
> Yes, I have explored alternative methods for generating diverse mathematical problems that don't depend on proprietary models.
>
> In Section 4.4 of our paper (Lines 484–510), we explore the feasibility of leveraging advanced open-source models, such as Llama-3-70B-Instruct, as alternatives to GPT-4 for synthesizing SFT training data using the same instruction evolution strategy. Subsequently, we also use Llama-3.1-405B-Instruct to annotate the IRM and PRM training data. All experiments are conducted under consistent training settings and we use Mistral-7B-v0.1 as the base model.
>
> As shown in the table below, during the SFT stage, WizardMath-SFT-Llama3-Evol achieves 76.7% accuracy on the GSM8k, representing a **33.8%** improvement on Mistral-7B-v0.1 (76.7 vs. 42.9). Similarly, it achieves 43.5% accuracy on the MATH, with a **30.6%** improvement on Mistral-7B-v0.1 (43.5 vs. 12.9). However, compared to WizardMath-SFT-GPT-4-Evol, which uses GPT-4 for instruction evolution, there is a performance gap of approximately 5%–6%.
>
> During the RL stage, using Llama-3.1-405B-Instruct to label IRM and PRM training data, WizardMath-RL-PRM achieves a 3.6% improvement over WizardMath-SFT-Llama3-Evol (80.3% vs. 76.7%), outperforming WizardMath-RL-ORM by 2.2% on the GSM8k. On the MATH, WizardMath-RL-PRM improves by 3.5% over WizardMath-SFT-Llama3-Evol (47.0% vs. 43.5%), exceeding WizardMath-RL-ORM by 2.4%.
>
> When IRM and PRM are combined for PPO training, WizardMath-RL-IRM-PRM achieves 83.2% accuracy on the GSM8k, a 2.9% improvement over WizardMath-RL-PRM and a **6.5%** improvement over WizardMath-SFT-Llama3-Evol (83.2 vs. 76.7). On the MATH, WizardMath-RL-IRM-PRM achieves 48.8% accuracy, improving by 1.8% over WizardMath-RL-PRM and **5.3%** over WizardMath-SFT-Llama3-Evol (48.8 vs. 43.5).
>
> These results demonstrate that our RLEIF approach, utilizing open-source models like Llama-3-70B-Instruct for data synthesis and Llama-3.1-405B-Instruct for reward model data annotation, is effective in significantly enhancing the model's mathematical reasoning capabilities without reliance on proprietary models such as GPT-4. This highlights the effectiveness and scalability of the approach. **Consequently, open-source models like Llama-3-70B-Instruct for instruction evolution and Llama-3.1-405B-Instruct for IRM and PRM data annotation offer a cost-efficient and accurate alternative to proprietary models, striking a favorable balance between cost and performance**.
>
> | **Models**                     | Data Synthesis Model        | **GSM8K** | **MATH** |
> |---------------------------------|------------------------------|-----------|----------|
> | Mistral-7B-v0.1                | -                            | 42.9      | 12.9     |
> | WizardMath-SFT-GPT-4-Evol      | GPT-4                        | 82.8      | 48.1     |
> | &nbsp;&nbsp;&nbsp;&nbsp; + ORM                    | GPT-4                        | 84.6      | 49.6     |
> | &nbsp;&nbsp;&nbsp;&nbsp; + PRM                    | GPT-4                        | 87.2      | 52.7     |
> | &nbsp;&nbsp;&nbsp;&nbsp; + IRM + PRM              | GPT-4                        | 90.7      | 55.4     |
> | WizardMath-SFT-Llama3-Evol     | Llama-3-70B-Instruct         | 76.7      | 43.5     |
> | &nbsp;&nbsp;&nbsp;&nbsp; + ORM                    | Llama-3.1-405B-Instruct      | 78.1      | 44.6     |
> | &nbsp;&nbsp;&nbsp;&nbsp; + PRM                    | Llama-3.1-405B-Instruct      | 80.3      | 47.0     |
> | &nbsp;&nbsp;&nbsp;&nbsp; + IRM + PRM              | Llama-3.1-405B-Instruct      | 83.2      | 48.8     |

---

> > ### Author Response · Authors · 2024-11-28
> >
> > > **Weaknesses-2**:  While RLEIF has undergone extensive evaluation in the context of mathematical reasoning, it may also hold potential for applications in other reasoning domains. Have the authors considered applying RLEIF to other reasoning tasks like logical deduction or causal inference? It would be valuable to see if the benefits observed in mathematical reasoning transfer to these domains, and what modifications might be necessary.
> >
> > We sincerely appreciate your thoughtful question and insightful suggestions, as well as your recognition of the effectiveness of the RLEIF method. Beyond its application to mathematical reasoning tasks, we have also explored extending the entire RLEIF pipeline to code reasoning tasks, as detailed below.
> >
> > During the SFT stage, we replicated the code Evol-Instruct method specifically proposed by WizardCoder for code-related tasks and further optimized the PRM step-level label prompts to improve compatibility with GPT-4 for annotating code-specific PRM training data. Additionally, we compared the performance of ORM and PRM during PPO training. We utilized CodeLlama-Python 7B and 34B as the base models.
> >
> > As shown in the table below, the results demonstrate that for both the CodeLlama-Python 7B and 34B models, Our-Coder-SFT achieved comparable performance to WizardCoder on the HumanEval and MBPP benchmarks in the SFT stage.
> >
> > During the PPO training phase, when using CodeLlama-Python 7B as the base model, Our-Coder-RL-PRM showed a 4%-5% improvement on HumanEval and MBPP over Our-Coder-SFT, and significantly outperformed the 2%-3% improvement achieved by Our-Coder-RL-ORM. Similarly, with CodeLlama-Python 34B as the base model, Our-Coder-RL-PRM shows approximately a 4% improvement over Our-Coder-SFT on HumanEval and MBPP, outperforming the 2%-3% improvement of Our-Coder-RL-ORM. These findings underscore the effectiveness of PRM in PPO training for code-related tasks.
> >
> > Notably, the effectiveness of Code Evol-Instruct in code-related tasks has been demonstrated by WizardCoder. Therefore, the above analysis indicates that the entire RLEIF pipeline is also applicable and effective for enhancing models' code reasoning capabilities. However, the process of designing task-specific Evol-Instruct prompts, PRM step-level labeling prompts, and synthesizing training data for SFT and PRM is highly time-consuming and resource-intensive. Due to constraints on time and computational resources, we are unable to explore more reasoning tasks. We promise to further apply the RLEIF method to more other reasoning tasks, such as logical deduction and causal inference in future camera-ready version, along with comprehensive exploration. Moreover, we also aspire to extend the RLEIF framework to a broader range of domains and will continue to actively explore its potential through ongoing research.
> >
> > | **Models**                                 | **Base**            | **Params** | **HumanEval** | **MBPP** |
> > |--------------------------------------------|---------------------|------------|---------------|----------|
> > | **CodeLlama-Python-7B as the base model**  |                     |            |               |          |
> > | CodeLlama-Python                           | -                   | 7B         | 37.8          | 57.6     |
> > | WizardCoder                                | CodeLlama-Python   | 7B         | 48.2          | 56.6     |
> > | Our-Coder-SFT                              | CodeLlama-Python   | 7B         | 49.0          | 56.2     |
> > | Our-Coder-RL-ORM                           | CodeLlama-Python   | 7B         | 50.5          | 58.1     |
> > | Our-Coder-RL-PRM                           | CodeLlama-Python   | 7B         | 53.5          | 60.4     |
> > | **CodeLlama-Python-34B as the base model** |                     |            |               |          |
> > | CodeLlama-Python                           | -                   | 34B        | 51.8          | 67.2     |
> > | WizardCoder                                | CodeLlama-Python   | 34B        | 73.2          | 73.2     |
> > | Our-Coder-SFT                              | CodeLlama-Python   | 34B        | 72.7          | 72.3     |
> > | Our-Coder-RL-ORM                           | CodeLlama-Python   | 34B        | 74.5          | 73.7     |
> > | Our-Coder-RL-PRM                           | CodeLlama-Python   | 34B        | 76.8          | 76.2     |

---

> > > ### Author Response · Authors · 2024-11-28
> > >
> > > Due to time and space limitations, we commit to integrating these discussions above into the relevant sections of the main text in future camera-ready version of our paper to further improve the quality of our research.
> > >
> > > We sincerely hope that the responses provided above can address your concerns. We deeply appreciate your recognition and kind words regarding our work. Your feedback is invaluable, and we warmly welcome any additional comments or suggestions you may have. We would be more than happy to engage in further discussions or clarify any remaining questions. Once again, thank you for your thoughtful and meticulous review of our paper, as well as your meaningful contributions to improving our work.
> > >
> > > Respectfully,
> > >
> > > Paper 4894 Authors.

---

### Official Review · Reviewer_8UVP · 2024-11-01

**Soundness:** 4
**Presentation:** 3
**Contribution:** 4
**Rating:** 8
**Confidence:** 3

**Summary:**

This paper proposes a method to enhance math reasoning using Reinforced-Evol-Instruct, called WizardMath. The process begins with Supervised Fine-Tuning (SFT) on evol-instruct data, followed by training Instruction Reward Model (IRM) and Process Reward Model (PRM). Finally, they train PPO using the trained IRM and PRM. The WizardMath model achieves high performance on GMS8K and Math datasets.

**Strengths:**

The paper is well-written, with comprehensive analysis and ablation experiments.

It represents a valuable exploration of process supervision and IRM in math reasoning, achieving impressive performance.

At the same time, their method is data efficient.

**Weaknesses:**

NA

**Questions:**

NA

---

> ### Author Response · Authors · 2024-11-28
>
> Dear Reviewer 8UVP,
>
> We sincerely extend our heartfelt gratitude for your recognition and commendation of our work, as well as for your thorough engagement with both our paper and rebuttal. We deeply appreciate the significant time and effort you dedicated to the review process, along with your insightful comments and positive feedback, which have been invaluable in our research.
>
> Once again, we are profoundly grateful for your thoughtful evaluation and generous recognition of our work.
>
> Respectfully,
>
> Paper 4894 Authors.

---

### Official Review · Reviewer_8CQg · 2024-11-02

**Soundness:** 4
**Presentation:** 3
**Contribution:** 3
**Rating:** 8
**Confidence:** 3

**Summary:**

This paper proposes the Reinforcement Learning from Evol-Instruct Feedback (RLEIF) framework to enhance the mathematical reasoning capabilities of language models. RLEIF involves two main steps: instruction tuning with MATH EVOL-INSTRUCT and reinforcement learning using an Instruction Reward Model (IRM) and a Process Reward Model (PRM).

The experimental results demonstrate that their model, after both supervised fine-tuning (SFT) and reinforcement learning (RL), surpasses many existing math-focused language models that have undergone only SFT.

The ablation study highlights the crucial role of the IRM during the reinforcement learning phase. Additionally, they show that their PRM, which utilizes GPT-4 as a step annotator, delivers superior performance compared to Math-Shepherd and PRM800K.

**Strengths:**

This paper is well-written and rich in detail. The introduction of the Instruction Reward Model (IRM) is novel and useful. The experiments are comprehensive, and the analysis is thorough, providing deep insights into the framework's effectiveness. I believe the idea of integrating IRM with PRM could be useful for any math LLMs (only undergoing SFT).

**Weaknesses:**

The primary concern with this paper is the unfair comparison of baseline models in the results. While the authors claim that both supervised fine-tuning (SFT) with Math Evol-Instruct and reinforcement learning (RL) with the Instruction Reward Model (IRM) and Process Reward Model (PRM) are beneficial for enhancing mathematical reasoning, these approaches—SFT with synthesized data and the use of various reward models for RL—represent parallel research lines.

- In Table 1, the authors compare their model, which has undergone both SFT and RL, with models that have only undergone SFT. This comparison is unfair because these SFT models could also be further enhanced with RL techniques to improve mathematical reasoning (e.g., using ORM for RL on DartMath). It would be more appropriate to isolate the effects of SFT and RL for a fair comparison. In Table 1, the authors should compare the performance of models that have undergone SFT with Math Evol-Instruct against existing baselines such as MetaMath and DartMath. Additionally, comparisons with baselines like MetaMath and DartMath on the LLaMA-3.2 backbone would be valuable, as their training data is publicly available.

- In analyzing the impact of training data size, the authors should compare their approach with the best method for SFT using synthesized data, specifically DartMath. MetaMath, which was developed around a year ago, uses GPT-3.5-turbo for data augmentation, making it an outdated and potentially unfair baseline.

- It appears that SFT with Math Evol-Instruct yields inferior results compared to other SFT methods. From Table 4, the LLaMA2-7B: WizardMath-SFT scores 35.6 on MATH, which lags behind models like XwinMath and Skywork. Likely, it would also lag behind LLaMA2-7B fine-tuned on the DartMath training data. This suggests that the main contribution of the paper is in the RL component. Therefore, the primary focus should be on the results obtained with different reward models, as presented in Table 4, utilizing various SFT backbones.

- Table 7 lacks adequate baselines; at least, the authors should include LLaMA-2-7B trained on the DartMath training set. This table also suffers from the same fairness issues as Table 1.

I recommend that the authors reorganize the paper better to emphasize their contributions to the "RL part."

**Questions:**

- In Equation (1), how is the parameter ( m ) set? Additionally, how is the Instruction Reward Model (IRM) trained?

- Lines 256-258 suggest that you retain solutions with incorrect answers. How might this influence the results? Have you considered using the IRM to filter out low-quality examples for supervised fine-tuning (SFT)?

- During PPO, do you use two reward models? Using two reward models in PPO can be time-consuming and computationally expensive. What are your strategies for addressing this?

- In Lines 89-90, you state that you "innovatively introduce PRM to address the False-Positive issue in the problem-solving process." This claim should be validated by comparing the false-positive rate on a test set both with and without your method.

- In Lines 88-89, you mention that existing methods "mainly focus on the SFT stage and are susceptible to learning hallucinated information from the teacher model." However, in Line 95, you still use GPT-4 to annotate step-level labels. Isn’t there a risk of obtaining incorrect step labels from GPT-4 as well?

---

> ### Author Response · Authors · 2024-11-24
>
> Dear Reviewer 8CQg,
>
> We thank you for your valuable comments and the time you spent reviewing our work! Your professional feedback provides valuable guidance for writing a more comprehensive and competitive paper. Below, we provide detailed responses to the **Weaknesses** and **Questions** raised in your review of our paper, addressing each point systematically.
>
> It is worth noting that we also have added these discussions with the Reviewer-8CQg about the weaknesses and questions of our paper **in Appendix C.1 of our latest upload of revised paper (pages 36–49, lines 1892–2589)**.
>
> Please find below a detailed discussion of the points you have raised:
>
> > **Weaknesses-1**: The primary concern with this paper is the unfair comparison of baseline models in the results. While the authors claim that both supervised fine-tuning (SFT) with Math Evol-Instruct and reinforcement learning (RL) with the Instruction Reward Model (IRM) and Process Reward Model (PRM) are beneficial for enhancing mathematical reasoning, these approaches—SFT with synthesized data and the use of various reward models for RL—represent parallel research lines.
>
>
>
> We sincerely appreciate your attention to our work and your careful and responsible review and thank you for your valuable suggestions. To ensure a fair comparison, we conducted evaluations using WizardMath-SFT against all current state-of-the-art (SOTA) models across different scales of base models, as presented **in Table 21 and Table 22, Appendix C.1.2 of our latest upload of revised paper (pages 38–39, lines 2009–2105)**. The results confirm the effectiveness of our proposed Math Evol-Instruct approach. Meanwhile, during the PPO training stage, we applied IRM and PRM to different SFT backbones, significantly enhancing the mathematical reasoning ability of these models. This demonstrates the effectiveness and generalizability of our IRM and PRM methods. Please refer to Weaknesses 1.1–1.4 below for details.
>
> Below, we provide detailed responses to Weaknesses 1.1– Weaknesses1.4 in the order you were raised.
>
>
> > **Weaknesses-1.1**: In Table 1, the authors compare their model, which has undergone both SFT and RL, with models that have only undergone SFT. This comparison is unfair because these SFT models could also be further enhanced with RL techniques to improve mathematical reasoning (e.g., using ORM for RL on DartMath). It would be more appropriate to isolate the effects of SFT and RL for a fair comparison. In Table 1, the authors should compare the performance of models that have undergone SFT with Math Evol-Instruct against existing baselines such as MetaMath and DartMath. Additionally, comparisons with baselines like MetaMath and DartMath on the LLaMA-3.2 backbone would be valuable, as their training data is publicly available.
>
> We sincerely appreciate your insightful questions and detailed observations. To provide a more comprehensive and fair comparison, we have included the WizardMath-SFT results **in Table 21 and Table 22, Appendix C.1.2 of our latest upload of revised paper (pages 38--39, lines 2009--2105)**. These results evaluate the performance of **WizardMath-SFT**, trained exclusively using SFT, against current SOTA models across various base models. The key findings are summarized as follows:
>
> 1. **Performance Comparison:**
>
>    (1). On Llama-2-7B and Mistral-7B-v0.1, WizardMath-SFT performs marginally below SOTA models (i.e., Xwin-Math and Skywork-Math) and outperforms other excellent models (i.e., DART-Math).
>
>     (2). On Llama-2-13B and Llama-2-70B, WizardMath-SFT achieves comparable performance to Xwin-Math and surpasses KPMath-Plus.
>
>    (3). On all various base models, WizardMath-SFT surpasses most existing SOTA models trained solely with SFT (i.e., DART-Math).
>
>    Notably, WizardMath-SFT achieves these results using only 418K synthetic data points, a significantly smaller dataset compared to DART-Math (580K–590K), Xwin-Math (1440K), and Skywork-Math (2500K).
>
> 3. **Comparison with advanced data synthesis methods (i.e., DART-Math, MetaMath)**
>
>    As shown in the following table, DART-Math demonstrates strong performance across various base models, and the data synthesis method proposed by DART-Math shows its effectiveness and outstanding performance. Meanwhile, WizardMath-SFT demonstrates comparable or superior performance to advanced data synthesis methods, such as DART-Math and MetaMath, across all base models. Key observations include:
>
>    (1). On Mistral-7B-v0.1 and DeepSeekMath, WizardMath-SFT performs on par with DART-Math (Uniform & Prop2Diff) on GSM8k and surpasses DART-Math (Uniform & Prop2Diff) on MATH.
>
>    (2). On Llama3.2-1B, Llama3.2-3B, Llama3-8B, Llama3.1-8B, and Llama2-7B, WizardMath-SFT exhibits a 2%\~7% improvement over DART-Math (Uniform & Prop2Diff) on the GSM8k benchmark. On the MATH benchmark, WizardMath-SFT outperforms DART-Math (Uniform & Prop2Diff) by approximately 5%\~10%.

---

> ### Author Response · Authors · 2024-11-24
>
> **[ Continue the response to above Weaknesses-1.1]**
>
>
> These findings highlight the effectiveness of the proposed **Math Evol-Instruct** for enhancing mathematical reasoning capabilities.
>
> Notably, to ensure the same training settings as in our paper during the SFT stage, we employ a learning rate of 2e-5 for the Llama series base models (i.e., Llama2 7B, Llama3.1 8B, Llama3.2 1B, and Llama3.2 3B) and a learning rate of 5e-6 for Mistral-7B-v0.1. All models are trained for 3 epochs with a batch size of 256, and 4 checkpoints are saved per epoch. Finally, we select the checkpoint with the highest accuracy on the GSM8k and MATH benchmarks for reporting.
>
> **We have added the discussions about the Weaknesses-1.1 in Appendix C.1.2 of our latest upload of revised paper (pages 36–40, lines 1921–2135)**.
>
> | **Model**                         | **Base**         | **Params** | **GSM8k** | **MATH** |
> |-----------------------------------|------------------|------------|-----------|----------|
> | **Llama-3.2-1B as Base Model**    |||||
> | DART-Math-Prop2Diff               | Llama 3.2       | 1B         | 49.2      | 23.4     |
> | MetaMath                          | Llama 3.2       | 1B         | 51.9      | 15.5     |
> | DART-Math-Uniform                 | Llama 3.2       | 1B         | 55.8      | 22.0     |
> | **WizardMath-Llama-SFT**          | Llama 3.2       | 1B         | **57.1**      | **29.7**    |
> | **Llama-3.2-3B as Base Model**    |||||
> | MetaMath                          | Llama 3.2       | 3B         | 72.6      | 25.9     |
> | DART-Math-Prop2Diff               | Llama 3.2       | 3B         | 74.0      | 37.8     |
> | DART-Math-Uniform                 | Llama 3.2       | 3B         | 77.8      | 36.4     |
> | **WizardMath-Llama-SFT**          | Llama 3.2       | 3B         | **80.3**      | **45.2**    |
> | **Llama-2-7B as Base Model**      |||||
> | MetaMath                          | Llama-2         | 7B         | 66.5      | 19.8     |
> | DART-Math-Prop2Diff               | Llama-2         | 7B         | 69.9      | 30.7     |
> | DART-Math-Uniform                 | Llama-2         | 7B         | 73.8      | 29.5     |
> | **WizardMath-Llama-SFT**          | Llama-2         | 7B         | **77.4**      | **35.6**    |
> | **Mistral-7B-v0.1 as Base Model** |||||
> | MetaMath                          | Mistral-v0.1    | 7B         | 77.9      | 28.6     |
> | DART-Math-Prop2Diff               | Mistral-v0.1    | 7B         | 81.1      | 45.5     |
> | DART-Math-Uniform                 | Mistral-v0.1    | 7B         | 82.6      | 43.5     |
> | **WizardMath-Mistral-SFT**        | Mistral-v0.1    | 7B         | **82.8**      | **48.1**    |
> | **DeepSeekMath-7B as Base Model** |||||
> | DART-Math-Prop2Diff               | DeepSeekMath    | 7B         | 86.8      | 53.6     |
> | DART-Math-Uniform                 | DeepSeekMath    | 7B         | 88.2      | 52.9     |
> | **WizardMath-DeepSeek-SFT**       | DeepSeekMath    | 7B         | **88.9**      | **58.2**    |
> | **Llama-3-8B as Base Model**      |||||
> | MetaMath                          | Llama 3         | 8B         | 77.3      | 20.6     |
> | DART-Math-Prop2Diff               | Llama 3         | 8B         | 81.1      | 46.6     |
> | DART-Math-Uniform                 | Llama 3         | 8B         | 82.5      | 45.3     |
> | **WizardMath-Llama-SFT**          | Llama 3         | 8B         | **88.9**      | **53.3**     |
> | **Llama-3.1-8B as Base Model**    |||||
> | MetaMath                          | Llama 3.1       | 8B         | 80.4      | 35.4     |
> | DART-Math-Prop2Diff               | Llama 3.1       | 8B         | 84.3      | 46.5     |
> | DART-Math-Uniform                 | Llama 3.1       | 8B         | 86.7      | 45.1     |
> | **WizardMath-Llama-SFT**          | Llama 3.1       | 8B         | **89.2**      | **55.8**     |

---

> ### Author Response · Authors · 2024-11-24
>
> > **Weaknesses-1.2**: In analyzing the impact of training data size, the authors should compare their approach with the best method for SFT using synthesized data, specifically DartMath. MetaMath, which was developed around a year ago, uses GPT-3.5-turbo for data augmentation, making it an outdated and potentially unfair baseline.
>
> Thank you for your insightful advice. **In Appendix C.1.3 of our latest upload of revised paper (pages 40--41, lines 2137--2154), Figure 5**, we explore the performance of WizardMath Evol-instruct in comparison with DART-Math and MetaMath across different training data scales on the GSM8k and MATH benchmarks in the SFT stage.
>
> As the volume of training data increases, WizardMath-Evol-Instruct consistently improves its performance on the GSM8k and MATH benchmarks, exhibiting a slightly higher growth rate than DART-Math.
>
> In the initial stages, WizardMath slightly underperforms compared to DART-Math. This advantage may stem from DART-Math being distilled from DeepSeekMath-RL, an advanced mathematical reasoning model pre-trained on 120B high-quality mathematical tokens, showcasing exceptional proficiency in mathematical reasoning. However, once the dataset exceeds 60k, its performance begins to surpass DART-Math. At a data scale of 390k, WizardMath slightly outperforms DART-Math by 2%\~3% on GSM8k and by 5%\~6% on MATH.
>
> Additionally, WizardMath-Evol-Instruct consistently exceeds MetaMath at the same data scales, achieving increases of 3%\~6% on GSM8k and 15%\~20% on MATH. This performance gain is attributed to the efficiency of Math Evol-Instruct's upward and downward evolution processes.
>
> These findings demonstrate that our Math Evol-Instruct method is also as scalable and effective as DART-Math for the large-scale synthetic data.
>
> **We have added the discussions about the Weaknesses-1.2 in Appendix C.1.3 of our latest upload of revised paper (pages 40-41, lines 2139--2189)**.
>
>
> > **Weaknesses-1.3**: It appears that SFT with Math Evol-Instruct yields inferior results compared to other SFT methods. From Table 4, the LLaMA2-7B: WizardMath-SFT scores 35.6 on MATH, which lags behind models like XwinMath and Skywork. Likely, it would also lag behind LLaMA2-7B fine-tuned on the DartMath training data. This suggests that the main contribution of the paper is in the RL component. Therefore, the primary focus should be on the results obtained with different reward models, as presented in Table 4, utilizing various SFT backbones.
>
> Thank you for your valuable questions and insightful suggestions. Below are detailed responses to each question.
>
> >>**Q1:** It appears that SFT with Math Evol-Instruct yields inferior results compared to other SFT methods. From Table 4, the LLaMA2-7B: WizardMath-SFT scores 35.6 on MATH, which lags behind models like XwinMath and Skywork. Likely, it would also lag behind LLaMA2-7B fine-tuned on the DartMath training data.
>
> In the following table , we show the performance comparison of WizardMath-SFT with DART-Math, Xwin-Math, and Skywork-Math on the Llama2-7B base model on the MATH benchmark.
>
> - **WizardMath-SFT vs. DART-Math:**
>   WizardMath-SFT, based on the Llama2-7B model, outperforms DART-Math-Uniform by 6.1% and DART-Math-Prop2Diff by 4.9% on the MATH benchmark. Notably, the amount of data used by WizardMath-SFT is only 70%\~71% of DART-Math (418k vs. 591k; 418k vs. 585k).
>
> - **WizardMath vs. Xwin-Math:**
>   Although WizardMath-SFT is 5% lower than Xwin-Math on the MATH benchmark, the amount of data used is only 29.0% of Xwin-Math (418k vs. 1440k), which is much less than Xwin-Math. Moreover, Xwin-Math leverages GPT-4-turbo for data synthesis. However, WizardMath-SFT outperforms Xwin-Math on the MATH when using different backbones such as Mistral-7B-v0.1, Llama2-13B, and Llama2-70B as shown **in Table 22 of our latest upload of revised paper (pages 39, lines 2052–2105)**. For instance, in Table 22, WizardMath-SFT exceeds Xwin-Math by 4.4% (48.1% vs. 43.7%) when using the Mistral-7B-v0.1 as the base model.
>
> - **WizardMath vs. Skywork-Math:**
>   WizardMath-SFT underperforms Skywork-Math-2500k on the MATH benchmark by 12.1%, but it uses only 16.7% of the amount of data used by Skywork-Math-2500k (418k vs. 2500k), which is much less than Skywork-Math. Furthermore, according to **Figure 5 About Synthetic Data Size in the Skywork-Math paper[1]**, Skywork-Math-720k scores 34.54% on MATH, and Skywork-Math-360k scores 29.36%. Therefore, WizardMath-SFT-418k performs comparably to Skywork-Math-720k on MATH (35.6% vs. 34.54%), and with the same amount of data, WizardMath-SFT outperforms Skywork-Math.

---

> > ### Author Response · Authors · 2024-11-24
> >
> > **[ Continue the response to above Weaknesses-1.3]**
> >
> >
> > In summary, the Math Evol Instruct data synthesis method proposed in our study is as effective and practical as  the current state-of-the-art data synthesis methods, such as DART-Math, Skywork-Math and Xwin-Math in the SFT stage. It significantly enhances the mathematical reasoning capabilities of the model, marking a key contribution of our work. Additionally, we acknowledge the contributions of methods such as DART-Math, Skywork-Math, and Xwin-Math, which are excellent data synthesis approaches excelling in generating high-quality datasets for mathematical tasks and significantly enhancing models' mathematical reasoning capabilities.
> >
> > [1] Zeng L, Zhong L, Zhao L, et al. Skywork-Math: Data Scaling Laws for Mathematical Reasoning in Large Language Models--The Story Goes On[J]. arXiv preprint arXiv:2407.08348, 2024.
> >
> > | **Llama2 7B as the base model** | **Data size** | **MATH** |
> > |------------------------------|---------------|-----------|
> > | DART-Math-Uniform           | 591k          | 29.5      |
> > | DART-Math-Prop2Diff         | 585k          | 30.7      |
> > | Xwin-Math                   | 1440k         | 40.6      |
> > | Skywork-Math                | 360k          | 29.36     |
> > | Skywork-Math                | 720k          | 34.54     |
> > | Skywork-Math                | 2500k         | 47.7      |
> > | WizardMath-SFT              | 418k          | 35.6      |
> >
> >
> > >>**Q2:** This suggests that the main contribution of the paper is in the RL component. Therefore, the primary focus should be on the results obtained with different reward models, as presented in Table 4, utilizing various SFT backbones.
> >
> > Thank you for your deep insights. The following table  shows the impact of applying the proposed Instruction Quality Scoring Reward Model (IRM) and Process Supervised Reward Model (PRM) to PPO training across various SFT backbones (i.e., DART-Math, MetaMath, and Xwin-Math). The results demonstrate that incorporating our IRM and PRM during PPO training led to a performance improvement of 5% to 8% on both GSM8k and MATH for most SFT models. For instance:
> >
> > - **When using DART-Math as the SFT backbone based on Llama2-7B:**
> >   On GSM8k, after reinforcement learning training with IRM and PRM, Prop2Diff-RL improved by 6.9% (69.9% vs. 76.8%), and Uniform-RL improved by 5.3% (73.8% vs. 79.1%).
> >   On MATH, Prop2Diff-RL achieved a 6.4% gain (30.7% vs. 37.1%), and Uniform-RL improved by 5.7% (29.5% vs. 35.2%).
> >
> > - **When using DART-Math as the SFT backbone based on Mistral-7B-v0.1:**
> >   On GSM8k, Prop2Diff-RL improved by 6.4% (81.1% vs. 87.5%), and Uniform-RL increased by 5.5% (82.6% vs. 88.1%).
> >   On MATH, Prop2Diff-RL rose by 5.9% (45.5% vs. 51.4%), and Uniform-RL saw a 5.2% enhancement (43.5% vs. 48.9%).
> >
> > - **For the MetaMath models based on Llama2-7B and Mistral-7B-v0.1:**
> >   Training with PPO using IRM and PRM led to performance improvements of 8% to 9% on GSM8k and 5% to 8% on MATH.
> > - **Similarly, for the Xwin-Math-Llama2-7B model, performance on both GSM8k and MATH improved by 6% to 8%.**
> >
> >
> >
> >
> > | **Model**                         | **Base**       | **Params** | **GSM8k** | **MATH** |
> > |-----------------------------------|----------------|------------|-----------|----------|
> > | **Llama-2-7B as the base model**      |||||
> > | MetaMath-SFT                      | Llama-2        | 7B         | 66.5      | 19.8     |
> > | MetaMath-RL                       | Llama-2        | 7B         | 75.6      | 25.1     |
> > | DART-Math-Prop2Diff-SFT           | Llama-2        | 7B         | 69.9      | 30.7     |
> > | DART-Math-Prop2Diff-RL            | Llama-2        | 7B         | 76.8      | 37.1     |
> > | DART-Math-Uniform-SFT             | Llama-2        | 7B         | 73.8      | 29.5     |
> > | DART-Math-Uniform-RL              | Llama-2        | 7B         | 79.1      | 35.2     |
> > | Xwin-Math-SFT                     | Llama-2        | 7B         | 82.6      | 40.6     |
> > | Xwin-Math-RL                      | Llama-2        | 7B         | 88.2      | 48.5     |
> > | WizardMath-Llama-SFT              | Llama-2        | 7B         | 77.4      | 35.6     |
> > | WizardMath-Llama-RL               | Llama-2        | 7B         | 84.1      | 43.5     |
> > | **Mistral-7B-v0.1 as the base model** |||||
> > | MetaMath-SFT                      | Mistral-v0.1   | 7B         | 77.9      | 28.6     |
> > | MetaMath-RL                       | Mistral-v0.1   | 7B         | 86.4      | 35.2     |
> > | DART-Math-Prop2Diff-SFT           | Mistral-v0.1   | 7B         | 81.1      | 45.5     |
> > | DART-Math-Prop2Diff-RL            | Mistral-v0.1   | 7B         | 87.5      | 51.4     |
> > | DART-Math-Uniform-SFT             | Mistral-v0.1   | 7B         | 82.6      | 43.5     |
> > | DART-Math-Uniform-RL              | Mistral-v0.1   | 7B         | 88.1      | 48.7     |
> > | WizardMath-Mistral-SFT            | Mistral-v0.1   | 7B         | 82.8      | 48.1     |
> > | WizardMath-Mistral-RL             | Mistral-v0.1   | 7B         | 90.7      | 55.4     |

---

> ### Author Response · Authors · 2024-11-24
>
> **[ Continue the response to above Weaknesses-1.3]**
>
>
> These findings highlight the significant contributions of our IRM and PRM during reinforcement learning, consistently enhancing the mathematical reasoning abilities of our SFT models while achieving robust generalization on different SFT backbones. This represents a key contribution of our study.
>
> Thus, our study primarily makes two core contributions:
>
> **1. The proposed Math Evol Instruct data synthesis method is also as effective and practical as  the current state-of-the-art data synthesis methods, such as DART-Math, Skywork-Math and Xwin-Math in the SFT stage**. It also significantly enhances the mathematical reasoning capabilities of our models.
>
> **2. The proposed IRM and PRM models substantially improve performance during the reinforcement learning phase. They not only continuously enhance the mathematical reasoning abilities of our SFT models but also achieve strong generalization across various SFT backbones (i.e., DART-Math)**. Outstanding performance is demonstrated on the GSM8k and MATH benchmarks.
>
> **We have added the discussions about the Weaknesses-1.3 in Appendix C.1.4 of our latest upload of revised paper (pages 41–43, lines 2187–2309).**
>
> > **Weaknesses-1.4**: Table 7 lacks adequate baselines; at least, the authors should include LLaMA-2-7B trained on the DartMath training set. This table also suffers from the same fairness issues as Table 1.
>
> Thank you for your constructive feedback. The table below presents the performance of WizardMath-SFT on 7 out-of-domain (OOD) evaluation tasks covering K-12, college, and competition-level math problems in the SFT stage. The results indicate that **WizardMath-SFT** consistently surpasses state-of-the-art open-source models (i.e., DART-Math, Xwin-Math, and MathScale) across various scales and tasks, achieving an average improvement of **3%\~6%**. For instance:
>
> - With the Llama2-7B base model, WizardMath-SFT outperformed DART-Math-Uniform by **11.0%** (38.3% vs. 27.3%) and DART-Math-Prop2Diff by **10.5%** (38.3% vs. 27.8%) on average.
> - With the Mistral-7B base model, WizardMath-SFT achieved an average improvement of **5.9%** over DART-Math-Uniform (43.5% vs. 37.6%) and **4.6%** over DART-Math-Prop2Diff (43.5% vs. 38.9%).
>
>
> These findings highlight the effectiveness of our **Math Evol-Instruct** method, demonstrating its robustness and superior generalization capabilities on out-of-domain tasks.
>
> **We have added the discussions about the Weaknesses-1.4 in Appendix C.1.5 of our latest upload of revised paper (pages 43–44, lines 2310–2363).**

---

> > ### Author Response · Authors · 2024-11-24
> >
> > **[ Continue the response to above Weaknesses-1.4]**
> >
> >
> > | **Models**               | **College Math** | **TAL** | **Math23k** | **Ape210k** | **Gaokao Bench Math** | **AGIE Gaokao Math** | **AGIE SAT Math** | **AVG** |
> > |---------------------------|------------------|---------|-------------|-------------|------------------------|-----------------------|-------------------|---------|
> > | **Proprietary models**    |                  |         |             |             |                        |                       |                   |         |
> > | GPT-4                    | **24.4**       | **51.8**| **76.5**    | **61.5**    | **35.4**               | **28.2**              | **68.6**          | **49.5**|
> > | GPT-3.5-Turbo            | 21.6             | 42.9    | 62.5        | 44.0        | 23.2                   | 15.3                  | 55.8              | 37.9    |
> > | **Models based on LLaMA-2 13B** |           |         |             |             |                        |                       |                   |         |
> > | LLaMA-2 13B              | 1.2              | 6.3     | 9.5         | 7.9         | 0.7                    | 0.4                   | 6.8               | 4.7     |
> > | MAmmoTH-CoT              | 6.5              | 17.3    | 39.5        | 28.1        | 5.9                    | 4.9                   | 20.5              | 17.5    |
> > | MetaMath                 | 10.1             | 25.4    | 48.6        | 31.6        | 9.6                    | 5.6                   | 38.2              | 24.2    |
> > | MathScale 13B            | 20.4             | 38.1    | 61.1        | 43.7        | 20.0                   | 12.3                  | 55.8              | 35.9    |
> > | WizardMath-SFT           | 22.2             | 42.5    | 65.9        | 47.6        | 31.6                   | 23.5                  | 59.7              | 41.9    |
> > | WizardMath-RL            | **22.9**    | **43.3**| **70.3**    | **50.8**    | **33.1**               | **25.7**              | **64.7**          | **44.4**|
> > | **Models based on LLaMA-2 7B** |            |         |             |             |                        |                       |                   |         |
> > | LLaMA-2 7B               | 2.3              | 7.6     | 6.8         | 7.3         | 2.1                    | 2.9                   | 2.9               | 4.6     |
> > | MAmmoTH-CoT              | 6.2              | 13.3    | 34.6        | 21.4        | 3.9                    | 2.7                   | 19.6              | 14.5    |
> > | MetaMath                 | 9.4              | 22.5    | 44.0        | 29.9        | 5.9                    | 5.1                   | 36.2              | 21.9    |
> > | DART-Math-Uniform        | 12.0             | 27.3    | 47.9        | 32.9        | 14.8                   | 11.1                  | 45.1              | 27.3    |
> > | DART-Math-Prop2Diff      | 11.9             | 27.7    | 49.9        | 34.3        | 12.8                   | 10.6                  | 47.1              | 27.8    |
> > | Xwin-Math-V1.1           | 14.9             | 29.7    | 59.6        | 40.8        | 15.9                   | 8.4                   | 51.0              | 31.5    |
> > | MathScale 7B             | 20.9             | 35.2    | 59.0        | 41.8        | 19.6                   | 12.6                  | 57.8              | 35.3    |
> > | WizardMath-SFT           | 21.1             | 38.5    | 62.4        | 43.8        | 26.3                   | 17.7                  | 58.3              | 38.3    |
> > | WizardMath-RL            | **21.2**         | **40.2**| **67.3**    | **46.1**    | **28.9**               | **18.7**              | **62.7**          | **40.7**|
> > | **Models based on Mistral 7B** |             |         |             |             |                        |                       |                   |         |
> > | Mistral 7B               | 7.5              | 17.9    | 18.5        | 15.5        | 6.2                    | 5.9                   | 22.5              | 13.4    |
> > | MetaMath Mistral         | 15.7             | 31.4    | 55.1        | 38.1        | 15.3                   | 10.1                  | 50.9              | 30.9    |
> > | DART-Math-Uniform        | 19.4             | 34.8    | 61.6        | 44.8        | 27.0                   | 16.1                  | 59.8              | 37.6    |
> > | MathScale Mistral        | 21.8             | 39.9    | 64.4        | 46.0        | 21.4                   | 14.3                  | 57.8              | 37.9    |
> > | DART-Math-Prop2Diff      | 19.9             | 37.4    | 62.2        | 44.9        | 27.2                   | 18.1                  | 62.7              | 38.9    |
> > | WizardMath-Mistral-SFT   | 24.3             | 42.7    | 66.6        | 49.7        | 35.2                   | 22.7                  | 63.1              | 43.5    |
> > | WizardMath-Mistral-RL    | **24.8**         | **44.8**| **71.2**    | **52.6**    | **37.2**               | **24.5**              | **64.7**          | **45.7**|

---

> > > ### Author Response · Authors · 2024-11-24
> > >
> > > > **2. Recommend**: I recommend that the authors reorganize the paper better to emphasize their contributions to the "RL part."
> > >
> > > Thank you for your deep insights and constructive suggestions. Due to time and space constraints, we promise to further emphasize our contributions to the "RL part" in future revisions of our paper. Specifically, we will provide more detailed descriptions of the contributions of our proposed RLEIF approach to the RL part in some sections (i.e., the Abstract, Introduction, and Experiment Sections).
> > >
> > > For instance, we will highlight that in RL training, we firstly propose the instruction quality scoring reward model combined with the process supervision reward model, not only continuously enhancing the mathematical reasoning abilities of the SFT model but also achieving strong generalization across various SFT backbones. Additionally, we will supplement the discussion on the application and impact of IRM and PRM on different advanced SFT backbones, as highlighted in the Weaknesses-1.3, to further strengthen the theoretical framework and experimental analysis.
> > >
> > > Thank you very much for your insightful questions and valuable suggestions. Below, we provide responses to your Question-1 through Question-5 in sequence.
> > >
> > > > **Questions-1**: In Equation (1), how is the parameter ( m ) set? Additionally, how is the Instruction Reward Model (IRM) trained?
> > >
> > > >>**Q1:** In Equation (1), how is the parameter (m) set?
> > >
> > > The parameter (m) denotes the margin in the Pairwise Ranking Loss, acting as a threshold to regulate the score difference between <Choose, Reject> pairs. Specifically, it ensures that during IRM training, the reward score for higher-quality instructions surpasses that of lower-quality instructions by at least the margin value. This mechanism encourages the model to emphasize the quality score gap between high-quality and low-quality instructions. In our experiments, the parameter (m) was set to a constant 1.
> > >
> > > >>**Q2:** Additionally, how is the Instruction Reward Model (IRM) trained?
> > >
> > > **In our paper, Section 3.2 <REWARD MODELS>, lines 187-201,** we conducted two rounds of downward evolution and three rounds of upward evolution based on the original instructions, generating a total of five evolved instructions. Subsequently, we leverage GPT-4 to rank the quality between those evolved instructions and the original instruction based on the difficulty and definition, with higher ranks assigned to instructions demonstrating greater difficulty and clearer definitions. The detailed ranking prompt template is provided in Appendix A.2.
> > >
> > > From the ranking results of the 6 instructions, we created 15 positive-negative sample pairs by combining C(6, 2). Applying this five-round evolution process to 15k original instructions, we ultimately generated 15k x 15 = 225k positive-negative pairs for training IRM data.
> > >
> > > During training, we employed the Pairwise Ranking Loss defined in Eq.1. For a given mathematical instruction q, the IRM quantifies its quality by assigning a score. The IRM was initialized with the SFT model and augmented with a header layer that outputs a scalar score. The design of the Pairwise Ranking Loss draws inspiration from the reward model training methods described in the Instruct-GPT paper[1].
> > >
> > > **We have added the answers about the Questions-1 in Appendix C.3.1 of our latest upload of revised paper (page 45, lines 2383–2415).**
> > >
> > > [1] Ouyang L, Wu J, Jiang X, et al. Training language models to follow instructions with human feedback[J]. Advances in neural information processing systems, 2022, 35: 27730-27744.

---

> ### Author Response · Authors · 2024-11-24
>
> > **Questions-2**: Lines 256-258 suggest that you retain solutions with incorrect answers. How might this influence the results? Have you considered using the IRM to filter out low-quality examples for supervised fine-tuning (SFT)?
>
> >>**Q1:** Lines 256-258 suggest that you retain solutions with incorrect answers. How might this influence the results?
>
> **The Lines 256-258, Sections 4.2 for SFT Training Data in our paper**, to prevent data leakage, we filter out the evolved data with high similarity to the GSM8k and MATH test sets, **so it does not refer to incorrect answers**. The data leakage detection method refers to the paper Appendix A.11, lines 1782–1815. Specifically, we employ instructions in the GSM8k and MATH test set as queries to retrieve the top-5 samples from all evolved training data with an embedding model, gte-large. Additionally, we employ GPT-4 to provide similarity judgment between the test sets and the retrieved samples, and remove the similar instructions. The prompt and additional details are provided in Appendix A.12.
>
> The table  below demonstrates the impact of unfiltering the potential data leaks on model performance. WizardMath-SFT-No-filter-data-leakage outperforms WizardMath-SFT-Filter-data-leakage by 1.3% on the GSM8k and by 1.7% on the MATH. We use Mistral-7B-v0.1 as the base model.
>
> | **Model**                                | **GSM8k** | **MATH** |
> |------------------------------------------|-----------|----------|
> | WizardMath-SFT-Filter-data-leakage       | 82.8      | 48.1     |
> | WizardMath-SFT-No-Filter-data-leakage    | 84.1      | 49.8     |
>
> >>**Q2:** Have you considered using the IRM to filter out low-quality examples for supervised fine-tuning (SFT)?
>
> Thank you very much for your insightful question and constructive suggestions. The table below highlights the effects of using IRM to filter out low-quality instructions during the SFT stage.
>
> - Filtering 15k low-quality instructions resulted in WizardMath-SFT-filter-15k outperforming WizardMath-SFT-original, with a 1.8-point improvement on GSM8k and a 2.1-point improvement on MATH.
> - Filtering 30k low-quality instructions improved GSM8k by 0.9% and MATH by 0.6%.
> - However, when the filtering reached 45k, WizardMath-SFT-filter-45k showed a performance decrease of 0.8% on GSM8k and 1.1% on MATH.
> - Filtering up to 60k resulted in a more pronounced decline, with WizardMath-SFT-filter-60k dropping by 1.7% on GSM8k and 2.5% on MATH.
>
> These results indicate that using IRM for moderate filtering of low-quality data (i.e., 15k or 30k) is effective for enhancing model performance, while excessive filtering can lead to significant performance degradation.
>
> **We have added the discussions about the Questions-2 in Appendix C.3.2  of our latest upload of revised paper(pages 45-46, lines 2416–2470).**
>
> | **Model**                    | **IRM Filter Data Size** | **GSM8k** | **MATH** |
> |------------------------------|--------------------------|-----------|----------|
> | WizardMath-SFT-original      | -                        | 82.8      | 48.1     |
> | WizardMath-SFT-filter-15k    | 15k                      | 84.6      | 50.2     |
> | WizardMath-SFT-filter-30k    | 30k                      | 83.7      | 48.7     |
> | WizardMath-SFT-filter-45k    | 45k                      | 82.0      | 47.0     |
> | WizardMath-SFT-filter-60k    | 60k                      | 81.1      | 45.6     |
>
>
> > **Questions-3**: During PPO, do you use two reward models? Using two reward models in PPO can be time-consuming and computationally expensive. What are your strategies for addressing this?
>
> >>**Q1:** During PPO, do you use two reward models?
>
> Yes, we use two reward models during the PPO training stage.
>
> >>**Q2:** Using two reward models in PPO can be time-consuming and computationally expensive. What are your strategies for addressing this?
>
> During the PPO training stage, we utilized the DeepSpeedChat[1] framework. To improve training efficiency and reduce memory consumption, we employed several optimization techniques, including DeepSpeed ZeRO-3 with CPU Offload, the DeepSpeed-Hybrid Engine, MixZ++, Gradient Checkpointing, Gradient Accumulation, and BFloat16 precision.

---

> ### Author Response · Authors · 2024-11-24
>
> **[ Continue the response to above Questions-3]**
>
> In the future, we try to implement GRPO by Deepseekmath[2] (a variant of PPO) for training and incorporate the VLLM[3] used by the OpenRLHF[4] framework to accelerate the policy model generation during PPO training, thus improving the training efficiency.
>
> **We have added the answer about Questions-3 in Appendix C.3.3  of our latest upload of revised paper(pages 46-47, lines 2472–2497).**
>
> [1] Yao Z, Aminabadi R Y, Ruwase O, et al. Deepspeed-chat: Easy, fast and affordable rlhf training of chatgpt-like models at all scales[J]. arXiv preprint arXiv:2308.01320, 2023.
>
> [2] Shao Z, Wang P, Zhu Q, et al. Deepseekmath: Pushing the limits of mathematical reasoning in open language models[J]. arXiv preprint arXiv:2402.03300, 2024.
>
> [3] Kwon W, Li Z, Zhuang S, et al. Efficient memory management for large language model serving with pagedattention[C]//Proceedings of the 29th Symposium on Operating Systems Principles. 2023: 611-626.
>
> [4] Hu J, Wu X, Wang W, et al. OpenRLHF: An Easy-to-use, Scalable and High-performance RLHF Framework[J]. arXiv preprint arXiv:2405.11143, 2024.
>
> > **Questions-4**: In Lines 89-90, you state that you "innovatively introduce PRM to address the False-Positive issue in the problem-solving process." This claim should be validated by comparing the false-positive rate on a test set both with and without your method.
>
> Thank you for your insightful feedback. We utilized GPT-4o-2024-0513 (the accuracy is 96.1% on GSM8k and 76.6% on MATH) to annotate the step-by-step correctness of responses generated by WizardMath-SFT, WizardMath-RL-ORM, and WizardMath-RL-PRM on the GSM8k and MATH test sets, and we calculated the model's false-positive rates.
>
> We define the **false-positive rate** as the proportion of responses in the test set where the final answer is correct, but errors occur in intermediate steps (i.e., computational or logical mistakes). The formula for calculating the False Positive Rate is as follows:
>
>
> **False Positive Rate = (Number of False Positives / Total Number of Test Sets)**
>
> The table below presents the statistical results:
>
> - The false-positive rate of WizardMath-SFT is 2.58% on GSM8k and 2.36% on MATH.
> - The false-positive rate of WizardMath-RL-ORM is 1.67% on GSM8k and 1.56% on MATH.
> - The false-positive rate of WizardMath-RL-PRM is 0.68% on GSM8k and 0.90% on MATH.
>
> Compared to WizardMath-SFT, WizardMath-RL-PRM reduced the false-positive rate by 1.90% on GSM8k and 1.46% on MATH. Similarly, compared to WizardMath-RL-ORM, WizardMath-RL-PRM achieved reductions of 0.99% on GSM8k and 0.66% on MATH.
>
> These results demonstrate that the incorporation of PRM significantly reduces the model's false-positive rate, effectively alleviating the occurrence of intermediate step errors during the problem-solving process.
>
> **We have added the discussions about Questions-4 in Appendix C.3.4 of our latest upload of the revised paper (page 47, lines 2500–2530).**
>
> | **Metrics**                      | **WizardMath-SFT** | **WizardMath-RL-ORM** | **WizardMath-RL-PRM** |
> |----------------------------------|--------------------|------------------------|------------------------|
> | **Reward Model for PPO**         | -                  | ORM                    | PRM                    |
> | **Number of GSM8k test sets**    | 1319               | 1319                   | 1319                   |
> | **Number of False Positive On GSM8k** | 34             | 22                     | 9                      |
> | **False Positive Rate On GSM8k** | 2.58%              | 1.67%                  | 0.68%                  |
> | **Number of MATH test sets**     | 5000               | 5000                   | 5000                   |
> | **Number of False Positive On MATH** | 118            | 78                     | 45                     |
> | **False Positive Rate On MATH**  | 2.36%              | 1.56%                  | 0.90%                  |
>
> > **Questions-5**: In Lines 88-89, you mention that existing methods "mainly focus on the SFT stage and are susceptible to learning hallucinated information from the teacher model." However, in Line 95, you still use GPT-4 to annotate step-level labels. Isn’t there a risk of obtaining incorrect step labels from GPT-4 as well?
>
> Yes, there is a potential risk of obtaining incorrect step labels from GPT-4 as well.
>
> The risk of the model learning hallucinatory information from the teacher model cannot be completely eliminated. Therefore, in order to ensure the reliability and effectiveness of using GPT-4 to annotate step-level labels during the problem-solving process in constructing PRM training data, we conducted the following two analyses:

---

> ### Author Response · Authors · 2024-11-24
>
> **[ Continue the response to above Questions-5]**
>
> 1. **Reliability of GPT-4 Annotations**
>
> Manually annotating large-scale step-level PRM training data demands extensive mathematical expertise, making it a challenging, time-intensive, and costly process. So, we employ a fully AI-powered automatic annotation using GPT-4 in our paper. To assess the reliability of GPT-4-generated annotations, in the early stages, we randomly selected 2k samples from the manually labeled PRM800k step-level training dataset and annotated them using GPT-4. GPT-4 annotations were evaluated against human annotations using the F1 score as a consistency metric. The results showed an F1 consistency of 78.1% between GPT-4 and human annotations.
>
> Additionally, for the GSM8k training set, which is relatively lower in difficulty, we randomly sampled 200 examples for step-level labeling using GPT-4 and manual annotations. The results show that the F1 consistency between GPT-4 and manual labeling on GSM8k is 87.2%. These findings demonstrate that the annotation using GPT-4 with manual evaluation exhibits high consistency on GSM8k and MATH, thus ensuring the reliability of step-level annotation using GPT-4 for PRM training data.
>
> 2. **Effectiveness of GPT-4 Annotations**
>
> The table below and Table 4 in our paper (lines 382–395, 415–424) discussed the impact of AI-labeled PRM data on model performance compared to manually labeled PRM800k and Math-Shepherd data generated via MCTS Tree Search. The experimental results reveal that the PRM trained on our fully AI-labeled data outperforms both the manually annotated PRM800k and Math-Shepherd. For instance:
>
> - When training WizardMath-Llama2-7B-SFT with PPO, GPT-4-labeled PRM data surpasses PRM800k by 2.0% and Math-Shepherd by 1.4% on GSM8k, and by 1.2% and 1.7%, respectively, on MATH.
> - Similarly, with WizardMath-Mistral-7B-SFT trained using PPO, GPT-4-labeled PRM data outperforms PRM800k by 1.8% and Math-Shepherd by 1.1% on GSM8k, and by 1.9% and 2.4%, respectively, on MATH.
>
> Moreover, PRM outperforms ORM by 2%\~3% on both GSM8k and MATH, achieving a notable improvement of 4%\~5% on WizardMath-SFT. These results highlight the effectiveness of GPT-4-labeled data for PRM training. **(It is worth noting that our evolved instructions lack correct answers, limiting compatibility with the methods employed by Math-Shepherd which needs the correct answers.)**
>
> The analysis above underscores both the reliability and effectiveness of using GPT-4 to annotate step-level PRM training data. However, we acknowledge that GPT-4 annotations are not immune to errors, and the possibility of incorrect step labels represents a limitation of this approach.
>
> **We have added the discussions about the Questions-5 in Appendix C.3.5 of our latest upload of revised paper (pages 47–49, lines 2532–2590).**
>
> | **Models**                              | **GSM8K** | **MATH** |
> |-----------------------------------------|-----------|----------|
> | **Llama2-7B as the base model**         |||
> | Llama2-7B: WizardMath-SFT               | 77.4      | 35.6     |
> | &nbsp;&nbsp;&nbsp;&nbsp;+ ORM (ours)    | 79.1      | 36.8     |
> | &nbsp;&nbsp;&nbsp;&nbsp;+ PRM800k       | 79.7      | 38.7     |
> | &nbsp;&nbsp;&nbsp;&nbsp;+ Math-Shepherd | 80.3      | 38.2     |
> | &nbsp;&nbsp;&nbsp;&nbsp;+ PRM (ours)    | **81.7**  | **39.9** |
> | **Mistral-7B-v0.1 as the base model**   |||
> | Mistral-7B: WizardMath-SFT              | 82.8      | 48.1     |
> | &nbsp;&nbsp;&nbsp;&nbsp;+ ORM (ours)    | 84.6      | 49.6     |
> | &nbsp;&nbsp;&nbsp;&nbsp;+ PRM800k       | 85.4      | 50.8     |
> | &nbsp;&nbsp;&nbsp;&nbsp;+ Math-Shepherd | 86.1      | 50.3     |
> | &nbsp;&nbsp;&nbsp;&nbsp;+ PRM (ours)    | **87.2**  | **52.7** |
>
> It is worth noting that in the latest uploaded revision, we have added the above detailed discussions with Reviewer-8CQg regarding the paper's weaknesses and questions in Appendix C.1  (pages 36–49, lines 1892–2589). **Due to time and space limitations, we commit to integrating these discussions into the relevant sections of the main text in future revisions. Additionally, we extend our heartfelt gratitude to the exceptional open-source mathematical data synthesis methods, such as DART-Math, Xwin-Math, and Skywork-Math**. These contributions have substantially enhanced the mathematical reasoning capabilities of the models and significantly advanced the development of the LLM for Math open-source community.
>
> We hope that the responses above can address your concerns. We are looking forward to receiving any additional feedback you may have and are very happy to engage in any follow-up discussions or address any additional comments. Thank you once again for your valuable contributions to our work.
>
> Respectfully,
>
> Paper 4894 Authors.

---

> > ### Comment · Reviewer_8CQg · 2024-11-25
> >
> > Thank you for your detailed responses and extra experiments conducted! All my concerns and questions are addressed! The detailed analysis of each part is really impressive. For the rebuttal, I am **raising my score to 8** to reflect the frankness and contribution of the authors.

---

> > > ### Author Response · Authors · 2024-11-28
> > >
> > > Dear Reviewer 8CQg,
> > >
> > > We would like to express our heartfelt gratitude for your recognition and kind words regarding our work. We deeply appreciate the time and effort you devoted to thoroughly engaging with both our paper and rebuttal, as well as your decision to improve its score. Your professional comments and valuable feedback have been incredibly insightful and instrumental in refining our research.
> > >
> > > We are committed to incorporating the relevant discussions  above into  the future camera-ready version of our paper to further enhance the quality of our research and makes a meaningful contribution to advancing research within the LLM community.
> > >
> > > Once again, we sincerely thank you for your thoughtful consideration, constructive feedback, and generous support of our work.
> > >
> > > Respectfully,
> > >
> > > Paper 4894 Authors.

---

### Official Review · Reviewer_dHFe · 2024-11-04

**Soundness:** 3
**Presentation:** 2
**Contribution:** 4
**Rating:** 8
**Confidence:** 3

**Summary:**

The paper takes Evol-Instruct (from the WizardLM paper) and extends it to the math domain, while at the same time integrating process reward models into the training pipeline.

## Method
- Generate questions of various complexities by prompting GPT to sequentially generate easier questions ("downard evolution") and harder questions ("upward evolution")
- Train Instruction Reward Model (IRM) to predict quality of instruction
- Train Process Reward Model (PRM) to predict quality of each individual step
- Use PPO

## Experiments
- GSM8k and MATH datasets
- Main base models are Llama3 and Mistral. Experiments done across various scales.
- Outperforms a number of strong closed-source models such as GPT and Claude2

**Strengths:**

**1. Strong Results** -- I put a lot of premium on this strength and use this to justify my overall rating. Many of the gains from training on Math Evol-Instruct are more than 10 points. More importantly, it is quite impressive to design something that outperforms strong proprietary models, so if this method is as strong as the paper claims, then this is something that the community will definitely quickly pick up on.

**2. Thorough Experiments and Baseline Comparisons** -- Various scales ranging from 100M to 1B to 70B, with different base models. Has comparisons with several math-specific models (e.g. Mammoth, Math-Shepherd, etc.) I also liked the analysis and ablations in Section 4.4

**3. Scalable Method** -- The whole process is fully automated, which makes it scalable. I imagine this can also be adapted to other process-intensive reasoning domains outside math (e.g. coding).

**Weaknesses:**

**1. PRM labels from GPT-4** -- Not really sure what to think of this. On one hand, I feel such direct distillation like this would limit the effectiveness of a method at larger data scales. On the other hand, the results seem to be good (and also this is one key part that makes the process fully AI-automated.)

**2. Unclear presentation** -- The paper assumes that readers are already previously familiar with Evol-Instruct, as it devotes very little time to talking about it in the intro or related work. The narrative is messy -- there are certain concepts (e.g. "grade school" and "high school" questions) that were introduced once out of nowhere then never mentioned again. There are a number of rows on Table 1 that were never discussed in the main text. Figure 1 is difficult to understand. There are also several typos and poorly-worded sentences.

**3. Somewhat marginal contribution** -- Evol-Instruct previously existed. PRM previously existed. This paper basically took Evol-Instruct and PRM and used them to train a model. To nitpick a bit, I think a more comprehensive paper would cover more domains such as code.

**Questions:**

- I find Figure 1 confusing. Why is there a pyramid in the top left and why is it pointing to a pie chart, cube, etc? What are these supposed to be showing? I feel like I am not understanding much from this figure.
- I feel like there's a few missing entries in Table 1. For example, Table 1 shows the results for WizardMath-Mathstral and WizardMath-Qwen2.5, but the base scores of these base models are not shown in the table, so the readers don't really know how much improvement there is.

Typos:
- L45: struggles --> struggle
- L80: "in recent"
- L91: should say what is IRM first before using the acronym.
- L91: We --> capitalization
- L93: later --> latter
- L105: Jiang et al mentioned without a model (should be mistral?)
- L107: as following --> as follows
- L140,143: spacing
- L145: should be Reinforcement Learning for Large Language Models instead of the other way around?
- L487: "reasoing"

---

> ### Author Response · Authors · 2024-11-28
>
> Dear Reviewer dHFe,
>
> We sincerely thank you for your insightful comments and the time you dedicated to reviewing our work. Your expert feedback has been invaluable in guiding us towards refining our paper and making it more comprehensive and competitive. We greatly appreciate your support and constructive suggestions. In the following, we offer detailed responses to the **Weaknesses** and **Questions** raised in your review, addressing each point in a systematic manner.
>
> Furthermore, **in Appendix C.4 of our latest upload of revised paper (pages 49–57, lines 2606–3077)**, we also have added the discussions with the Reviewer-dHFe on the weaknesses and questions of our paper to respond to the Reviewer-dHFe's comments and to further improve the quality of our research.
>
> Please find below a detailed discussion of the points you have raised:
>
>
> > **Weaknesses-1**:  PRM labels from GPT-4 -- Not really sure what to think of this. On one hand, I feel such direct distillation like this would limit the effectiveness of a method at larger data scales. On the other hand, the results seem to be good (and also this is one key part that makes the process fully AI-automated.)
>
> Thank you very much for your constructive feedback and recognition of the effectiveness of our approach. To explore the effectiveness and reliability of using GPT-4 to annotate PRM training data at larger data scales, we conducted a Data Scaling law analysis of using GPT-4 to annotate PRM training data. Additionally, we explore the feasibility of leveraging open-source models (i.e., Llama-3.1-405B-Instruct) as cost-effective alternatives to GPT-4 for annotation.
>
> **1. Effectiveness Analysis of the Scaling Law in GPT-4 Annotated PRM Training Data.**
>
> **1.1 Impact of PRM Data Scaling when PRM as the Verifier for the Best of N Metric.**
>
> To assess the influence of data scale when PRM acts as the verifier, we randomly sampled subsets of 50k, 150k, and 300k data from our total 450k PRM training dataset. Models were trained on these subsets, and we evaluated the Best of N metric on GSM8k and MATH as shown in the following table. Following the same settings as Table 5 in our paper, we sampled 256 answers for each problem, scored them with the PRM Verifier, and selected the highest-scoring answer. Key results are summarized as follows:
>
> - On GSM8k, the Best of N performance of PRM significantly improved as the training data size increased. For instance, PRM-450k achieved 95.2%, outperforming PRM-300k by 1.6% and PRM-150k by 2.9%.
> - On MATH, PRM-450k reached 64.7%, marking a 1.4% improvement over PRM-300k and a 3.2% improvement over PRM-150k.
>
> | **Generators**    | **Verifiers** | **PRM Data Size** | **GSM8K** | **MATH** |
> |-------------------|---------------|-------------------|-----------|----------|
> | WizardMath-SFT    | PRM-50k       | 50k               | 89.6      | 58.9     |
> | WizardMath-SFT    | PRM-150k      | 150k              | 92.3      | 61.5     |
> | WizardMath-SFT    | PRM-300k      | 300k              | 93.6      | 63.3     |
> | WizardMath-SFT    | PRM-450k      | 450k              | 95.2      | 64.7     |
>
> **1.2 Impact of PRM Data Scaling on PPO Training Performance.**
>
> In the following table, we further investigated the effect of PRM training data scaling during the PPO training stage. Increasing the PRM data size yielded substantial performance gains for WizardMath-RL on GSM8k and MATH:
>
> - On GSM8k, PPO training with PRM-450k achieved 87.2%, surpassing PRM-300k by 1.4% and PRM-50k by 3.7%.
> - On MATH, PPO training with PRM-450k reached 52.7%, exceeding PRM-300k by 1.5% and PRM-50k by 4.0%.
>
> These findings confirm that scaling PRM training data consistently enhances PRM performance as a Verifier on the Best of N metric and significantly improves PPO training outcomes. This validates the effectiveness of GPT-4 annotated PRM training data in adhering to the Data Scaling Law, demonstrating their robustness and utility even at larger data scales.
>
> | **Models**                    | **GSM8K** | **MATH** |
> |-------------------------------|-----------|----------|
> | Mistral-7B: WizardMath-SFT    | 82.8      | 48.1     |
> | &nbsp;&nbsp;&nbsp;&nbsp; + PRM-50k               | 83.5      | 48.7     |
> | &nbsp;&nbsp;&nbsp;&nbsp; + PRM-150k              | 84.9      | 49.8     |
> | &nbsp;&nbsp;&nbsp;&nbsp; + PRM-300k              | 85.8      | 51.2     |
> | &nbsp;&nbsp;&nbsp;&nbsp; + PRM-450k              | **87.2**  | **52.7** |

---

> ### Author Response · Authors · 2024-11-28
>
> **[ Continue the response to above Weaknesses-1]**
>
> **2. Reliability analysis of GPT-4 labeled PRM training data compared to manual labeling**
>
> To assess the reliability of GPT-4-generated annotations, in the early stages, we randomly selected 2k samples from the manually labeled PRM800k step-level training dataset and annotated them using GPT-4. GPT-4 annotations were evaluated against human annotations using the F1 score as a consistency metric. The results showed an F1 consistency of 78.1% between GPT-4 and human annotations.
>
> Additionally, for the GSM8k training set, which is relatively lower in difficulty, we randomly sampled 200 examples for step-level labeling using GPT-4 and manual annotations. The results show that the F1 consistency between GPT-4 and manual labeling on GSM8k is 87.2%. These findings demonstrate that the annotation using GPT-4 with manual evaluation exhibits high consistency on GSM8k and MATH, thus ensuring the reliability of step-level annotation using GPT-4 for PRM training data.
>
> **3. Feasibility of using advanced open-source models instead of GPT-4 to label PRM training data**
>
> We realize that there is a high cost of directly distilling GPT-4 in large-scale data scenarios, which is a limitation of this study. Additionally, manual annotation demands mathematical expertise and entails a challenging, time-intensive, and costly process. Moreover, our evolved instructions lack correct answers, limiting compatibility with the methods employed by Math-Shepherd[1], which needs the correct answers.
>
> To mitigate these challenges, we also explore the feasibility of leveraging advanced open-source models, such as Llama-3.1-405B-Instruct, instead of GPT-4 for PRM training data labeling, using the same label prompts and training settings. As shown in the table below, WizardMath-PRM-Llama-3.1-405B-Instruct achieves **85.8%** on the GSM8k, marking a 3.0% improvement over WizardMath-SFT and lagging behind WizardMath-PRM-GPT-4 by **1.4%**. On the MATH, it achieves a score of **51.5%**, representing a **3.4%** improvement over WizardMath-SFT with a **1.2%** gap compared to WizardMath-PRM-GPT-4.
>
> Balancing cost and accuracy, Llama-3.1-405B-Instruct demonstrates considerable potential as a substitute for GPT-4 in PRM training data labeling.
>
> In conclusion, GPT-4-based labeled PRM data also follows the data scaling law and offers an effective solution for larger data scales. For scenarios requiring a balance between cost and accuracy, advanced open-source models like **Llama-3.1-405B-Instruct** provide a viable alternative. We hope these analyses above can address your concerns.
>
> [1]. Wang P, Li L, Shao Z, et al. Math-shepherd: Verify and reinforce LLMs step-by-step without human annotations[C]//Proceedings of the 62nd Annual Meeting of the Association for Computational Linguistics (Volume 1: Long Papers). 2024: 9426-9439.
>
> | **Models**                   | **AI-Label**             | **GSM8k** | **MATH** |
> |------------------------------|--------------------------|-----------|----------|
> | WizardMath-SFT               | -                        | 82.8      | 48.1     |
> | &nbsp;&nbsp;&nbsp;&nbsp; + PRM-Llama-3.1-405B-Instruct | Llama-3.1-405B-Instruct | 85.8      | 51.5     |
> | &nbsp;&nbsp;&nbsp;&nbsp; + PRM-GPT-4            | GPT-4                    | 87.2      | 52.7     |

---

> ### Author Response · Authors · 2024-11-28
>
> > **Weaknesses-2**: Unclear presentation -- The paper assumes that readers are already previously familiar with Evol-Instruct, as it devotes very little time to talking about it in the intro or related work. The narrative is messy -- there are certain concepts (e.g. "grade school" and "high school" questions) that were introduced once out of nowhere then never mentioned again. There are a number of rows on Table 1 that were never discussed in the main text. Figure 1 is difficult to understand. There are also several typos and poorly-worded sentences.
>
> I sincerely appreciate the valuable questions you have raised. Please allow me to address them one by one.
>
> >>**Q1:** Unclear presentation -- The paper assumes that readers are already previously familiar with Evol-Instruct, as it devotes very little time to talking about it in the intro or related work.
>
> Thank you for highlighting the valuable questions and we sincerely apologize for any inconvenience caused. We have added an introduction to Evol-Instruct **in our latest upload of revised paper (Section 1 <INTRODUCTION>, Lines 50–82)**. A more comprehensive description can be found in Appendix C.4.2, with the relevant details outlined below. In future camera-ready version of our paper, we promise to integrate this section in the "Introduction" or "Related Work" sections and provide a more comprehensive explanation of Evol-Instruct. The relevant detailed descriptions as follows:
>
> Evol-Instruct proposed by WizardLM[1] is an innovative framework designed to automate the generation of diverse and complex open-domain instructions using large language models (LLMs). Instead of relying on human-crafted instruction datasets, it leverages the generative capabilities of LLMs to iteratively evolve an initial set of instructions through two complementary strategies: In-depth Evolving and In-breadth Evolving. Starting from an initial dataset D_0, it iteratively evolves instructions over M  turns, producing datasets (D_1, D_2, ..., D_M).
>
> Two evolution strategies are employed:
>
> In-depth Evolving incrementally enhances instruction complexity by introducing additional constraints, deepening, concretizing, increasing reasoning steps, and complicating input, while maintaining logical coherence and ensuring instructions remain solvable by humans, I_k(t+1) = In-Depth Operation(I_k(t)).
>
> In-breadth Evolving focuses on improving topic diversity and dataset richness by creating entirely new instructions, expanding the coverage of skills and scenarios, particularly in underrepresented areas, I_k(t+1) = In-Breadth Operation(I_k(t)).
>
> To ensure dataset quality, failed evolutionary instructions are filtered. Evol-Instruct supports scalable, high-quality dataset creation, significantly enhancing LLM performance in reasoning and open-domain tasks. Notably, WizardCoder[2] incorporates instruction evolution specifically tailored to coding tasks, leading to substantial improvements in code generation capabilities.
>
> [1]. Xu C, Sun Q, Zheng K, et al. WizardLM: Empowering large pre-trained language models to follow complex instructions[C]//The Twelfth International Conference on Learning Representations. 2024.
>
> [2]. Luo Z, Xu C, Zhao P, et al. Wizardcoder: Empowering code large language models with evol-instruct[J]. arXiv preprint arXiv:2306.08568, 2023.
>
> >>**Q2:** The narrative is messy -- there are certain concepts (e.g. "grade school" and "high school" questions) that were introduced once out of nowhere then never mentioned again.
>
> We greatly appreciate you highlighting this writing issue and sincerely apologize for any inconvenience it may have caused. In our paper, we primarily evaluate the model's mathematical performance on two popular benchmarks: GSM8k and MATH. **GSM8k represents problems at the grade school level, while MATH focuses on high school competition problems**, such as AMC 10, AMC 12, and AIME. In Appendix C.4.2, we have included a detailed introduction to the GSM8k and MATH datasets. Additionally, **in our latest upload of revised paper, we have incorporated descriptions of "grade school" and "high school," as reflected in lines 90-91, 102-103, 114, 240, 340, and 537.**
>
> We also include a detailed description of the evaluation benchmark as follows:
>
> The GSM8k dataset contains approximately 7500 training data and 1319 test data, mainly on grade school level math problems, each of which consists of basic arithmetic operations (addition, subtraction, multiplication, and division), and generally requires 2 to 8 steps to solve. The MATH dataset collects math problems from prestigious math competitions such as AMC 10, AMC 12, and AIME. It contains 7500 training data and 5,000 challenging test data in seven academic areas: Prealgebra, Algebra, Number Theory, Counting and Probability, Geometry, Intermediate Algebra, and Precalculus.

---

> ### Author Response · Authors · 2024-11-28
>
> **[ Continue the response to above Weaknesses-2]**
>
> Furthermore, these problems on MATH are divided into five levels of difficulty, with ‘1’ denoting the relatively lower difficulty level and ‘5’ indicating the highest level.
>
>
>
> >>**Q3:** There are a number of rows on Table 1 that were never discussed in the main text.
>
> Thank you for pointing out this valuable question. Below, we present a detailed analysis of the performance improvements across various model scales (0.1B to 70B) with different base models on GSM8k and MATH benchmarks:
>
> **1. Using GPT-2 Series Models as the Base Model:**
> - **GPT-2-Small-0.1B:** WizardMath-GPT-2-Small improves by 19.5% (26.4 vs. 6.9) on GSM8k and 6.9% (12.3 vs. 5.4) on MATH.
> - **GPT-2-Medium-0.3B:** WizardMath-GPT-2-Medium enhances by 27.5% (38.7 vs. 11.2) on GSM8k and by 9.4% (15.6 vs. 6.2) on MATH, outperforming Llama2-13B.
> - **GPT-2-Large-0.7B:** WizardMath-GPT-2-Large increases by 36.5% (50.1 vs. 13.6) on GSM8k and by 14.8% (21.2 vs. 6.4) on MATH, surpassing Mistral-7B-v0.1.
> - **GPT-2-XL-1.5B:** WizardMath-GPT-2-XL shows a 43.5% (58.9 vs. 15.4) gain on GSM8k and 18.5% (25.4 vs. 6.9) on MATH, exceeding MAmmoTH-CoT-Llama2-13B.
>
> These results demonstrate that the RLEIF method significantly enhances the mathematical reasoning capabilities of GPT-2 series base models.
>
> **2. Using Llama Series Models as the Base Model:**
>
> - **Llama-3.2-1B:** WizardMath-Llama-3.2-1B improves by 18.9% (63.3 vs. 44.4) on GSM8k and by 2.9% (33.5 vs. 30.6) on MATH compared to Llama-3.2-1B-Instruct.
> - **Llama-3.2-3B:** WizardMath-Llama-3.2-3B enhances GSM8k by 7.8% (85.5 vs. 77.7) and MATH by 1.9% (49.9 vs. 48.0).
> - **Llama-2-7B:** WizardMath-Llama-2-7B achieves a 69.5% improvement on GSM8k (84.1 vs. 14.6) and 41.0% on MATH (43.5 vs. 2.5), surpassing Xwin-Math-Llama-2-7B, MathScale-Llama-2-7B, and MetaMath-Llama-2-7B.
> - **Llama-3-8B:** WizardMath-Llama-3-8B attains 90.3% on GSM8k (1.7% higher than Jiuzhang3.0) and 58.8% on MATH (7.8% higher than Jiuzhang3.0), also outperforming Baichuan-3, GLM-4, Gemini-Pro, Claude2, and GPT-3.5-Turbo, and is comparable to GPT-4-0314.
> - **Llama-2-13B:** WizardMath-Llama-2-13B improves GSM8k by 61.0% (89.7 vs. 28.7) and MATH by 46.7% (50.6 vs. 3.9), outperforming SOTA models such as Xwin-Math and KPMath-Plus.
> - **Llama-2-70B:** WizardMath-Llama-2-70B enhances GSM8k by 36.0% (92.8 vs. 56.8) and MATH by 45.1% (58.6 vs. 13.5).
>
> **3. Using Mistral Series Models as the Base Model:**
>
> - **Mistral-7B-v0.1:** WizardMath-Mistral-7B-v0.1 improves GSM8k by 47.8% (90.7 vs. 42.9) and MATH by 42.5% (55.4 vs. 12.9).
> - **Mistral-7B-v0.3:** WizardMath-Mistral-7B-v0.3 achieves 90.4% on GSM8k and 55.6% on MATH, comparable to WizardMath-Mistral-7B-v0.1.
> - **Mathstral-7B-v0.1:** WizardMath-Mathstral-7B-v0.1 attains 93.8% on GSM8k and 70.9% on MATH, comparable to GPT-4-Turbo-0125 and Claude 3.5 Sonnet, and superior to GPT-4 (original version).
>
> **4. Using DeepSeekMath as the Base Model:**
> WizardMath-DeepSeek improves GSM8k by 26.8% (91.0 vs. 64.2) and MATH by 28.4% (64.6 vs. 36.2), outperforming DART-Math and DeepSeekMath-RL.
>
> **5. Using Qwen2.5 Series Models as the Base Model:**
>
> - **Qwen2.5-Math-2.5B:** WizardMath-Qwen2.5-Math-2.5B achieves 86.7% on GSM8k and 68.6% on MATH.
> - **Qwen2.5-Math-7B:** WizardMath-Qwen2.5-Math-7B attains 93.9% on GSM8k and 77.8% on MATH.
> - **Qwen2.5-7B:** WizardMath-Qwen2.5-7B achieves 94.0% on GSM8k and 74.5% on MATH, performing comparably to GPT-4o-2024-0516 and Claude 3.5 Sonnet.
>
> The proposed RLEIF method significantly enhances the mathematical reasoning performance across various scales ranging from 0.1B to 70B with different base models, consistently outperforming all state-of-the-art open-source models. Notably, WizardMath-Mathstral-7B-v0.1 and WizardMath-Qwen2.5-Math-7B surpass some proprietary models such as GPT-4 (original version), Gemini-Pro, and GPT-3.5-Turbo, and perform comparably to GPT-4-Turbo-0125, GPT-4o-2024-0516, and Claude 3.5 Sonnet. These findings further underscore the effectiveness, robustness, and scalability of the proposed RLEIF method in our study.
>
> In future camera-ready versions of our paper, we promise to incorporate the results presented above into Section 4.3 <Main Results> and Appendix C.4.2, to provide a more comprehensive and in-depth analysis of the effectiveness of the proposed RLEIF method in enhancing the model's mathematical reasoning capabilities across a range of model scales (0.1B to 70B) with various base models.

---

> ### Author Response · Authors · 2024-11-28
>
> **[ Continue the response to above Weaknesses-2]**
>
> >>**Q4:** Figure 1 is difficult to understand.
>
> Thank you for highlighting these important questions and for pointing out any confusion caused by Figure 1. We sincerely apologize for any lack of clarity and greatly appreciate your valuable feedback. In future camera-ready versions of our paper, we are committed to providing a more comprehensive explanation of the diagram. Furthermore, we offer a detailed clarification of our method flow below, including the significance of colors and shapes as well as the direction of the arrows in Figure 1, to facilitate clearer understanding.
>
> In Figure 1, the various colored squares represent specific elements:
>
> - **Blue squares** denote original instructions,
> - **Orange squares** indicate evolved instructions,
> - **Cyan squares** signify model-generated solution processes, and
> - **Grey squares** correspond to a series of training-related operations such as supervised fine-tuning (SFT), reward modeling, and reinforcement learning (RL).
>
> To enhance the mathematical reasoning capabilities of large language models, we propose the **RLEIF method**, which integrates instruction evolution with reinforcement learning. This method consists of three primary steps:
>
> 1. **Instruction Evolution and SFT**
>    In the first step, we apply upward and downward instruction evolution on the GSM8k and MATH datasets, generating evolved instructions for the SFT. On the leftmost side of Figure 1, the three blue arrows, from top to bottom, represent:
>    - the adoption of the instruction evolution technique,
>    - the generation of evolved instruction data,
>    - its application to SFT training.
>
> 2. **Reward Model Training**
>    The second step involves two reward models: the **Instruction Quality Scoring Reward Model (IRM)** and the **Process-Supervised Reward Model (PRM)**, depicted in the central section of Figure 1.
>    - **IRM**: We employ upward and downward evolution on a seed instruction, yielding five instructions (original + evolved). These instructions are ranked by quality (e.g., C > A = E > B > D) using GPT-4. Based on the rankings, we train the Instruction Ranking Model (IRM) to assess instruction quality. In Figure 1, this process is shown in the left-central segment: "A" represents the original instruction, while "B," "C," "D," and "E" denote the evolved instructions. The first blue arrow illustrates the ranking process via GPT-4, the second arrow shows the ranking outcomes, and the third arrow highlights the use of this ranked data to train the IRM.
>    - **PRM**: In the middle-right section of Figure 1, the process for training the PRM is depicted. The SFT model generates step-by-step solutions from the given instructions, which are then evaluated and labeled by GPT-4. This labeled data is subsequently used to train the PRM.
>
> 3. **Reinforcement Learning with PPO**
>    In the final step, we integrate the IRM and PRM within a PPO-based reinforcement learning framework. As depicted in the far-right section of Figure 1, the process is as follows:
>    - The first blue arrow represents instruction scoring by the IRM.
>    - The second blue arrow shows PPO initialization and the start of reinforcement.
>    - The third blue arrow illustrates the policy model generating responses based on instructions.
>    - The fourth blue arrow shows the scoring of each response step using the PRM.
>    - Arrows five through eight depict the combination of IRM and PRM scores to calculate the final reward score.
>    - The ninth blue arrow highlights the use of the reward score for the PPO training.
>
> By integrating instruction evolution and reward-based optimization, the RLEIF method significantly enhances the reasoning capabilities of large language models. We hope this explanation resolves some ambiguities and provides a clearer understanding of Figure 1. Thank you again for your valuable suggestions, which will guide us in improving the presentation and clarity of our work.
>
> >>**Q5:** There are also several typos and poorly-worded sentences.
>
> We sincerely appreciate your thorough review and the time you dedicated to identifying these typos and poorly-worded sentences in our paper. Your attention to detail has been very invaluable. We have corrected all the issues you highlighted in our latest upload of revised paper.

---

> ### Author Response · Authors · 2024-11-28
>
> > **Weaknesses-3**: Somewhat marginal contribution -- Evol-Instruct previously existed. PRM previously existed. This paper basically took Evol-Instruct and PRM and used them to train a model. To nitpick a bit, I think a more comprehensive paper would cover more domains such as code.
>
> Thank you sincerely for raising these thoughtful and valuable questions. Please allow me to address them one by one.
>
> >>**Q1:** Somewhat marginal contribution -- Evol-Instruct previously existed. PRM previously existed. This paper basically took Evol-Instruct and PRM and used them to train a model
>
> We sincerely appreciate your valuable feedback. We highlight the key contributions of our paper as follows:
>
> **1. Unlike WizardLM/WizardCoder, which primarily focus on increasing instruction difficulty, we are the first to propose the novel concept of downward evolution, a major distinction in instruction evolution.**
>
> In Table 6 (lines 397–413) of our paper, we provide a detailed analysis of the effects of downward evolution. Specifically, two rounds of downward evolution led to a remarkable improvement in GSM8k performance by 14.8% (74.5 vs. 59.7) and in MATH performance by 19.6% (34.7 vs. 15.1) compared to the original, significantly enhancing the model's mathematical reasoning capabilities.
>
> Furthermore, our Math Evol-Instruct method outperforms the general evol-instruct approach employed by WizardLM, as elaborated in Appendix A.5 (lines 1608–1629).
>
> This demonstrates that Math Evol-Instruct is instrumental in significantly boosting the model’s mathematical reasoning ability, **as you kindly acknowledged in Strength 1 above**.
>
> **2. In reinforcement learning (RL) training, we firstly propose the instruction quality scoring reward model (IRM) combined with the process supervision reward model (PRM) further enhancing WizardMath mathematical reasoning ability.** As demonstrated in Table 3 (lines 325–336, lines 370-380) of our paper, this approach achieves a remarkable 7%–9% improvement on GSM8k and MATH performance over the SFT backbone across models of various sizes, leveraging PRM and IRM for the PPO training. Notably, employing IRM can achieve a significant 3%–5% improvement .
>
> **3. We first propose to use AI to annotate the step-level PRM training data.** Additionally, the training datasets for SFT, PRM, and IRM are fully synthesized using AI systems. This fully AI-automated data generation pipeline ensures effectiveness and scalability, **as highlighted in Strength 3 of your feedback**.
>
> **4. WizardMath demonstrates outstanding performance across a wide range of model scales, from 100M to 1B and 70B parameters, on benchmarks such as GSM8k, MATH, and out-of-distribution (OOD) tasks like MWPBench.** It surpasses all existing open-source state-of-the-art models, showcasing the effectiveness and robustness of the RLEIF approach proposed in our study, **as you recognized in Strength 1 above**.

---

> ### Author Response · Authors · 2024-11-28
>
> **[ Continue the response to above Weaknesses-3]**
>
> >>**Q2:** I think a more comprehensive paper would cover more domains such as code.
>
> We sincerely appreciate your insightful suggestions. To explore the effectiveness of our proposed RLEIF method in more other domains such as Code, we replicated the code evol-instruct specifically proposed by WizardCoder for code-related tasks during the SFT stage, and further optimized the PRM step-level label prompts to enhance its compatibility with GPT-4 for annotating PRM training data. Additionally, we compared the performance of ORM and PRM during PPO training. We utilized CodeLlama-Python 7B and 34B as the base models.
>
> As shown in the table below, the results demonstrate that for both the CodeLlama-Python 7B and 34B models, Our-Coder-SFT achieved comparable performance to WizardCoder on the HumanEval and MBPP benchmarks. During the PPO training phase, when using CodeLlama-Python-7B as the base model, Our-Coder-RL-PRM showed a 4%–5% improvement on HumanEval and MBPP over Our-Coder-SFT, and significantly outperformed the 2%–3% improvement achieved by Our-Coder-RL-ORM.
>
> Similarly, with CodeLlama-Python-34B as the base model, Our-Coder-RL-PRM shows approximately a 4% improvement over Our-Coder-SFT on HumanEval and MBPP, outperforming the 2%–3% improvement of Our-Coder-RL-ORM. These findings underscore the effectiveness of PRM in PPO training for code-related tasks.
>
> In future camera-ready version of our paper, we commit to conducting comprehensive comparisons across more code benchmarks and a broader range of baseline models to further validate the effectiveness of the proposed RLEIF approach in code task.
>
> | **Models**                                 | **Base**           | **Params** | **HumanEval** | **MBPP** |
> |--------------------------------------------|--------------------|------------|---------------|----------|
> | **CodeLlama-Python-7B as the Base model**                   |                    |            |               |          |
> | CodeLlama-Python                           | -                  | 7B         | 37.8          | 57.6     |
> | WizardCoder                                | CodeLlama-Python   | 7B         | 48.2          | 56.6     |
> | Our-Coder-SFT                              | CodeLlama-Python   | 7B         | 49.0          | 56.2     |
> | Our-Coder-RL-ORM                           | CodeLlama-Python   | 7B         | 50.5          | 58.1     |
> | Our-Coder-RL-PRM                           | CodeLlama-Python   | 7B         | 53.5          | 60.4     |
> | **CodeLlama-Python-34B as the Base model** |                    |            |               |          |
> | CodeLlama-Python                           | -                  | 34B        | 51.8          | 67.2     |
> | WizardCoder                                | CodeLlama-Python   | 34B        | 73.2          | 73.2     |
> | Our-Coder-SFT                              | CodeLlama-Python   | 34B        | 72.7          | 72.3     |
> | Our-Coder-RL-ORM                           | CodeLlama-Python   | 34B        | 74.5          | 73.7     |
> | Our-Coder-RL-PRM                           | CodeLlama-Python   | 34B        | 76.8          | 76.2     |
>
>
>
> > **Questions-1**: I find Figure 1 confusing. Why is there a pyramid in the top left and why is it pointing to a pie chart, cube, etc? What are these supposed to be showing? I feel like I am not understanding much from this figure.
>
> We sincerely apologize for any confusion or inconvenience caused by the current presentation of Figure 1. The pyramid in the top-left corner represents the original seed instructions, while the pie chart, cube, and other icons symbolize the evolved instructions generated through the Math Evol-Instruct method, encompassing both upward and downward evolution. A detailed explanation of the training process depicted in Figure 1 was provided in our response to Weaknesses-2-Q4 above, which we hope will help clarify any uncertainties. In the future camera-ready version of our paper, we will provide a more detailed explanation of Figure 1 to ensure its clarity and comprehensibility. We greatly appreciate your understanding and patience.

---

> ### Author Response · Authors · 2024-11-28
>
> > **Questions-2**: I feel like there's a few missing entries in Table 1. For example, Table 1 shows the results for WizardMath-Mathstral and WizardMath-Qwen2.5, but the base scores of these base models are not shown in the table, so the readers don't really know how much improvement there is.
>
> We sincerely appreciate your constructive feedback. The table below supplements the performance comparison of Mathstral-7B-v0.1-Base, Qwen2.5-7B-Base, Qwen2.5-Math-1.5B-Base, and Qwen2.5-Math-7B-Base on the GSM8k and MATH datasets.
>
> The results demonstrate that using Mathstral-7B-v0.1-Base as the base model, WizardMath-Mathstral improves performance by 16.7% on GSM8k (93.8 vs. 77.1) and 14.5% on MATH (70.9 vs. 56.6).
>
> When employing Qwen2.5-Math-1.5B-Base as the base model, WizardMath-Qwen2.5-Math-1.5B achieves 9.9% improvement on GSM8k (86.7 vs. 76.8) and 18.8% on MATH (68.6 vs. 49.8).
>
> Similarly, with Qwen2.5-Math-7B-Base, WizardMath-Qwen2.5-Math-7B shows a 2.3% increase on GSM8k (93.9 vs. 91.6) and 22.4% on MATH (77.8 vs. 55.4).
>
> Finally, using Qwen2.5-7B-Base as the base model, WizardMath-Qwen2.5-7B improves by 8.6% on GSM8k (94.0 vs. 85.4) and 24.7% on MATH (74.5 vs. 49.8).
>
> Notably, both Mathstral-7B-v0.1-Base and Qwen2.5-Math-Base, pre-trained on extensive mathematical corpora, exhibit robust mathematical reasoning capabilities and deliver strong performance on GSM8k and MATH datasets. However, our proposed RLEIF method achieves substantial performance enhancements even with these highly math-optimized models. Specifically, on the MATH, RLEIF delivers a performance boost of 15%–25%, while on GSM8k, the improvement ranges from 8%–16% (with the exception of Qwen2.5-Math-7B-Base, which achieves a high baseline of 91.6 on GSM8k but still benefits from a 2.3% enhancement). These results underscore the continuous improvement enabled by our RLEIF method on models pre-trained with specialized mathematical corpora, further validating its effectiveness and scalability.
>
> | **Models**                                   | **Base**             | **Params** | **GSM8k** | **MATH** |
> |----------------------------------------------|----------------------|------------|-----------|----------|
> | **Mathstral-7B-v0.1-Base as the Base model** |                    |            |               |          |
> | Mathstral-v0.1-Base                          | -                    | 7B         | 77.1      | 56.6     |
> | WizardMath-Mathstral                         | Mathstral-v0.1-Base  | 7B         | 93.8      | 70.9     |
> | **Qwen2.5-Math-1.5B-Base as the Base model** |                    |            |               |          |
> | Qwen2.5-Math-Base                            | -                    | 1.5B       | 76.8      | 49.8     |
> | WizardMath-Qwen2.5-Math                      | Qwen2.5-Math-Base    | 1.5B       | 86.7      | 68.6     |
> | **Qwen2.5-Math-7B-Base as the Base model**   |                    |            |               |          |
> | Qwen2.5-Math-Base                            | -                    | 7B         | 91.6      | 55.4     |
> | WizardMath-Qwen2.5-Math                      | Qwen2.5-Math-Base    | 7B         | 93.9      | 77.8     |
> | **Qwen2.5-7B-Base as the Base model** |                    |            |               |          |
> | Qwen2.5-Base                                 | -                    | 7B         | 85.4      | 49.8     |
> | WizardMath-Qwen2.5                           | Qwen2.5-Base         | 7B         | 94.0      | 74.5     |
>
> > **Typos**: L45: struggles --> struggle ; L80: "in recent"; L91: should say what is IRM first before using the acronym. ; L91: We --> capitalization ; L93: later --> latter ; L105: Jiang et al mentioned without a model (should be mistral?); L107: as following --> as follows ; L140,143: spacing ; L145: should be Reinforcement Learning for Large Language Models instead of the other way around? ; L487: "reasoing"
>
> We sincerely appreciate your effort in identifying these typos and poorly-worded sentences in our paper, as well as your thorough and thoughtful review. All of these "Typos" have been carefully addressed and corrected in our latest upload of revised paper.
>
> **In the latest uploaded revision, we have added the above detailed discussions with Reviewer-dHFe regarding our paper's weaknesses and questions in Appendix C.4  (pages 49–57, lines 2606–3077). Due to time and space limitations, we commit to integrating these discussions into the relevant sections of the main text in future camera-ready version of our paper.**
>
> We hope that the responses above can address your concerns. We eagerly await any further feedback you may have and would be more than happy to engage in additional discussions or respond to any further comments. Thank you once again for your invaluable contributions to our work and for your careful and thorough review of our paper.
>
> Respectfully,
>
> Paper 4894 Authors.

---

### Meta-Review · Area_Chair_j6HP · 2024-12-20

**Metareview:**

This paper takes the Evol-Instruct method invented in the WizardLM paper and applied that to the math domain to create a strong math SFT dataset. Then the authors also perform reinforcement learning to further boost the performance.

This paper receives a high score of 8, 8, 8, 8 and I have no doubt to accept this paper.

**Additional Comments On Reviewer Discussion:**

The reviewers mainly complain about fair comparison with the baselines, such as comparing RL variants to SFT-only models is unfair, evol-instruct may not perform better than other SOTA SFT datasets if SFT only, and the RL parts should be emphasized more. The authors added comprehensive experiments to show that the constructed SFT dataset is able to outperform other open-source datasets in apple-to-apple comparison.

---

### Decision · Program_Chairs · 2025-01-22

Accept (Oral)